# Low-Complexity Methods to Mitigate the Impact of Environmental Variables on Low-Cost UAS-based Atmospheric Carbon Dioxide Measurements

Gustavo B. H. de Azevedo[1,2*], Bill Doyle[2], Christopher A. Fiebrich[3], and David Schvartzman[1,4]

[1]Advanced Radar Research Center (ARRC) at The University of Oklahoma, Norman, OK, USA
[2]Center for Autonomous Sensing and Sampling (CASS) at The University of Oklahoma, Norman, OK, USA
[3]Oklahoma Mesonet, Oklahoma Climatological Survey at The University of Oklahoma, Norman, OK, USA
[4]School of Meteorology at The University of Oklahoma, Norman, OK, USA

**Correspondence:** Gustavo B. H. de Azevedo (gust@ou.edu)

**Abstract.** This article assesses the individual and joint impact of pressure, temperature, and relative humidity on the accuracy of atmospheric $CO_2$ measurements collected by Unmanned Aerial Systems (UAS) using low-cost commercial Non-Dispersive Infrared sensors (NDIR). We build upon previous experimental results in the literature and present a new dataset with increased gradients for each environmental variable to match the abrupt changes found in UAS-based atmospheric vertical profiles. As a key contribution, we present a low-complexity correction procedure to mitigate the impact of these variables and reduce errors in this type of atmospheric $CO_2$ measurement. Our findings support the use of low-cost NDIR sensors for UAS-based atmospheric $CO_2$ measurements as a complementary in-situ tool for many scientific applications.

## 1 Introduction

Over the past two decades, Unmanned Aerial Systems (UAS) have grown as a complementary in-situ observation tool for local atmospheric $CO_2$ profiles (Villa et al., 2016). This growth is justified by the relatively low cost of UAS and its ability to provide atmospheric $CO_2$ measurements with high spatiotemporal resolution (Piedrahita et al., 2014). In a literature survey, Villa et al. (2016) highlight other motivations, such as in-situ validation of remote instruments, autonomous plume tracking, and locating hazardous emission sources. In many of these applications, the low-cost aspect of UAS-based solutions is crucial to the application's feasibility (Nelson et al., 2019; Cartier, 2019; Kunz et al., 2018; Martin et al., 2017; Mitchell et al., 2016; Kiefer et al., 2012; Yasuda et al., 2008; Watai et al., 2006). In addition, the sensor's size, weight, and power requirements are also critical to the design of UAS-based solutions (Martin et al., 2017). For these reasons, many UAS-based atmospheric $CO_2$

---

*Gustavo B. H. de Azevedo is now with the Unmanned Systems Research Institute at Oklahoma State University, Stillwater, OK, USA.
Email: gus@okstate.edu

measurement systems use commercial low-cost Non-Dispersive Infrared (NDIR) sensors (B. H. de Azevedo, 2020; Kunz et al., 2018; Martin et al., 2017; Gibson and MacGregor, 2013; Stephens et al., 2011; Yasuda et al., 2008; Pandey and Kim, 2007; Watai et al., 2006; Chen et al., 2002). However, abrupt changes in pressure, temperature, and relative humidity associated with atmospheric vertical profiles can interfere with low-cost NDIR $CO_2$ sensors.

In this article, we review the main concerns regarding the use of commercial low-cost NDIR sensors for atmospheric $CO_2$ measurements found in the literature. We then build upon previous experimental results in the literature by investigating the impact of each environmental variable on low-cost NDIR $CO_2$ sensors. We also present a new dataset with stronger rates of change than previously found in the literature. These stronger rates of change are obtained by increasing the span of change in the test variables and decreasing experimental time scales. Finally, we evaluate if a set of low-cost and simple benchtop procedures can be used to characterize and mitigate the impact of these variables on the same sensors. All the experiments in this article were performed with low-complexity and repeatable methods. These methods used reference gas analyzers, non-gas specialized environmental chambers, and resources accessible to most researchers. The methods demonstrated were capable of correcting the measurements of low-cost NDIR sensors to a few ppm of more expensive reference benchtop gas analyzers. We believe these low-complexity procedures are a way to lower the entry barriers to this research field while improving the accuracy of UAS-based $CO_2$ measurements.

## 1.1 Background and Motivation

Many low-cost NDIR $CO_2$ sensors are available on the international market (Tab. S1, in the supplement, lists a few examples with some basic specifications). Besides the attractive low cost, most of these sensors are also lightweight and have low power requirements. However, as shown in Tab. S1, the errors reported by their manufacturers are larger than what might be measured as the maximum concentration variation when performing an atmospheric vertical profile. To mitigate this accuracy issue, some researchers investigated methods to characterize and correct them in post-processing (Ashraf et al., 2018; Martin et al., 2017; Gaynullin et al., 2016; Yasuda et al., 2012; Mizoguchi and Ohtani, 2005). In some cases, accuracy was improved from $\pm 30$ ppm to $\pm 1.9$ ppm (Martin et al., 2017). However, according to Kunz et al. (2018), the improvements achieved by Martin et al. (2017); Piedrahita et al. (2014); Yasuda et al. (2012); Mizoguchi and Ohtani (2005) are not applicable to UAS-based sampling due to the stronger rates of change in pressure, temperature, and relative humidity associated with UAS-based atmospheric profiles. Recent publications, such as Arzoumanian et al. (2019), partially address these concerns by increasing the variation range of the test variables. However, these newer results may also not be valid for UAS-based applications due to their longer time scales. Results from Arzoumanian et al. (2019); Martin et al. (2017); Piedrahita et al. (2014); Yasuda et al. (2012); Mizoguchi and Ohtani (2005) were obtained for experiments done in months, weeks, and days. The changes in pressure, temperature, and relative humidity associated with UAS atmospheric profiles occur in the time scales of minutes.

In their work, Martin et al. (2017) present a comparison between a sequential method to correct the impacts of pressure, temperature, and humidity versus a joint correction method using multivariate linear regression. This provides some insight into the impact of each variable on low-cost NDIR $CO_2$ sensors. However, important questions for UAS-based measurements remain unanswered. For example, is the $0.1$ ppm improvement in RMSE for temperature corrections (sequential method) a

factor of the small impact of temperature on NDIR $CO_2$ sensors or a factor of the small range of temperatures tested? Was the impact of temperature obfuscated by the larger impact of pressure? Even though a realistic method to mitigate the impact of environmental variables on low-cost NDIR $CO_2$ sensors for UAS-based measurements should account for the joint variation of pressure, temperature, and relative humidity, understanding the isolated behavior of each variable is important to inform the design of UAS-based sensor packages. This knowledge can help system developers address some of these measurement issues during the sensor package design phase (e.g., heat shielding), thus reducing issues to be corrected in post-processing.

Another motivation for isolating the impacts of each of these three variables is the study of the impact of relative humidity. Many low-cost systems for atmospheric $CO_2$ measurements rely on desiccants to eliminate errors induced by variations in relative humidity. Therefore, there are few correction methods for this variable in the literature. Understanding the impact of this variable is crucial for UAS-based applications due to the design impacts in aircraft size, weight, and power, from the addition of a desiccant compartment. Desiccants need to be replaced periodically. Thus, their placement choice on the aircraft is limited by their accessibility requirement. Furthermore, a desiccant container creates an additional air-volume in the measurement system, which can impact the spatiotemporal resolution of UAS-based systems. Finally, using of desiccants in UAS-based applications implies on the use of pumps to actively control the system's airflow. The use of pumps increases the total system weight and power requirements when compared to ram-air solutions.

Finally, any system used to support long-term research or forecast operations should also account for temporal drift and sensor decay. In the case of UAS-based applications, this decay may happen in short periods due to the intense exposure to the elements and the amount of dust collected during aircraft take-off and landing. Sensor decay periods vary with application and require a case-by-case length determination. Therefore, another concern regarding the adoption of the correction methods currently available in the literature is their complexity. Most of the correction methods for low-cost NDIR $CO_2$ sensors available in the literature rely on periodic recalibration using a traceable gas canister. These can be done either through complex laboratory setups or day-long field calibrations using ambient pressure and temperature variations. Although there is no question that traceable gas canisters provide the most precise means of calibration and correction, this method is not practical for UAS-based field applications. Certainly, a UAS-based measurement system can be calibrated in a laboratory before and after a field campaign. Nevertheless, for field operations involving multiple flights per day over multiple days, a low-complexity method using a reference gas analyzer may be beneficial for field calibrations.

In this study, we attempt to address some of the abovementioned concerns. First, we test a low-complexity method using a reference gas analyzer on a chamber setup to study the isolated impacts of pressure, temperature, and relative humidity on low-cost NDIR $CO_2$ sensors. Then, we evaluate a low-cost benchtop setup to characterize and correct the impact of these variables on the same sensors. For all of these experiments, we attempt to increase the test range for each variable and reduce the experiment time scales. More details on each experiment and their results are shown in sections 2, 3, and 4.

## 2 Methodology

As mentioned previously, the strong rates of changes in pressure ($P$), temperature ($T$), and relative humidity ($RH$) associated with UAS-based atmospheric measurements can interfere with low-cost NDIR $CO_2$ sensors. For a given test variable, these rates of change are determined by the number of units changed per time interval (e.g., $\Delta P/\Delta t$, $\Delta T/\Delta t$, and $\Delta RH/\Delta t$, where $t$ is the time). In this study we are interested in variations between 10 and 45 °C, 5 and 95 %RH, and 60,000 and 101,325 Pa[2] that occur in time intervals from 10 to 120 minutes. We have chosen these intervals based on the performance limitations of most of the commercially available low-cost UAS and the sampling pattern recommendations for UAS-based measurements found in the literature (Houston and Keeler, 2018; Hemingway et al., 2017). We are aware there is interest in UAS-based sampling of atmospheric $CO_2$ outside of these intervals. However, as they may fall outside the capabilities of low-cost NDIR $CO_2$ sensors and low-cost UAS, they are not the focus of this study.

Besides the desire to characterize and mitigate the impact of pressure, temperature, and relative humidity on low-cost NDIR $CO_2$ sensors, this study is also focused on performing this task via a low-complexity method that would be accessible to a larger portion of the scientific community and industry. Therefore, the experiments in this study were performed via comparison to a calibrated reference gas analyzer. This strategy eliminates the need for traceable gas canisters and their plumbing and chamber-sealing requirements while increasing the number of potential instruments to produce the desired changes in pressure, temperature, and relative humidity. Nonetheless, it is important to note that this strategy is limited to producing results relative to the reference gas analyzer. It is also important to note that the selected reference gas analyzer must be independent of changes in the test variables within the test range. More details about this requirement and other limitations of this method can be found in sections 2.2, 3, and 4.

The experiments in this article were organized into two parts. The first part is a collection of experiments done in the environmental chambers of the Oklahoma Mesonet's calibration laboratory. These experiments provide a baseline of the impact of each variable on low-cost NDIR $CO_2$ sensors and an initial evaluation of the correction methods based on a reference gas analyzer. The second part is a collection of low-cost benchtop experiments performed to evaluate if a method using a reference gas analyzer and limited resources can be developed for field calibrations. A complete list of experiments can be found on Tab. S3 (see supplement). The following subsections detail the selection process of the low-cost NDIR $CO_2$ test sensors and the characteristics of the selected reference sensors.

### 2.1 Test Sensors

Due to the large number of low-cost NDIR-based $CO_2$ sensors available and the unfeasibility of evaluating all of them, we searched the literature for model comparison studies and the rate of adoption of each model. We used this methodology to select a model that would represent the current state of the art for low-cost UAS-based atmospheric $CO_2$ sampling. In a comparison study, Yasuda et al. (2012) evaluated five different models and concluded that the Senseair K30 NDIR $CO_2$ sensor offered the

---

[2]This pressure interval may seem large for low-cost commercially available UAS. However, these pressures are commonly experienced for UAS flights at elevated locations. For example, flights near Boulder, Colorado (Barbieri et al., 2019)

best combination of cost, weight, and accuracy among the models considered. A similar result was found by Al-Hajjaji et al. (2017), who compared five other sensors to the K30.

The adoption of the K30 for UAS-based measurements was compared to the adoption of other models by their use in the reviewed literature, and the adoption of these sensor models in the literature was evaluated through a search on the GoogleScholar™ database. This search followed the method from the literature review on UAS-based gas sampling by Villa et al. (2016). The list of search terms and resulting analysis can be found in Tab. S2. The analysis suggests that the K30 is more prevalent in the literature than the other models tested by Yasuda et al. (2012) and Al-Hajjaji et al. (2017). For these reasons, all experiments in this article were performed with the Senseair K30 NDIR $CO_2$ sensor.

Neither the Senseair K30 nor the other low-cost NDIR $CO_2$ sensors evaluated by Yasuda et al. (2012) and Al-Hajjaji et al. (2017) were designed for UAS-based deployment. Their optical chambers assume a natural air exchange with the environment over a long period (minutes to hours). This design characteristic creates an artificially slow time-response. To mitigate this issue, some manufacturers offer optional airflow intakes for the sensors (e.g., CO2Meter's pump cap for the K30), and some researchers design custom sensor housings to control airflow and integrate the sensors into the aircraft. These custom sensor housings, such as the ones shown in B. H. de Azevedo (2020), can improve the sensor time response from $30\,\mathrm{s}$ to approximately $1$ s (under $0.5\ \mathrm{Ls^{-1}}$ flow). However, it is important to note that spatiotemporal results from systems using this technique are averaged and assume some degree of spatiotemporal homogeneity. Furthermore, sensor housings can directly impact the propagation of changes in temperature and relative humidity from the environment to the sensors.

Even though the impacts of sensor housing design are not within the scope of this study, the evaluation of a method to mitigate environmental variables on UAS-based measurements that did not consider the requirements of UAS-based sensor deployment would not be complete. For this reason, we collocated test sensors in different housing configurations whenever the chamber space allowed for. In total, we used three housing configurations. The first housing configuration is a simple box of approximately $200\,\mathrm{mL}$ that houses two K30 units and an IST HYT-271[3] temperature and humidity sensor. The second configuration is similar to the first but has its volume reduced to only expose the optical chambers of the two K30 and the HYT-271 sensor to the controlled airflow. Its volume is approximately $8\,\mathrm{mL}$. Both configurations use a $0.5\ \mathrm{Ls^{-1}}$ diaphragm pump to control the airflow in and out of the housing. Details for the shape and design of both sensor housings can be found in B. H. de Azevedo (2020). The third and final configuration has two exposed K30 sensors without any sensor housing, and it serves as a control.

This strategy was adopted to increase the confidence in the results obtained and evaluate considerations found in the literature regarding the need for distinct correction coefficients for each sensor unit. Finally, it is important to note that all results and analyses in this article considered only the $CO_2$ concentration values reported by each sensor unit. In other words, each unit was assumed to be immutable from its factory-performed calibration. Therefore, no attempts were made to analyze and correct the light absorption signals within the K30. Instead, each sensor unit was evaluated and corrected as a "black-box". This method was adopted to evaluate if these sensors could produce satisfactory results only with post-processing techniques.

---

[3]https://www.ist-ag.com/en/products/humidity-module-hyt-271-pluggable-sil-contacts

## 2.2 Reference Sensors

The reference gas analyzers used in this study were the LI-COR LI-840A and LI-820. These gas analyzers served as a control for the experiments because they are also light-based sensors, but they use sample conditioning and auxiliary sensors to eliminate interference from pressure, temperature, and humidity. Both sensors heat the sampled air to 50 °C before measuring its $CO_2$ concentration. Therefore, the temperature variations tested in this study do not affect their measurements. Both sensors measure the pressure inside their optical chambers and use algorithms for active compensation. However, only the LI-840A measures $H_2O$ (mmol/mol) for algorithmic compensation. For this reason, the LI-840A was used as the comparison reference, placed inside the test volumes with the test sensors (when the designed experiment allowed for it), and the LI-820 monitored the ambient near the experiment for potential variations in the experimental conditions.

Monitoring the ambient conditions near the experiments is important for this comparative study because the unsealed chambers and benchtop setups used can be affected by external increases in $CO_2$. These chambers and benchtop-setups take ambient air and condition it to create the desired test conditions (e.g., heating the air). Due to this experimental limitation, we reduced external sources of $CO_2$ and monitored the ambient conditions near the test chambers to ensure that pressure, temperature, relative humidity, and $CO_2$ did not change significantly during the experiments. This article's supplement shows the ambient conditions for all experiments in this study and two comparison experiments between the test and reference sensors (see Fig. S4). More details on specific experimental setups are given in sections 3 and 4.

## 3 Chamber Experiments

To investigate the impact of pressure, temperature, and relative humidity on low-cost NDIR $CO_2$ sensors and evaluate the correction method based on a reference gas analyzer, we performed five chambered experiments at the Oklahoma Mesonet Calibration Laboratory. The environmental chambers of the Oklahoma Mesonet Calibration Laboratory are not specialized for gas experiments and present many similarities to other environmental chambers found in other universities and research laboratories. The two chambers used for these experiments were the Thunder Scientific 2500 and the Cincinnati Sub-Zero Z16. This particular Z16 was outfitted with a custom gasket-based vacuum and compression system, developed by the laboratory's manager, David L. Grimsley. A description of the Oklahoma Mesonet and its facilities can be found in McPherson et al. (2007).

### 3.1 Pressure

The pressure dependence experiment performed at the Oklahoma Mesonet Calibration Laboratory used the Cincinnati Sub-Zero Z16 chamber and its custom gasket-based vacuum and compression system. This system produced a pressure variation from 105,000 to 60,000 Pa, in 1,000 Pa increments, at 25 °C. Each pressure change was followed by a two-minute dwell period. Even though the Cincinnati Sub-Zero Z16 chamber can control temperature and humidity, its controlled conditions are not reflected inside the custom pressure system. This occurs because the Thompson vacuum and compression pumps on the Mesonet's custom pressure system use air from outside the controlled chamber. Therefore, in this experiment, the temperature

control is limited to the impacts caused by keeping the entire Mesonet custom pressure system at the chamber's temperature. This setup also does not allow for any active control of relative humidity. This characteristic also means that changes in $CO_2$

concentration near the chamber could affect the experiment. This type of contamination can create effects that obfuscate the effects of pressure. To mitigate this problem, we reduced the experiment's duration to the pressure system's limits and used the LI-840A to monitor potential contaminations and validate the experimental conditions.

Although the LI-840A pressure compensation range is specified within 15,000 and 115,000 Pa, we chose not to connect this reference sensor to the pressure system based on a consultation with an LI-COR engineer. In this consultation, we were

185 informed that the compensation algorithm could fail for large pressure changes in short intervals. To avoid any problems, the reference sensor was placed adjacent to the intake and exhaust nozzles of the vacuum and compression pumps (as shown in Fig. 1). This placement still allowed us to monitor the parameters of the air used by the pressure system to produce the changes on the test sensor. The metrics for the experimental conditions can be seen on Tab. S10. In this experiment, it was only possible to deploy the K30 sensors using the first housing configuration (200 mL box, see section 2.1 for more details). This limitation

was created by the connection requirements of the Mesonet custom pressure system.

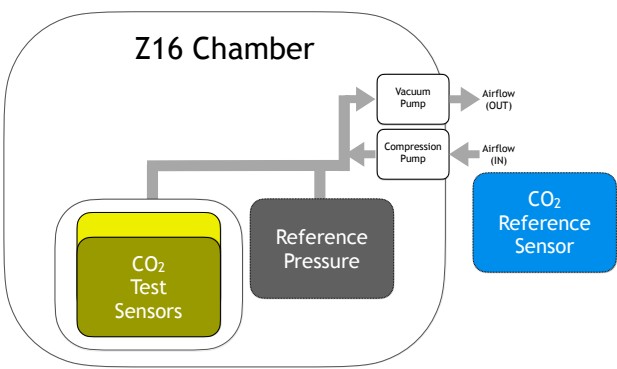

**Figure 1.** Diagram for the Pressure Chamber Experiment. Two test sensors were placed inside the chamber, and a reference sensor was placed outside to indicate possible contamination and monitor the experimental conditions.

As can be seen on Tab. S10, the HYT-271 sensor inside the K30 sensor housing reported standard deviations of 0.98 %RH for relative humidity and 0.03 °C for temperature. This indicates that the majority of the 230 ppm change, seen in Fig. 2, in both test sensors was caused by the 45,000 Pa change in pressure. This result is impressive considering that the air used by the pumps to produce these pressures showed only a 2.55 ppm standard deviation for $CO_2$ during the same period.

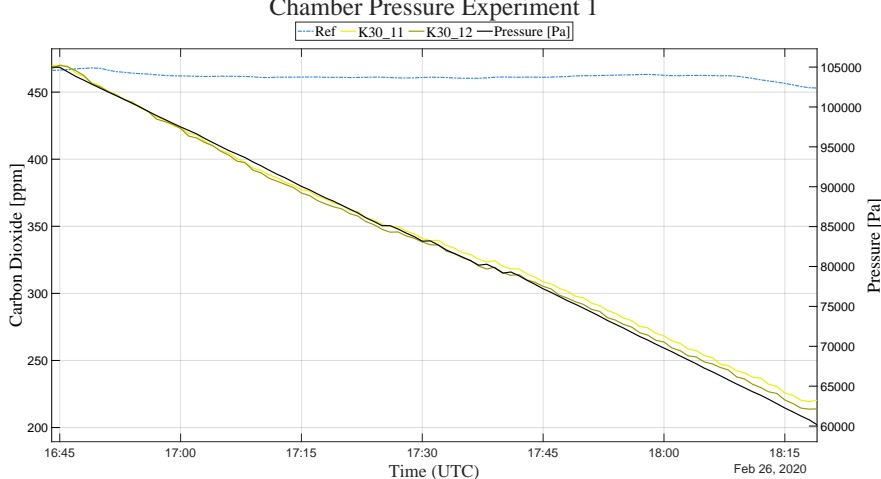

**Figure 2.** Time-series data for the Pressure Chamber Experiment. The solid black curve represents the pressure inside the chamber. The yellow curves represent the $CO_2$ values reported by the test sensors. The dashed blue curve represents the $CO_2$ values reported by the reference sensor.

### 3.1.1 CO₂ Pressure Correction

Within the NDIR sensor literature, the article by Gaynullin et al. (2016) offers an excellent description of the determination of the pressure correction coefficients for the Senseair K30 NDIR $CO_2$ sensor. According to the authors, the $CO_2$ concentration reported by the sensors can be corrected by the following equation,

$$PPM_{corrected} = \frac{PPM_{measured}}{k_1 * (P - P_0)^3 + k_2 * (P - P_0)^2 + k_3 * (P - P_0) + k_4}, \tag{1}$$

where the coefficients $k_1$ through $k_4$ need to be determined for each sensor unit, and $P_0$ is 101,325 Pa. In their article, Gaynullin et al. (2016) report a maximum deviation between the corrected and true value between 2 and 4 ppm. However, their results were obtained using a complex multilayered chamber that pressurized a reference gas. Unfortunately, such an experimental setup is not practical for low-cost UAS-based applications. In this section, we evaluate the feasibility of determining the pressure correction coefficients using the $CO_2$ values measured by the reference gas analyzer. We assume the low variability in pressure, temperature, and humidity found in a short-duration experiment mimics the controlled conditions found in Gaynullin et al. (2016). Our results for this first evaluation can be found in Fig. 3 and Tab. 1.

Over the span of 45,000 Pa, the maximum absolute errors (MxAE) reported by the test sensors were 8.7 and 8 ppm, and the root mean squared errors (RMSE) were 2.15 and 1.91 ppm. These are considerable improvements over the original 233.9 and 239.65 MxAE and the 140.09 and 143.75 RMSE. Nonetheless, it is important to highlight that these results are not absolute. They are relative to the values reported by the reference gas analyzer. Unfortunately, the test sensors were damaged after this

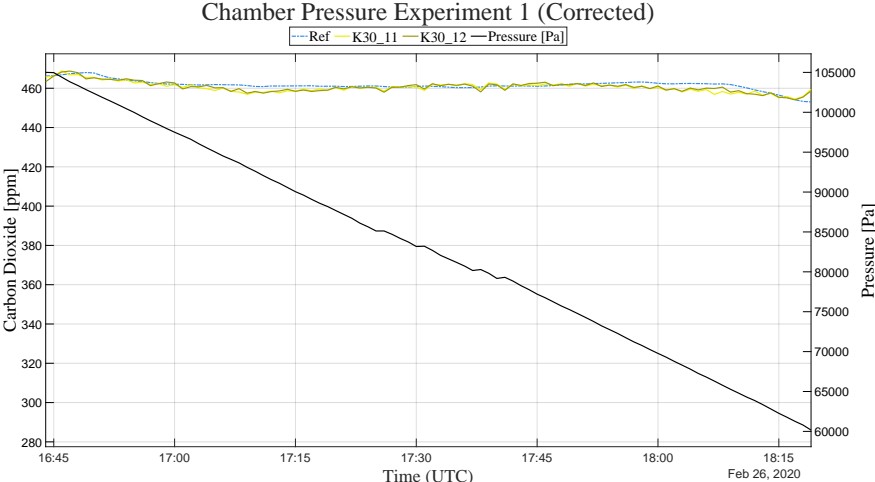

**Figure 3.** Time-series data for the Pressure Chamber Experiment after the application of the correction method. The solid black curve represents the pressure inside the chamber. The yellow curves represent the $CO_2$ values reported by the test sensors. The dashed blue curve represents the $CO_2$ values reported by the reference sensor.

**Table 1.** Coefficients from the pressure correction method and root mean square errors for the test sensors relative to the reference sensor, before and after the correction.

| Sensor | $k_1$ | $k_2$ | $k_3$ | $k_4$ | $R^2$ | RMSE Before | RMSE After |
|--------|-------|-------|-------|-------|-------|--------|-------|
| **K30_11** | -2.3291e-16 | 4.1525e-12 | 1.2380e-05 | 1.0648 | 0.9995 | 140.09 | 2.15 |
| **K30_12** | -3.4693e-16 | 2.8776e-12 | 1.2778e-05 | 1.0706 | 0.9996 | 143.75 | 1.91 |

experiment and a second validation run was not possible. However, the results for four other cases using this method on the low-cost bench setup are reported in section 4.1.

## 3.2 Temperature

The temperature dependence experiment performed at the Oklahoma Mesonet Calibration Laboratory used the Thunder Scientific 2500 chamber to produce a temperature variation from 10 to 40 °C, in ten-degree increments, at a constant 45 %RH. In this experiment, the temperature is slowly raised from 10 to 40 °C in approximately 210-minutes, then reduced back to 10 °C in approximately 90-minutes. The operational limits of the chamber defined these time intervals. Nonetheless, this experiment setup allows us to acquire many samples for each temperature and produces conditions that match UAS flight conditions in the final 90-minutes.

The Thunder Scientific 2500 chamber uses a three-chamber system where air from the laboratory is taken and conditioned to the desired set points in the first two inner chambers and then inserted into the test chamber. Besides the potential for external interference through the chamber's air intake, the chamber's test volume also has a cable port that is only partially closed. To counter this external potential external interference, the chamber constantly corrects small changes in temperature and relative humidity, but it offers no control over pressure. For our experiments, besides any pressure-induced changes in the

reported $CO_2$, actual concentration changes in the laboratory taken in by the chamber can also obfuscate the impacts of the test variables. To mitigate potential contamination, we reduced the experiment's duration to the chamber's operational limits and performed our experiments overnight when there were no people in the laboratory. To validate the experimental conditions, we adopted a strategy similar to the one used for the pressure experiment. However, in this case, the LI-840A reference sensor (Ref) was colocated with the test sensors inside the chamber, and the LI-820 (Ref_Lab) was placed near the chamber's air

intake to monitor the experimental conditions. Fig. 4 illustrates this sensor arrangement.

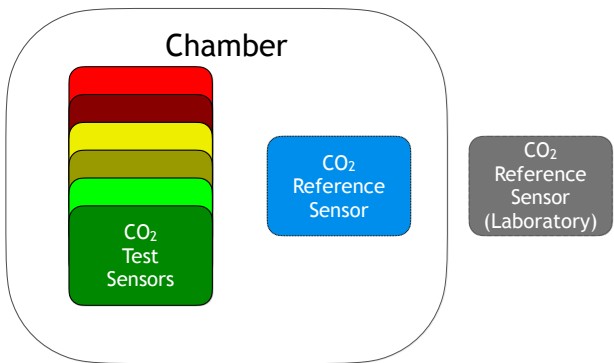

**Figure 4.** Diagram for the sensor placement during the temperature and relative humidity experiments at the Oklahoma Mesonet Calibration Laboratory. Six test sensors were placed inside the chamber with a reference sensor, and another reference sensor was placed outside to detect possible contamination.

In this experiment, we used six K30 test sensors, organized in three pairs, following the three sensor housing configurations detailed in section 2.1. The sensors labeled K30_13 and _14 (Test System 1) are in the third configuration (without sensor housing) and serve as a control. As can be seen on Tab. S11, the HYT-271 sensors inside the K30 sensor housings for Test Systems 2 and 3 reported standard deviations of 1.3 and 1.51 %RH. The pressure sensors for all three test systems reported an

235 average standard deviation of 135 $\mathrm{Pa}$. During the same period, the reference sensor inside the chamber showed a 4.02 $\mathrm{ppm}$ standard deviation for $CO_2$. This leads us to believe that the majority of the 36 $\mathrm{ppm}$ change seen in five of the six test sensors (Fig. 5) was caused by the 30 $^\circ$C change in temperature.

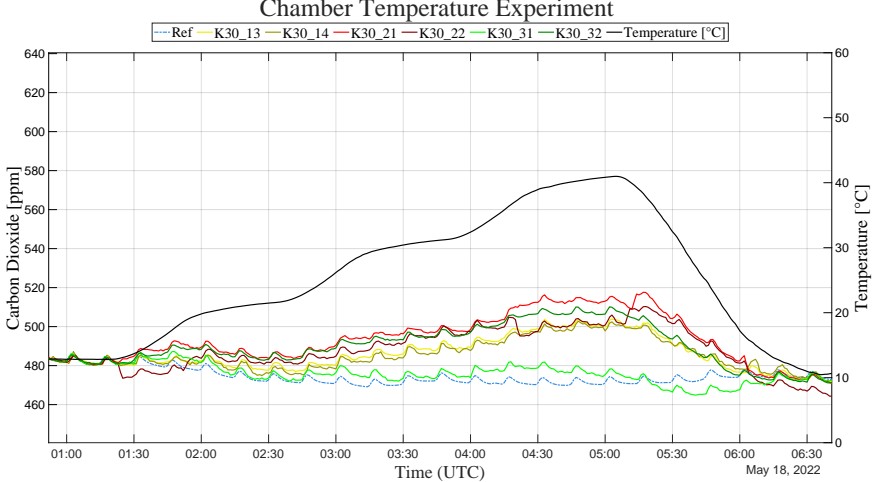

**Figure 5.** Time-series data for the chambered temperature experiment. The solid black curve represents the temperature inside the chamber. The yellow, green, and red curves represent the $CO_2$ values reported by the test sensors. The dashed blue curve represents the $CO_2$ values reported by the reference sensor.

### 3.2.1 $CO_2$ Temperature Correction

Many of the authors cited in section 1.1 employ a linear regression to correct the impacts of temperature on low-cost NDIR $CO_2$
sensors. However, the fast reduction from 40 to. $10\,°\mathrm{C}$ in our experiments produced some variations in the $CO_2$ values reported by the test sensors that were better captured by a cubic fitting (similar to the one presented in section 3.1.1, with $T_0 = 15\,°\mathrm{C}$). This cubic-like behavior could be a function of small variations in the other variables (e.g., pressure and $CO_2$), given that our simplified setup does not actively control them. However, the small scale of the variations in pressure and $CO_2$ during the experiment lead us to suspect other sources (e.g., a temperature time response effect). Unfortunately, our experimental setup
does not allow us to investigate this variation further. Table 2 shows the coefficients and the R-squared for the fitting, and the RMSE relative to the reference sensor, before and after the correction. Figure 6 shows the time series for the corrected test sensors.

After we determined the coefficients for each sensor, we also used them to correct the data obtained by another run of the same experiment (a test run). This independent test allows us to better evaluate the method's performance. The plots and tables
with the data for the test run of the chambered temperature experiment can be found in this article's supplement (S11 through S14). The RMSE for the test run can be found here in Tab.2. These two experiments demonstrate satisfactory error reductions for all sensors except for K30_31. This sensor did not seem to respond to temperature in the same manner as the other five test sensors. Evaluating the behavior of the K30_32, which was placed in the same sensor housing as the K30_31 and behaved similarly to all other sensors, we can eliminate any housing-induced effects. Furthermore, the temperature and relative humidity

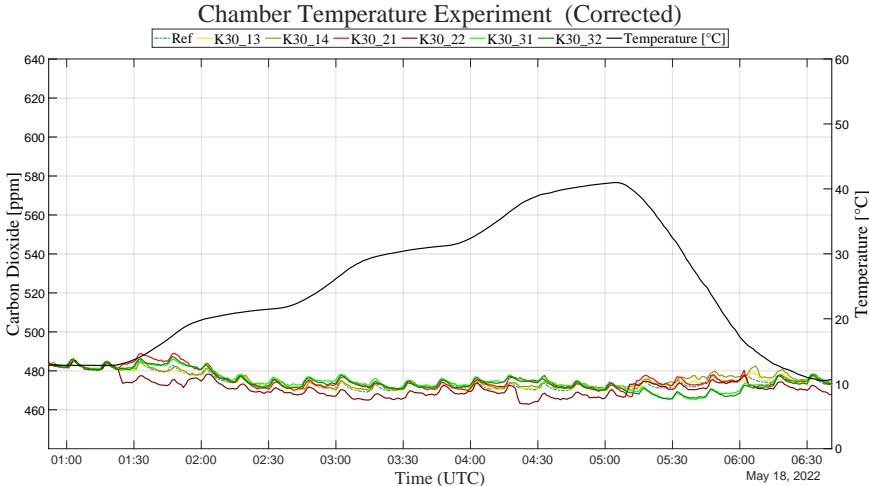

**Figure 6.** Time-series data for the chambered temperature experiment after the application of the correction method. The solid black curve represents the temperature inside the chamber. The yellow, green, and red curves represent the $CO_2$ values reported by the test sensors. The dashed blue curve represents the $CO_2$ values reported by the reference sensor.

**Table 2.** Coefficients for the temperature correction method. The RMSE values are relative to the reference sensor inside the test volume.

| | | | | | | RMSE | | | |
|---|---|---|---|---|---|---|---|---|---|
| | | | | | | Learn | | Test | |
| Sensor | $k_1$ | $k_2$ | $k_3$ | $k_4$ | $R^2$ | Before | After | Before | After |
| K30_13 | -2.7197e-06 | 0.0002 | 0.0023 | 1.0814 | 0.9873 | 15.77 | 1.12 | - | - |
| K30_14 | -2.3352e-06 | 0.0002 | 0.0018 | 1.1381 | 0.9738 | 14.83 | 1.69 | - | - |
| K30_21 | 6.6562e-07 | 2.1862e-05 | 0.0048 | 0.8204 | 0.9823 | 23.99 | 2.38 | 20.84 | 4.09 |
| K30_22 | -5.9848e-06 | 0.0002 | 0.0043 | 1.0170 | 0.9887 | 19.26 | 3.91 | 19.28 | 3.25 |
| K30_31 | 2.0487e-07 | -1.4397 | 0.0008 | 1.2151 | 0.2949 | 4.3 | 2.99 | 5.87 | 4.79 |
| K30_32 | -7.8179e-07 | 6.3524e-05 | 0.0038 | 0.8544 | 0.9554 | 20.24 | 2.88 | 17.96 | 2.88 |

recorded by this test system's internal HYT-271 followed the chamber's state. Therefore, we have to consider the K30_31 as an outlier for these experiments.

### 3.3  Relative Humidity

The chambered relative humidity experiment was also performed on the Thunder Scientific 2500 chamber with the same sensor arrangement described in section 3.2. The two runs for this relative humidity experiment were performed immediately after
each temperature experiment run. This strategy allowed us to use the contamination mitigation techniques in a stable laboratory

environment. In this experiment, the chamber produced a relative humidity (RH) variation from 15 to 85 %RH at a constant 25 °C. In the first 75 minutes, the RH was raised from 15 to 85 %RH and then reduced back to 15 %RH over a 13 minutes interval. Again, these time intervals were defined based on the chamber's operational limitations. For the duration of this experiment, the HYT-271 sensors inside the K30 sensor housings for Test Systems 2 and 3 reported standard deviations of 0.43 and 0.31 °C for temperature, and all three test systems reported an average standard deviation of 94.5 Pa. This indicates that the average 16 ppm increase across the six test sensors was caused by the 70 %RH change in relative humidity (Fig. 7).

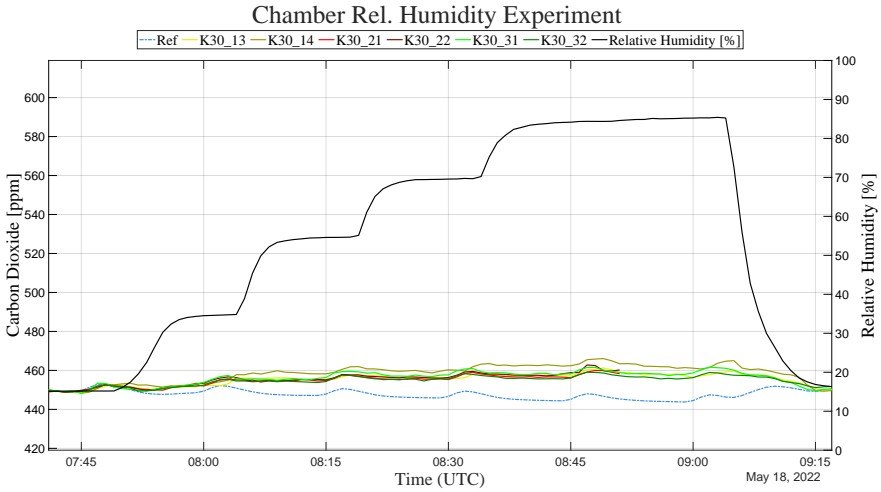

**Figure 7.** Time-series data for the chambered relative humidity experiment. The solid black curve represents the relative humidity inside the chamber. The yellow, green, and red curves represent the $CO_2$ values reported by the test sensors. The dashed blue curve represents the $CO_2$ values reported by the reference sensor.

### 3.3.1 $CO_2$ Relative Humidity Correction

As mentioned in section 1.1, there are few methods in the literature to correct the impact of humidity on low-cost NDIR $CO_2$ sensors. Most of the methods found adopt a simple linear regression correction, but for the reasons mentioned in section 3.2.1, we also adopted a cubic fitting (see section 3.1.1) for our correction. In this case, with $RH_0 = 36\%$. We believe the 70 %RH change in 13 minutes is considerably stronger than any other experiments shown in the literature. Thus, more prone to reveal effects not seen before. Table 3 shows each test sensor's coefficients, the R-squared for the cubic fitting, and the RMSE relative to the reference sensor. Figure 8 shows the results of this correction method.

To better evaluate the method's performance, we repeated the experiment and applied the previously determined correction coefficients to it. This independent test mimics how the method would be applied to correct field data. The plots and tables with the data for the other chambered relative humidity experiment run, the test run, can be found in this article's supplement (S15 through S17). The RMSE for the test run can be found here in Tab. 3. These two experiments demonstrate satisfactory error

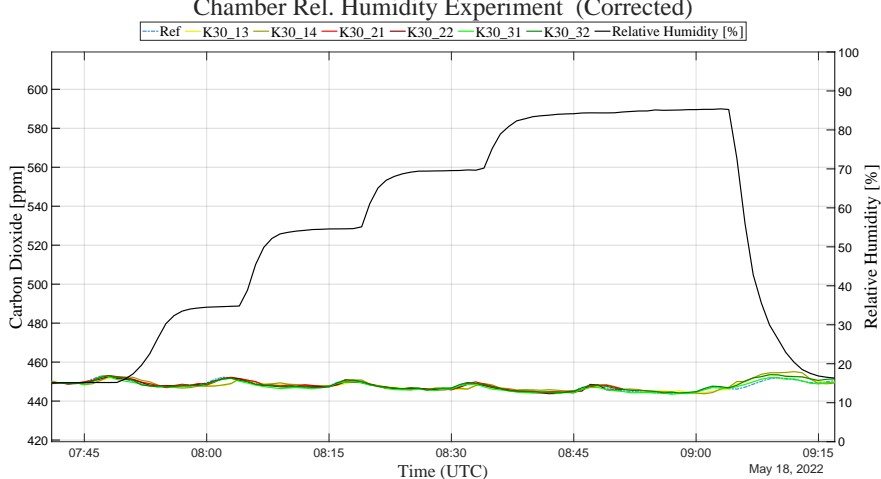

**Figure 8.** Time-series data for the chambered relative humidity experiment after applying the correction method. The solid black curve represents the relative humidity inside the chamber. The yellow, green, and red curves represent the $CO_2$ values reported by the test sensors. The dashed blue curve represents the $CO_2$ values reported by the reference sensor.

**Table 3.** Coefficients from the relative humidity correction method. The RMSE values are relative to the reference sensor inside the test volume.

| Sensor | $k_1$ | $k_2$ | $k_3$ | $k_4$ | $R^2$ | RMSE | | | |
|---|---|---|---|---|---|---|---|---|---|
| | | | | | | Learn | | Test | |
| | | | | | | Before | After | Before | After |
| K30_13 | 8.1074e-08 | -4.6056e-06 | 0.0008 | 1.1071 | 0.9180 | 9.02 | 1.31 | - | - |
| K30_14 | 1.200e-07 | -7.4641e-06 | 0.0011 | 1.1559 | 0.9262 | 12.37 | 1.62 | - | - |
| K30_21 | 2.0367e-07 | -4.5196e-06 | 0.0009 | 0.8858 | 0.9819 | 8.09 | 0.66 | 9.73 | 2.38 |
| K30_22 | 5.2379e-07 | -2.5196e-06 | 0.0009 | 1.0617 | 0.9848 | 8.82 | 0.58 | 10.91 | 2.52 |
| K30_31 | 9.6294e-08 | -5.9337e-06 | 0.0011 | 1.2191 | 0.9864 | 9.72 | 0.63 | 5.36 | 4.81 |
| K30_32 | 1.4859e-07 | -4.9691e-06 | 0.0007 | 0.9179 | 0.9681 | 8.07 | 0.85 | 16.14 | 9.58 |

reductions for all sensors except for the sensors in Test System 3 (K30_31 and K30_32) during the test run. Further evaluating these results, we noted that the method overcorrected these two sensors on the test run. This overcorrection can be explained by the difference in the range of %RH effectively transferred inside the sensor housing between the two experiments. In the first run of the experiment, when the coefficients were determined ("learn" case), the minimum and maximum %RH inside the housing of TS_3 were 12.15 and 68.8 %RH. During the second run, when the coefficients were tested, the minimum and maximum %RH inside the housing were 14.53 and 74.41 %RH.

Even though this particular result may point to potentially negative effects of the sensor housings, we highlight that all four housed sensors outperformed the unhoused (control sensors) in the first run (see Tab. 3). Similar errors, caused by slight differences in experimental conditions between the "learn" and "test" cases, were also seen in the development of the benchtop pressure experiments. This error can be mitigated by increasing the number of "learn" cases presented to the coefficient determination algorithm. This strategy creates an averaged set of coefficients for a particular sensor unit that is more robust.

## 3.4  Joint Correction

In their work, Martin et al. (2017) present a comparison between a sequential and joint method to correct the impacts of pressure, temperature, and humidity. The sequential method corrects each variable independently in a predetermined order, and the joint method uses multivariate linear regression to correct all variables at once. Their results indicate that the joint method was only 0.27 ppm (on average) better than the sequential method. This slight difference between the two methods should allow researchers to choose the method that is better suited for their experimental setup. For example, in our setup, the pressure experiments were performed in a different chamber than our temperature and relative humidity experiments. Therefore, the correction coefficients were determined based on different datasets. In this section, we offer an example of a hybrid method where the coefficients for temperature and humidity were determined together, and the pressure coefficients were determined separately. We then demonstrate the joint correction of all three variables on a test case.

Even though the pressure correction method was tested on sensors K30_11, K30_12, K30_21, and K30_22, in this example, we only present the results for Test System 2, with test sensors K30_21 and K30_22 because sensors K30_11 and K30_12 were damaged after the chambered pressure experiments (see Sec. 3.1). The pressure correction coefficients were not determined for the other test sensors used in this study because of the Mesonet pressure system's custom connection requirement (see Sec. 3.1), and the benchtop pressure chamber's size limitation (see Sec. 4.1) and radio frequency shielding requirement (see Sec. 5). Therefore, we test our assumption of the equivalence between the sequential and joint methods in Martin et al. (2017) using the pressure correction coefficients determined in section 4.1 and a new cubic fitting of the joint variation of temperature and relative humidity obtained from the data shown in section 3.2. This hybrid set of coefficients requires the data to be corrected for pressure first and then jointly corrected for temperature and relative humidity.

The coefficients, $R^2$, and RMSE for the pressure correction step used here can be seen in section 4.1. This article's supplement shows the ten coefficients for the joint temperature and relative humidity cubic fitting (Tab. S18). The cubic fitting's $R^2$ for sensors K30_21 and K30_22 were 0.9869 and 0.9855. The dataset for the test of the hybrid method presented changes of 378 Pa, 30.51 °C, and 34.76 %RH. During the same period, the reference gas analyzer presented a change in $CO_2$ of 10.54 ppm. Table 4 and figure 9 show the results for this test. As mentioned above, this test was performed in a hybrid format. The first step, pressure correction, only accounted for an average improvement in the RMSE relative to the reference sensor of 0.64 ppm. The second step, joint correction of temperature and relative humidity, produced an average improvement in RMSE of 26.76 ppm across both sensors. The final RMSEs of 1.73 and 3.15 ppm support our assumption.

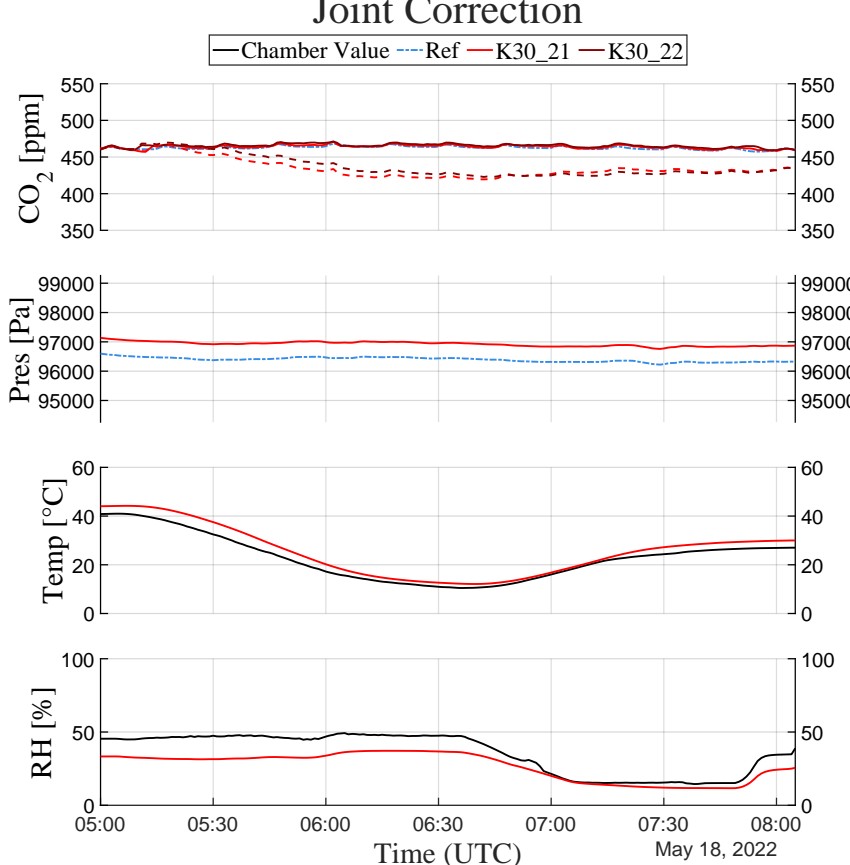

**Figure 9.** Time-series data for the joint correction test for pressure, temperature, and relative humidity. The solid black curve represents the experimental conditions for the test. The red $CO_2$ curves represent the test system and its sensors, the dashed curves represent original data, and the solid curves represent the corrected data. The other solid red curves represent the conditions for the test system and its sensors. The dashed blue curve represents the $CO_2$ and validation conditions for the reference sensor.

**Table 4.** Root mean square error relative to the reference sensor. Step 1 represents the pressure correction, and step 2 represents the joint temperature and relative humidity correction after the pressure correction.

| Sensor | RMSE | | |
| --- | --- | --- | --- |
| | Original | Step 1 | Step 2 |
| K30_21 | 30.66 | 30.07 | 1.73 |
| K30_22 | 29.02 | 28.33 | 3.15 |

## 4 Benchtop Experiments

Many UAS-based atmospheric $CO_2$ applications involve multiple flights per day over multiple days. In these intense operational periods, the exposure to the elements and the dust collected during take-off and landing may greatly impact sensor decay and temporal drift. Given the uncertainties regarding the decay period for each sensor unit, it is recommended to perform system calibration and correction coefficient determination procedures as often as operationally possible. This recommendation is particularly important for systems supporting long-term research. Unfortunately, most of the procedures available in the literature are not practical for many UAS-based field applications. In this section, we evaluate a series of low-cost benchtop setups to characterize and correct the impact of pressure, temperature, and relative humidity on low-cost NDIR $CO_2$ sensors.

### 4.1 Pressure

The experimental setup for this low-cost procedure (illustrated in Fig. 10) consists of a BACO Engineering 5-Gallon Vacuum Chamber Kit, available at multiple retailers for USD189.99, and the LI-840A gas analyzer. In this setup, the gas analyzer provides the reference $CO_2$ values for the experiment's initial state. Then, the chamber is closed and isolated from the external environment. Finally, the chamber is depressurized until the top of the emulated UAS flight is reached. The pressure changes are produced by a microcontroller turning the system's pump "ON" for 2 seconds and then "OFF" for 1.5 minutes. This method uses the ambient $CO_2$ concentration, pressure, temperature, and relative humidity as its initial state. Therefore, it also requires the ambient monitoring strategies detailed throughout section 3.

Besides the ambient monitoring strategies, the benchtop version of the coefficient determination method described in section 3.1 also requires multiple runs of the experiment to achieve a robust result. This is necessary due to the small variations in the test range created by using the ambient conditions as an initial state. If only a small sample is used to determine the coefficients, these small variations in the test range can bias the coefficients. In this section, we demonstrate an example of this technique. We used two "learning" cases to generate data points for the cubic fitting (shown in 3.1.1) and then evaluated the performance of the correction on two "test" cases.

Since each test case is performed with two test sensors, the method was evaluated four different times. The use of only two test sensors (one test system) was determined by the size of the BACO Engineering 5-Gallon Vacuum Chamber Kit. Still, the variations in experimental conditions between all four cases provide insight into the method's repeatability. Nevertheless, it is important to note that the initial pressures in all tests are lower than sea level pressure. This occurs because the experiments were performed in Oklahoma (approximately 360 m above sea level). All cases emulate a UAS-based atmospheric $CO_2$ vertical profile, where there is a dwell period (in this case, 1.5 minutes) to ensure samples from the previous altitude are discarded from the system after a change in altitude. The pressure range tested emulates a flight to the average height of the top of the Atmospheric Boundary Layer in Oklahoma. The results for all four cases can be seen in Fig. 11 and Tab. 5, where the time-series data for the reference, original, and corrected concentrations (for both test sensors) are plotted together for comparison. The results demonstrate how the low-cost coefficient determination method successfully produced errors smaller than 2.5 ppm

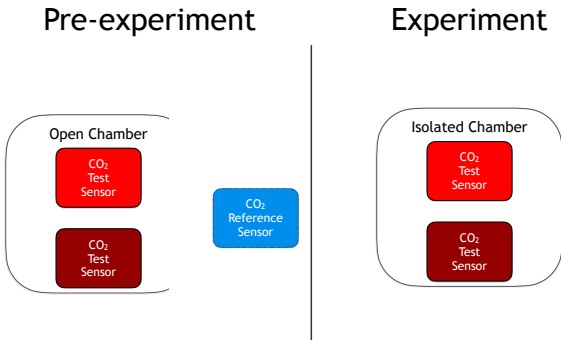

**Figure 10.** Diagram for the benchtop pressure correction experiment. Chamber and sensors stabilize to environment conditions (pre-experiment). Then, the chamber's isolation maintains the initial $CO_2$ values while pressure changes.

in all four cases. These results are even more impressive considering the data represents emulated flights up to 5,200 ft above ground level in Oklahoma or 6,500 ft above sea level, performed in less than 30 minutes.

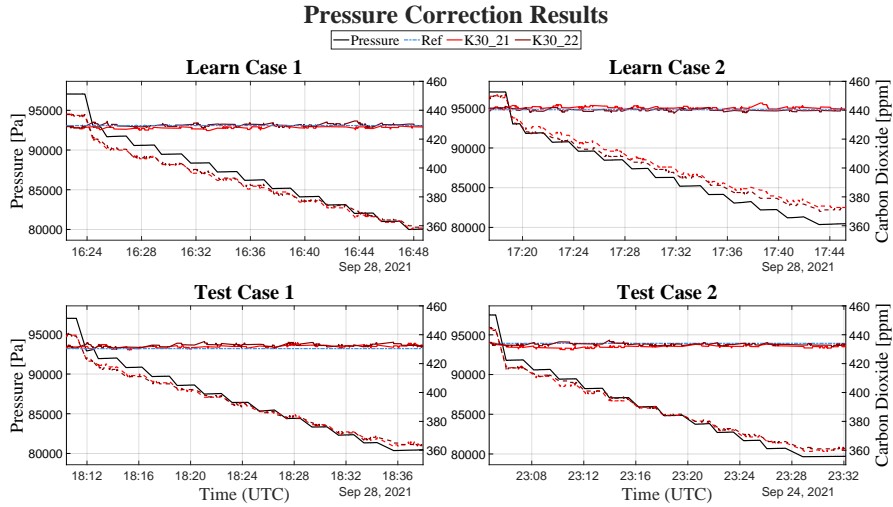

**Figure 11.** Dataset for development and validation of the pressure correction coefficient determination method. The first row data was used to determine the coefficients for each test sensor, and the second row data was used to evaluate the performance of the coefficients. The solid black curve represents the pressure inside the chamber. The red curves represent the $CO_2$ values reported by test sensors (dashed lines represent the original data, and the solid lines represent the corrected data).

**Table 5.** Coefficients from the benchtop pressure correction method.

| Sensor | $k_1$ | $k_2$ | $k_3$ | $k_4$ | $R^2$ | RMSE | | | |
|--------|-------|-------|-------|-------|-------|---------|---------|--------|--------|
| | | | | | | Learn 1 | Learn 2 | Test 1 | Test 2 |
| K30_21 | -3.6254e-12 | -1.5353e-07 | 0.0027 | 22.3675 | 0.9952 | 1.6650 | 1.7060 | 1.6818 | 2.4470 |
| K30_22 | -1.7450e-12 | -6.3144e-08 | 0.0040 | 26.2825 | 0.9992 | 0.9588 | 0.8368 | 2.3899 | 1.0270 |

### 4.1.1 Time Response to Pressure

While analyzing the data for the pressure correction experiment, a delay in $CO_2$ concentration change due to pressure change was noticed. While time response to pressure, temperature, and relative humidity should have its own dedicated study, we elected to add to this article a small experiment to illustrate the time response to pressure due to its impacts being independent of sensor housing design. Another reason to add a small commentary here is to at least create awareness of its potential impact since no mention of such an effect was found in all the literature reviewed for this article. In this experiment, we used the BACO Engineering 5-Gallon Vacuum Chamber Kit to produce examples of impulses, steps, and stairs. These three distinct patterns of pressure variation are shown in Fig. 12.

Analyzing the four cases presented in Fig. 12, we noticed the effects of the time response to pressure had two components. There is a constant delay that causes a time shift (illustrated in case 2) and an exponential delay similar to an e-folding effect. Because the pressure chamber is completely isolated from the external environment, once closed, we can conclude that the time response to pressure is independent of the effects of the sensor's time response to actual changes in $CO_2$. This time response to pressure can introduce errors when performing pressure corrections on low-cost NDIR $CO_2$ sensors because fitting algorithms would map multiple distinct $CO_2$ values to a single pressure value. There are two strategies to mitigate this problem.

The first strategy is to discard $CO_2$ samples near pressure changes. This strategy is fairly common when post-processing data from UAS-based gas sampling that uses any sensor housing and controlled airflow. In these cases, removing samples near pressure changes is necessary because the plumbing and housing add a memory to the system. In other words, air samples from one pressure/altitude are transported by the UAS to another pressure/altitude before the samples complete their course through the plumbing and housing. Perhaps this common practice of discarding $CO_2$ samples near pressure changes is why this effect does not appear in the literature.

The second strategy is to correct the time-response induced errors before correcting the pressure-induced errors. Since no mentions of this error were found in the reviewed literature, no correction methods were found either. Therefore, we attempted to correct this error using known techniques for other atmospheric sensors, following the time response modeling from Houston and Keeler (2018) and Miloshevich et al. (2004). We used the steps and stairs (cases 3 and 4) to calculate an averaged constant ($\tau$) for the exponential correction and the peak distances of the impulses (cases 1 and 2) for the averaged shift.

To evaluate the performance of this correction method, we created an artificial signal to represent the ideal response to pressure. This artificial signal represents what the sensor response to pressure should have been without the pressure time-

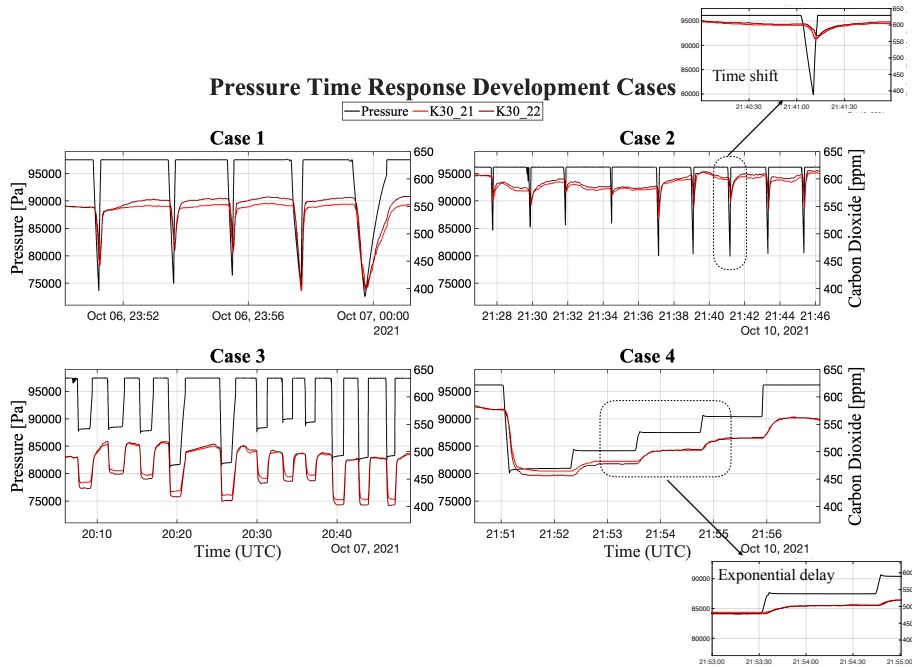

**Figure 12.** Development data for investigation of the pressure time-response. Cases 2 and 4 highlight the time shift and exponential delay. The solid black series represents the pressure inside the chamber for all plots. The two red series represent the $CO_2$ values reported by the test sensors.

response error. In this artificial signal, the pressure-induced error is instantaneously reflected on the sensor output. Such a signal would minimize (or not produce) the mapping of multiple distinct $CO_2$ values to a single pressure value during the curve fitting algorithm for the pressure correction method. Therefore, this artificial signal represents the benchmark for a pressure time-response correction method. The artificial signal representing the ideal response to pressure was created using the timestamps of the pressure changes and the average $CO_2$ concentration for each pressure level. This average $CO_2$ concentration was obtained for each pressure level after all exponential delays. The results of our correction attempt are shown in Fig. 13.

Our proposed correction method improved the mean absolute error (MAE) for both sensor units, when compared to the artificial signal. MAE for Sensor 1 improved from $0.9806$ to $0.6633$ ppm, and Sensor 2 improved from $0.8702$ to $0.5940$ ppm. The improvements are even more expressive when we analyze the maximum absolute error (MxAE). Sensor 1 improved from $MxAE = 12.965$ to $5.3024$ ppm and Sensor 2 improved from $MxAE = 11.533$ to $4.4393$ ppm. The experiment was repeated on another test case with similar results (see supplement S19).

Although the results presented here indicate the feasibility of a repeatable method to correct pressure time-response errors on low-cost NDIR $CO_2$ sensors, we highlight again our intention to only create awareness of this potential source of error. As mentioned above, the time response to pressure, temperature, and relative humidity should have its dedicated study. Despite improving MAE and MxAE, our proposed correction still presented errors. Most notably during the period from 18:26 to

18:32, highlighted on the time series for Sensor 2 (Fig. 13). For those whom this time response is an issue, we recommend repeating these experiments on a better quality chamber, one capable of producing smaller and better-defined pressure changes, or adopting the first mitigation strategy presented in this section.

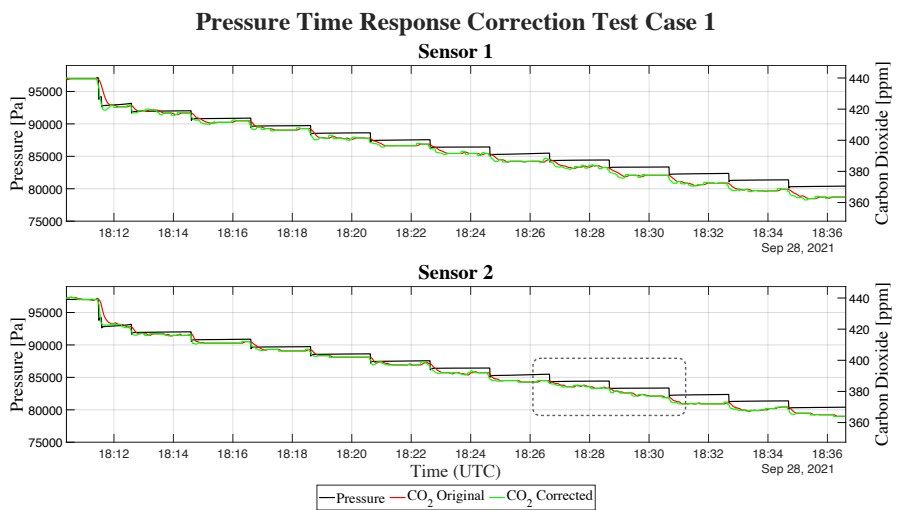

**Figure 13.** Correction Results for Test Case 1 for the sensor's time response to pressure changes. The solid black curve represents the pressure inside the chamber. The red and green curves represent the test sensor's original and corrected $CO_2$ values.

## 4.2    Temperature and Relative Humidity

In this section, we investigate four low-cost benchtop setups to characterize and correct the impact of temperature and relative humidity on low-cost NDIR $CO_2$ sensors. For these experiments, we are considering the combined effects of temperature and relative humidity due to the difficulty of isolating them in a benchtop setting. As mentioned in section 4, the goal is to devise practical methods for field calibrations. In all four experiments, the test sensors were compared to a reference gas analyzer (LI-840A or LI-820), and the thermodynamic sensor package for UAS measurements described in B. H. de Azevedo (2020) was used to monitor the experimental conditions. This thermodynamic sensor package consists of three IMET glass bead thermistors and three IST HYT-271 hygrometers. In the cases where the test sensors used sensor housing, the HYT-271 hygrometer inside the sensor housing was used to compare the experiment's temperature and relative humidity to the values inside the housing. For more information on the test sensor configuration and housings used, refer to section 2.1.

The first benchtop setup tested was a large plastic container with an electric heater and a water spray. Inside the container were the UAS thermodynamic sensor package, a medium mixing fan, and the reference and test sensors. In this setup, the container (open lid) was placed near an open window and two large fans. After the temperature, relative humidity, and $CO_2$ levels were stable, an experiment operator partially closed the lid and activated either the heater or the water spray. Our initial assessment indicates that the large fans were not able to mitigate the impact of the $CO_2$ produced by the proximity of the

operator and the test sensors. To mitigate the impacts of the operator, we also attempted to reduce the experiment's duration
and intensify the test variable stimulus, similar to the pressure impulse shown in Fig. 12. In this short duration format, the UAS
thermodynamic sensor package registered the short stimulus (for temperature and relative humidity). However, the same was
not confirmed by the HYT-271 sensors inside the K30 housings, and the $CO_2$ test sensors did not produce a coherent response.
Even though the UAS thermodynamic sensor package and the pump intake for the reference and test sensors were placed a few
centimeters apart, the approximately 68 L (18 gal) of the container may have been too large for the short stimulus to produce a
relevant change inside the K30 sensor housings. An example of the results produced by this setup can be seen in this article's
supplement (Fig. S20).

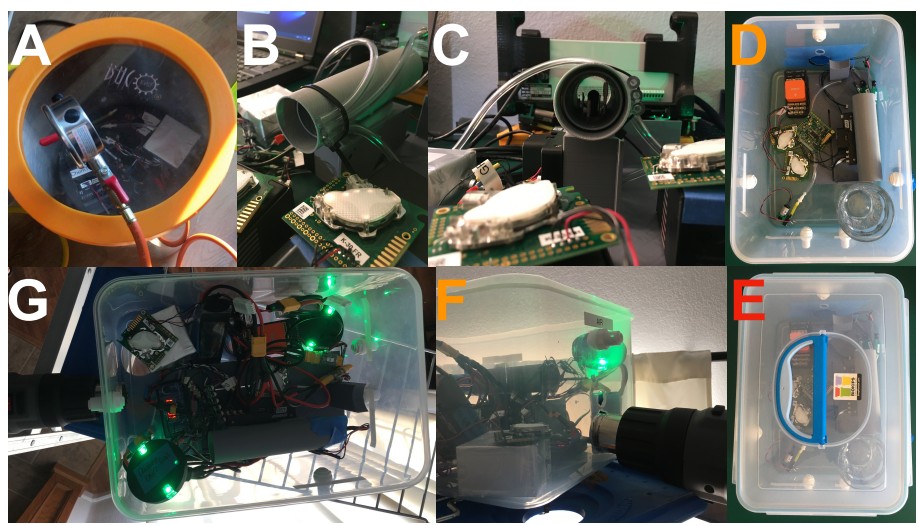

**Figure 14.** Benchtop experiments to characterize and mitigate the impacts of pressure, temperature, and relative humidity on low-cost NDIR
sensors. Panel A shows the Baco Engineering pressure chamber. The remaining panels represent the second (B and C), third (D and E), and
fourth (F and G) benchtop temperature and relative humidity experimental setups.

In the second benchtop setup tested, we removed the large plastic container and allowed the room with the reference and
test sensors to stabilize to constant levels of pressure, temperature, relative humidity, and $CO_2$. With a long extension cord,
we allowed the electric heater to simultaneously warm-up in a separate room. Then we moved the electric heater to the test
room and placed it immediately in front of the reference and test sensors. In this experiment (panels B and C in Fig. 14), we
colocated all six test sensors in all three test configurations (see Sec. 2.1). An example of the results produced by this setup can
be seen in this article's supplement (Fig. S21). Again, the HYT-271 sensors inside the housed test systems did not indicate the
same temperature and relative humidity changes as the UAS thermodynamic sensor package. Nonetheless, the behavior of the
425 housed test sensors was similar to the unhoused sensors (except for the sensor noise caused by temperature on the unhoused
test sensor K30_13).

The third benchtop setup tested used a small plastic container (approx. 12 L). Inside the container were the UAS thermody-
namic sensor package, the test sensors, and a small mixing fan. Due to the container size, we could only use four test sensors

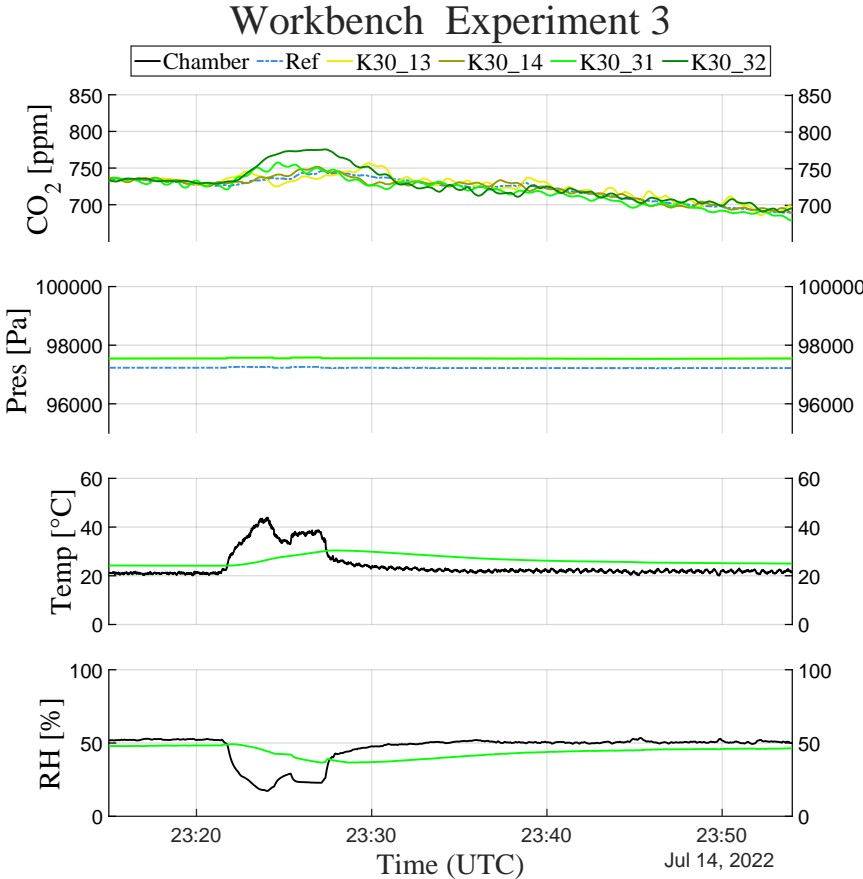

**Figure 15.** Results for the third benchtop setup tested. The top panel shows the reported $CO_2$ values for all sensors, the following panels show the experimental conditions (pressure, temperature, and relative humidity) for the reference and test systems.

in this setup, and the reference sensor had to be placed outside the plastic container. To maintain reference colocation, we
used a plumbing port to allow the reference sensor to sample air from inside the container (panels F and G in Fig. 14). At the
beginning of the experiment, the container's lid was open, and all the sensors were allowed to stabilize to the room levels of
pressure, temperature, and relative humidity. Then, the container lid was closed, and an electric heater was turned on. After
five minutes, the heater was turned off, and the lid was opened. All four test sensors (both housed and unhoused) responded to
the increase in temperature. However, the reference also presented a slight increase in $CO_2$ for the same period. An example of
the results produced by this setup can be seen in Fig. 15.

The fourth and final benchtop setup tested used the same arrangement from the previous setup, with the heater being replaced
with a glass for boiling water (panels D and E in Fig. 14). Again, the container's lid was open, and all the sensors were allowed
to stabilize to the room levels of pressure, temperature, and relative humidity. Then, the boiling water was added to the glass,

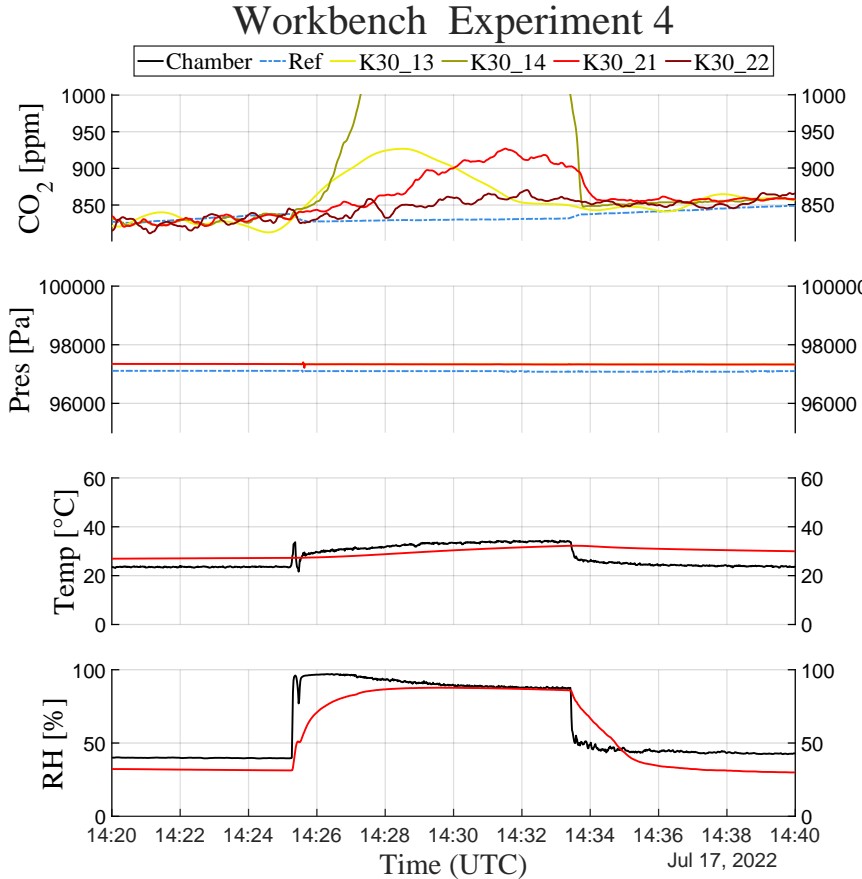

**Figure 16.** Results for the fourth benchtop setup tested. The top panel shows the reported $CO_2$ values for all sensors. The following panels show the experimental conditions (pressure, temperature, and relative humidity) for the reference and test systems.

and the container's lid was closed. After eight minutes, the lid was opened. This setup's results can be seen in Fig. 16. All four test sensors (both housed and unhoused) responded to the increase in temperature and relative humidity, while the reference sensor did not indicate a change in $CO_2$.

The results from experiment setups three and four are encouraging. However, repeated executions of both showed some variation on how much and how fast the test sensors reflected the chamber conditions. Also, potential contamination from the experiment's operator opening and closing the container lid make it less consistent. Therefore, we limit the analyses of these results to indicate only that a low-cost, low-complexity method can be developed for field calibrations of low-cost NDIR $CO_2$ sensors. A broader discussion of the general considerations for testing low-cost NDIR $CO_2$ sensors is presented in the following section.

# 5 Discussion

In this article, we presented many different chambered and benchtop low-complexity experiments in hopes of exploring the behavior of low-cost NDIR $CO_2$ sensors under the strong rates of changes in pressure, temperature, and relative humidity commonly associated with UAS-based measurements. In our total time working with these sensors, we noticed some characteristics worth highlighting in this section. The first characteristic worth discussing is the sensor's construction. The Senseair K30 and all other low-cost NDIR $CO_2$ sensors commercially available were not designed for UAS-based applications. Many of them were designed for indoor, medical, and industrial applications. This design assumes a natural air exchange with the environment over a long period (minutes to hours). Therefore, their optical chambers offer little control over the air entering and exiting the chamber. This long and uncertain permeation period directly impacts the spatiotemporal resolution of UAS-based measurements with these sensors. To mitigate this permeation issue, some researchers and manufacturers adopt fans or custom airflow solutions with diaphragm pumps (e.g., CO2Meter's pump cap for the K30).

Even though we did not study the impacts of airflow control solutions on the sensors and their responses to pressure, temperature, and relative humidity, the difference in results between the chambered and benchtop experiments indicates a possible impact. Within the scope of our study, the main impact noticed was the failure to generate impulse-like responses on the test sensors with some setups of the benchtop experiments. For example, by comparing the temperature experiments for the chambered and benchtop setups one, two, and three, we can identify cases where the airflow control solution may have isolated the $CO_2$ test sensors from the external temperature stimulus. During the chambered and benchtop (setup three) experiments, the test sensors, sensor housing, and pump were exposed to the temperature stimulus. The $CO_2$ test sensors presented the expected temperature-induced errors in these two cases. However, during the benchtop experiments for setups one and two, where only the airflow control solution's intake was exposed to the temperature stimulus, the $CO_2$ test sensors did not present the expected temperature-induced errors. The test conditions for these experiments were validated by comparison with unhoused sensors and comparison of the temperature sensors inside and outside the sensor housings (more details in sections 3.2 through 4.2). Similar errors in temperature and relative humidity probes associated with filters and airflow have been reported in the literature (e.g., Richardson et al., 1998). This potential impact of the airflow control solution limited our analyses of the experiments to the cases where temperature and relative humidity changes occurred in intervals between 5 and 10 minutes, and the entire test system was immersed in the test conditions. We do not consider this limitation a problem for UAS-based applications using these low-cost NDIR $CO_2$ sensors because of the slower flight speeds already required to match the system's $CO_2$ time-response and allow oversampling techniques necessary to improve their accuracy.

Another interesting effect noticed during the study was the impact of radio frequency on the sensor's reported values. To keep the chamber and benchtop arrangements as simple as possible, we avoided using complementary computers to log the data from the sensors. Instead, we used the sensors in their flight package format, completely independent from external resources. This choice reduced the complexities of running power and data cables to the chambers and benchtop setups. This choice also better reflected how the sensors were expected to behave and provide data during flights. The GPS and flight telemetry modules used to time-position stamp the $CO_2$ data and transmit it, in near real-time, to the ground-station computer produce electromagnetic

interference (EMI) on the K30 sensors. This EMI generates oscillations of the reported $CO_2$ values in the order of hundreds of ppm. Even though this effect can be mitigated with proper grounding and by adding EMI tape to the sensor's airflow control housing, this effect impacted our ability to colocate sensors in some experimental setups. This was particularly impactful for the unhoused control sensors inside the BACO Engineering benchtop pressure chamber and the small plastic container for the benchtop temperature and relative humidity experiments. A video of this EMI effect is provided in this article's video supplement section.

Finally, it is important to highlight that the low-complexity methods shown in this study are very sensitive to changes in background $CO_2$. This sensitivity comes from using reference gas analyzers as the true $CO_2$ values. This use of an external reference implies the constant comparison of the values reported by the test sensors and the reference gas analyzer. However, the sensitivity to $CO_2$ changes of these two different categories of sensors differs considerably. Therefore, obfuscating the impacts of the environmental test variable. Thus, any repetition of the methods shown in this article requires an environment with small to no changes in $CO_2$ conditions. As mentioned previously, these conditions can be achieved by isolating the environment and reducing the experiment's duration.

## 6 Conclusions

In this article, we reviewed the main concerns regarding the use of commercial low-cost NDIR sensors for atmospheric $CO_2$ measurements found in the literature. We then built upon experimental results in the literature by investigating the isolated impact of pressure, temperature, and relative humidity under emulated UAS flight conditions. We presented a new dataset with stronger rates of change than previously found in the literature and a low-complexity method using a reference gas analyzer. This low-complexity method successfully produced error correction algorithms for each studied variable within a few ppm of the more expensive reference sensors. Even though we were not able to successfully demonstrate the low-cost benchtop setups to characterize and mitigate the impact of these variables on the same sensors, this article provides important insights for the future development of these setups. We believe these low-complexity procedures are a way to lower the entry barriers to this research field while improving the accuracy of UAS-based $CO_2$ measurements through frequent recalibration.

Another important contribution of this study is to raise awareness around other issues associated with UAS-specific deployment of low-cost NDIR $CO_2$ sensors, in particular the potential impacts of these sensors' time response to pressure, temperature, and relative humidity. We strongly believe these issues should have their own dedicated study. We also recommend the investigation of the impact of custom airflow control solutions on the propagation of temperature through the measurement system.

We also note that the statements from Gaynullin et al. (2016) regarding the need for a distinct set of correction coefficients for each sensor were verified in this study. This requirement is also supported by Martin et al. (2017), who found that a generalized set of coefficients could make the accuracy worse than when uncorrected.

In our concluding remarks, we emphasize the importance of sensor placement, sensor housing design, and airflow control for successful UAS-based measurements. Furthermore, the characterization of UAS-based systems should consider the potential

contamination introduced by the aircraft and its mode of operation (e.g., vertical profile, transects, hover, and other flight patterns). Finally, any system used to support long-term research or forecast operations should also account for temporal drift and sensor decay.

*Video supplement.* https://doi.org/10.5446/58195

*Author contributions.* This study was conceptualized by GBHA and DS. The methodology was designed by GBHA and DS. The Mesonet
experiments were designed by GBHA and CAF. The Benchtop experiments were designed by GBHA, DS, and BD. The formal analysis and post-processing of the data were carried out by GBHA under the supervision of DS. Supporting software was developed by GBHA, DS, and BD. The original draft was written by GBHA and reviewed and edited by GBHA, DS, BD, and CAF.

*Competing interests.* The authors declare that they have no conflicts of interest.

*Acknowledgements.* The authors would like to recognize the efforts of David L. Grimsley, the manager of the Oklahoma Mesonet Calibration
Laboratory, for his technical support during the Mesonet chamber experiments. The authors would like thank Dr.Jacob from the Unmanned Systems Research Institute at the Oklahoma State University, for providing the LI-COR LI-820 reference gas analyzer used in the Mesonet chamber experiments. The authors also thank RMSA for the thorough grammatical revisions. This study was supported in part by the Vice President for Research and Partnerships (VPRP) of the University of Oklahoma OU. Publication was supported by the University of Oklahoma Libraries.

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
