# Peer review of "Low-Complexity Methods to Mitigate the Impact of Environmental Variables on Low-Cost UAS-based Atmospheric Carbon Dioxide Measurements"

_Atmospheric Measurement Techniques, 2022_

## Author Comment (AC1)

This document contains item-by-item responses to the reviewer's comments. The reviewer's comments are in black, non-italicized, regular fonts. Author responses are in *blue, italicized* fonts. Changes to the manuscript will be provided during revised submission, per instructions of the Copernicus editorial team (i.e., revised manuscript and diff file). Nonetheless, examples of the corrections and a detailed view of each experiment are provided in the attached discussion supplement.

General comments:

We thank Reviewer #1 for his very thorough analysis of the manuscript. His commentary was encouraging and helpful in improving the final results. Following his (and reviewer's #2) suggestions, we repeated the Mesonet temperature and relative humidity experiments under more controlled conditions, added more test sensors, and adjusted the result presentation. Therefore, the final comment below is written in light of the following changes:

- The Li-820 sensor originally named $CO_2$ Independent Sensor Outside is renamed Reference Sensor Out (Ref_OUT).
- The Li-840A sensor originally named $CO_2$ Independent Sensor Inside is renamed Reference Sensor In (Ref_IN).
- The Senseair K30 sensors originally named $CO_2$ Test Sensors 1 and 2 are renamed K30_##, where the first digit refers to its attached system (i.e. logger, temperature, and relative humidity sensors) and the second digit is its identification. This way the first Senseair K30 sensor of test system 2 is named K30_21.
- The Mesonet temperature and relative humidity experiments now have a "run" identification (i.e., "Mesunet Run 1 - Temperature", "Mesunet Run 2 - Temperature", "Mesunet Run 1 - Relative humidity", etc), where Run 1 is the data originally presented in the manuscript and Runs 2 and 3 are the repetition runs.

A "Discussion Supplement" (appended to the end of this reply) shows a summary of experiments and sensors used following the model in Arzoumanian (2019; doi:10.5194/amt-12-2665-2019), supporting material to explain the low $CO_2$ values seen in the Mesonet Experiment (Run 1). The results of the correction application on the Mesonet T/RH Experiment (Run 1) dataset, followed by the results for the Mesonet T/RH Experiment (Run 2 and 3), and the adjusted Bench temperature and relative humidity Experiments. The document ends with the supporting plot for the Pressure time-response correction ("ideal signal") and a brief discussion about the reported temperature from the Li-820 reference sensor.

Please note that even though the results for the correction application on the Mesonet T/RH Experiment (Run 1) dataset are presented in the attached "Discussion Supplement", they serve only as a comparison of the correction method. Following the reviewer's suggestion, the Mesonet T/RH Experiment (Run 1) dataset will be discarded. Only Mesonet T/RH Experiment (Run 2 and 3) will be analyzed in the revised manuscript.

—-----------------——-------------------------——------------------------——----------------------

*Reply for* Anonymous Referee #1

General comments:

- It is difficult to review the scientific validity of the author's comments regarding the different sensors' dependences on environmental parameters, as the analyses are mostly of a qualitative nature. Increasing the level of quantification would also enable a comparison of their results with other analyses and would be of chief interest for readers of this paper, as they will potentially be looking to apply these corrections to their sensors.
  - Some examples (not complete):
    - L 148 "This hypothesis is supported by a stronger correlation between the test and independent sensors."
    - L 156/157 "In this experiment, the absence of humidity dependence is evident." L 174/175 "This experiment showed an extreme dependence between the $CO_2$ concentration values [...] and pressure."
  - For reference on possible metrics and desired level of quantification and rigor, please consult publications in this journal like Arzoumanian (2019; doi:10.5194/amt-12-2665-2019)
- For all fits in this study, only r-values are provided and slope as well as intercept including respective errors are missing. Additionally, the factors found for the corrections (incl. time response correction) are not presented for any of the analyses, so neither the absolute dependence on environmental variables (or time response) nor the difference in dependence between the two sensors can be evaluated or compared to the scientific literature.

  *We agree with the reviewer and have added plots and tables detailing each experiment's environmental conditions (pressure, temperature, relative humidity) to the revised manuscript's supplement. In addition, we have also added plots and tables with the slope, Y-intercept, $R^2$, and RMSE (to the linear fit) for each test sensor relative to each environmental variable and reference $CO_2$ sensor, for all experiments. The discussion supplement (attached) shows examples of these plots and tables.*

- Since not all environmental parameters are controlled in Benchtop experiments, showing the other environmental factors in the time-series and analysing them is important to ensure the isolation of relevant parameters which is a central goal of this paper.

  *We agree with the reviewer. However, showing all conditions and results in one plot can obfuscate the results and make the plot harder to interpret. To address this need we have standardized the presentation of the experiments, showing all 3 environmental variables for all experiments in separate plots and tables in the revised manuscript's supplement. The discussion supplement (attached) shows examples of these plots and tables.*

*Note: Even though the environmental parameters could not be controlled for the bench experiments, their short duration limits impactful changes in pressure and temperature. Nonetheless, changes in relative humidity are seen during the bench temperature experiment due to the nature of capacitive hygrometers and their dependence on temperature to calculate RH. To help readers evaluate the impact of humidity, the $H_2O$ (ppt or mmol/mol) of the Licor 840a is also shown in the revised manuscript's supplement.*

- Some of the experiments were conducted months apart. Since NDIR sensors' optical properties (and thus dependences on environmental parameters) are known to change with time, I recommend against putting these experiments simply next to each other without some further justification or analysis.

*We agree with the reviewer. To facilitate the reader's understanding of the results we have added a table, similar to E. Arzoumanian et al (2019), summarizing all experiments and their respective sensors. We have also changed the nomenclature of our test sensors to make this more evident in plot labels and result tables.*

Scientific comments

- L 60 "We also isolated the effects of pressure, temperature, and relative humidity on an NDIR sensor and analyzed their impact separately."
  - Did you consider also performing a linearity analysis?

*Upon suggestion from the reviewer, we have added the slope, Y-intercept, and $R^2$ of the first order least square fit for each test sensor against each test variable and the reference sensor. In addition, we have added the RMSE for each test sensor to their fit lines. Considering the importance of isolating the other variables, for each experiment, these same parameters were also calculated for the other environment variables that are not being tested. This way we can evaluate the quality of the experiment.*

- L 79 "[...] creating appropriate conditions to simulate UAS flights."
  - Is this the case for a pressure range from 1050 to 600 hPa (~4km height) as used in the Mesonet Pressure Dependence Experiment?

*Yes. There are many research teams that perform UAS-based CO2 measurements up to 2 km above ground level (AGL). In our particular case, we perform these 2 km AGL profiles in Norman, OK (~400 m above sea level) and near Boulder, CO (~1,655 m ASL). The higher take-off altitude near Boulder makes our sensors experience pressures equivalent to ~3,700 m ASL. Amongst teams with similar flight conditions, we can highlight the researchers at the National Center for Atmospheric Research (U.S.), Finnish Meteorological Institute, Meteo France, and Cyprus Institute.*

- L 87 "[...] this article considered only the CO2 concentration values reported by each sensor unit."

- How were the Senseair K30-FR units configured? Which kind of onboard compensations do they already have and which ones were active?

*Different from other more robust sensors from Senseair, the K30 does not offer configuration options to the average user. According to the manufacturers, the sensors are factory calibrated and "ready-to-use" out of the box. The product documentation does not offer any information on any onboard compensation of pressure, temperature, and humidity. The only compensation mentioned is their ABC algorithm that compensates drift:*

*''The default sensor OEM unit is maintenance-free in normal environments thanks to the built-in self-correcting ABC algorithm (automatic baseline correction). This algorithm constantly keeps track of the lowest reading of the sensor over an interval of 7.5 days, and slowly corrects for any long term drift detected as compared to the expected fresh air value of 400 ppm $CO_2$."*

*Through their development kit (or custom solution) users can change the factory calibration for "ZERO" (at 0 ppm) and "BACKGROUND" (at 400 ppm), per documentation. This allows users to do zero and span calibrations.*

*The K30 sensor does not offer any onboard pressure, temperature, or humidity compensation based on an auxiliary measurement, at least not one that can be turned ON or OFF. However, through personal correspondence and conference calls with Bakhram Gaynullin (SenseAir Engineer and co-author of Arzoumanian et. al 2019) and Carl Bengtsson (CTO of Senseair USA and Engineer on the K30 prototype), it was found that the sensor uses a proprietary algorithm levering the longer light path and factory calibration to reduce errors associated with the proximity of Water Vapor and Carbon Dioxide in wave-length. Additionally, it was found that the temperature of the sensor's light source is monitored during measurements. Corrections based on this measurement are unknown to the public.*

*None of the above-mentioned characteristics are available as configuration options for the average user. Therefore, the sensors were treated as a "black box" under factory settings and calibrations.*

- Li-840A temperature independence
  - Please check your assumptions here, as the manual provides a total drift of the Licor of 0.4 ppm/°C. For the temperature range of 10 to 40C, this would equate to a drift of 12ppm, for 20 to 40C, a drift of 8ppm.

*The Li-840A and Li-820 have internal heaters that elevate their sampling chambers to a temperature of ~51 °C (reported in the discussion supplement submitted with this response), which is above the temperatures tested in our experiments (max:~40 °C). This characteristic eliminates their dependence within the range tested. These assumptions are validated by the new plots and tables for experimental conditions (e.g., discussion supplement figure 14 and table 13) and the results from the mesonet runs 2 and 3 (e.g. discussion supplement tables 14 and 20, and figures 14, 15, 20, and 21). In addition, the discussion supplement tables 15 and 21 show the slope, y-intercept, $R^2$, and RMSE for the Li-840A against pressure, temperature, relative humidity, and Ref_OUT measurements*

*during mesonet temperature experiments 2 and 3. Comparing the $R^2$ estimates for temperature and Ref_OUT (a.k.a. the Li-820), we can conclude that temperature measurements are approximately equal or worse at predicting the behavior of the Li-840A.*

*Besides our analysis, we also forwarded this question to Li-cor's technical support team. The answer received was the following:*

*"Thank you for your email. If the heater is turned on, you should not see any significant drift in the response over the entire specified range of the LI-840A."*

- Reference sensor deployment
    - Why did you choose to setup the 840A inside the chamber and the 820 outside? Was this an arbitrary choice?

*The Li-840A is a $CO_2$ and $H_2O$ gas analyzer and it is considered to be independent of temperature and humidity changes in the time-scale of hours, within the temperature and RH ranges tested. The Li-820 is only a $CO_2$ analyzer and does not have any humidity measurement for active compensation. Both the Li-840A and the Li-820 have active pressure compensation. The Mesonet Calibration Laboratory is an air-conditioned room that does not experience high levels of humidity. For this reason, the Li-840a was used as a reference sensor inside the test chamber where humidity was expected to vary and the Li-820 was used as a reference outside the test chamber to characterize $CO_2$ changes in the laboratory. This reasoning was added to the experiment's description in the revised manuscript.*

- Mesonet Temperature Dependence Experiment
    - Why is the data range for CO2 160 to 320 ppm? These measurements don't seem to be realistic.

*This considerable reduction in $CO_2$ concentration is caused by the Thunder Scientific 2500 chamber, used in the experiments. Figure 1 in the discussion supplement shows the raw ppm values for the Li-840A (Ref_IN), the Li-820 (Ref_OUT), and four K30 sensors, before, during, and after the second run of the mesonet temp/RH experiments. In this figure, it is possible to see the Ref_IN and all four K30 sensors drop their reported values from ~500 ppm to ~250 ppm after the start of the experiment, and return to ~450 ppm after the experiment ends. In the same figure, we can see the outside reference is not affected. The data in figure 2 and table 1, show that none of the pressure sensors varied greatly (all sensors show a standard deviation of approximately 106 Pa). The data also shows this effect happens before large temperature and humidity changes and is also seen on the Ref_IN control sensor (which is independent of temperature and humidity changes in the test ranges).*

*Looking at the documentation of the Thunder Scientific 2500 chamber, there is mention of the use of Nitrogen to control pressure, temperature, and humidity. However, the chamber at the Mesonet Calibration Laboratory does not use this feature. In their configuration, the chamber uses water, a series of compressors, and pre-chambers to generate standard test conditions. More information about the inner workings of this chamber can be found at*

*https://www.thunderscientific.com/humidity_equipment/model_2500.html*. *After our thorough review of the documentation, it was not clear the exact source of this effect. Nevertheless, it is apparent when analyzing the data that this effect has a near-constant behavior throughout the experiments. Therefore, an offset correction can be applied without loss of generality.*

- ○ If you consider the independent sensors to be actually independent from environmental factors, why don't you use those to correct for external disturbances? Analyzing the difference between the test sensors and the independent sensor for temperature dependencies would, then, yield more robust results.

*We agree with the reviewer. We can use the reference sensors to correct the data. In fact, that is what we did in the submitted manuscript. However, we did not do a good job explaining our choice to bring all data to the level of the Ref_IN sensor (Li-840A). This decision was based on the robustness of the Li-840A when compared to the Li-820 (Ref_OUT), and our wish to analyze the data at the relative reference point of the chamber environment. In hindsight, we understand how the unrealistic data presented with a lack of an explanation would confuse readers.*

*Figure 3 of the discussion supplement illustrates the impact of these two different correction strategies. In the left panel, the result of correcting the data to the Ref_IN sensor, and in the right panel the result of correcting the data to the Ref_OUT sensor.*

*To address this issue we used the 60 minutes prior to the experiment (e.g., from 02:00 to 03:00 in figures 1 and 2) where the conditions are stable, to find an average offset for each sensor to the Ref_OUT sensor. This offset was then applied to the entire time series. An explanation of this strategy was also added in the revised manuscript.*

*We have added plots and tables in the discussion supplement showing intercomparisons between the reference and test sensors, before and after each mesonet experiment. These comparisons support our strategy to use the Ref_OUT sensor to correct the data from the mesonet experiments. In these plots, the $CO_2$ concentration measured by the Atmospheric Radiation Measurement - Southern Great Plains (ARM SGP) reference tower was added to show these are reasonable atmospheric values for Oklahoma.* **We understand the distance between Norman and the ARM-SGP tower does not allow for a direct comparison of the data.** *Nevertheless, it is presented here to demonstrate to the reviewers that the two reference sensors were measuring values within reasonable expectations. A version of these additional plots and tables with basic statistical metrics (without the ARM-SGP data), was added to the revised manuscript supplement.*

- ○ As you performed the pressure experiment already 9 days before the temperature experiments, the high pressure dependence was already known. Did you perform any pressure measurements during the temperature experiment to disentangle the influence of the ambient pressure changes

from the temperature changes? Or did you assess the possible effect the pressure dependence might have on the measurements?

*We did measure pressure during the temperature and humidity experiments. However, we did not deem the data from the mesonet pressure experiment good enough to determine correction coefficients. The main reason for this decision is the large $CO_2$ fluctuation in the room during the pressure experiment.  Therefore, the coefficients were only determined much later during the bench experiments. Nonetheless, within our decision to repeat the mesonet temperature and humidity experiments we were able to achieve conditions where the pressure measurements showed less than 140 Pa in standard deviation (across all pressure sensors, across all temp and RH experiments). Considering the small pressure deviation, we chose not to apply a pressure correction as it could introduce uncertainty in our analysis of the impact of temperature and humidity. To mitigate our choice to not correct, we added the slope, Y-intercept, $R^2$, and RMSE to pressure for all sensors, for all experiments on the revised manuscript.*

- ○ Did you measure the temperature at the external independent sensor? Did you preclude any temperature interference here?

*The only temperature measured outside the test chamber was the temperature of the external reference's optical chamber, which stayed around 50 °C. As shown in the experimental conditions plot and table for each experiment. At this point, we believe the external interference was an actual increase of $CO_2$ in the lab due to the presence of mesonet technicians and building security personnel. The repeated experiments were performed overnight to eliminate this interference. The results of this strategy can be noticed in the experimental conditions table for each run (e.g., table 11 of the discussion supplement shows the second temperature experiment had standard deviations of < 30 Pa, across all sensors, 1.23 %RH inside the chamber, and 3.69 $CO_2$ ppm outside the chamber).*

- ○ Quantification:
    - ■ As one of the major sources of temperature dependence in NDIR spectrometers is a change in optics properties, did you consider time-lags in increasing sensor reading?

*The effects of variation in $CO_2$ in the laboratory during the first mesonet experiments were so large (e.g., std 45.14 ppm during the first temperature experiment) that it became too hard to separate the impact of the environmental variables on the test sensors. Based on the reviewer's comments the mesonet experiments were repeated. However, the Thunder Scientific chamber is limited to producing temperature and relative humidity changes in the order of tens of minutes. Since we reduce the dwell period of each step (following recommendations from reviewer #2), we believe that our data set does not support this type of study.*

*Nonetheless, our data can support the importance of a future study to evaluate the potential impact of this NDIR temperature time-lag effect in UAS-based measurements. Our results show that we were not able to produce a distinguishable temperature response within 3 minutes (during the bench temperature experiments). However, the mesonet experiments*

*did show a temperature-induced reported concentration increase within 15 minutes or less. Considering a typical UAS vertical atmospheric profile with a 3 m/s ascent rate, a continuous ascent to 1500 m takes about 8 minutes. In gas sampling flights due to slower sensor responses and pressure issues, the same type of vertical atmospheric profile includes horizontal transects at predetermined altitudes and can take about 27 minutes. Therefore, such a study would certainly be of interest to the community.*

- For which time periods did you calculate the correlation coefficients between sensor reading and temperature? If these were the actual transitions, this might be problematic because of time-lag, non-linearities or material effects.

*The correlation coefficients were calculated for the entire experiment and later only for the actual transition intervals (both shown in the original manuscript). The transition interval in those experiments was approximately 25 minutes. Therefore, using this transition interval would only be a problem if the potential time-lag issues occurred on a longer time scale. If this was the case, these effects might not even manifest in the time scales of typical UAS flights (see flight duration explanation above). Thus, we believe using this relatively slow 25-minute transition interval was not a problem. In fact, we adopted it to reduce the impact of the external changes in $CO_2$ concentration and to investigate the impact of temperature. Nonetheless, this problem was mitigated by repeating the mesonet experiments under more controlled conditions. All $R^2$ values shown in the discussion supplement are calculated for the entire time series.*

- After identifying an external interference in your measurements, it seems to me like you should not be able to draw conclusions about a sensor temperature dependence without EITHER: discarding the data and repeating the experiment without this source of interference (even Gaynullin, 2016 says that an "[a]bsolute elimination of contaminating leakages from ambient air" must be provided "to provide a reliable test environment") OR: treating the interference by e.g. analysing the difference in readings between the Licor and K30 sensors (under the assumption that the Licor readings are independent of temperature, which has to be proven given its design specifications – cf. above)

*As mentioned above, the experiments were repeated two more times and the new results are presented in the discussion supplement attached.*

- Benchtop Temperature Dependence Experiment
  - CO2 readings seem to decrease at the beginning of the experiment. Did you ensure the experiment was in a steady state with respect to CO2?

*We agree. The $CO_2$ reported by all three sensors presents a decreasing trend before the temperature change occurs. However, in a more quantitative analysis, we can show the reference sensor and the test sensor 1 present a change smaller than 3 ppm during the 90*

*seconds before the temperature change. Although 90 seconds can be considered a small stabilization period for a $CO_2$ sensor, it represents 50% of the total duration of the temperature change. Again, we agree it is not an ideal testing condition, but unfortunately, it is the limitation of this low-cost testbench setup. Nonetheless, we have revised the manuscript to highlight this experimental limitation and have limited the interpretation of the results. We also added to the discussion supplement a repetition of this experiment in slightly more stable conditions (figure 35).*

- ○ L130/131 "Even though there is a slight 10 ppm increase [...], it occurs a full minute after the temperature is brought back near its original state."
    - ■ Again, might this be due to the temperature response time of optics/electronics, a large mixing volume or even some drift in timestamps of the recording PC (especially since this experiment was conducted 18 months after the other ones)? If this were the case, the increase in signal would fit very well with the expected one of the Licor due to temperature drift, which is 8ppm (cf. above).
    - ■ Where was the temperature measured? Was the possibility of a time-lag between temperature measurement and sensor measurement of an air parcel excluded?

*As demonstrated previously the reference sensor (Li-840A) can be considered independent of temperature within the tested ranges. Therefore, we assume the reported 10 ppm increase to be an actual increase in $CO_2$ during the experiment.*

*Regarding the temperature measurements, the diaphragm pump intakes for the $CO_2$ sensors (one for the reference sensor and one for the two test sensors) were placed immediately after three PT-100 bead thermistors (10 Hz sampling, 1 Hz time response, sold and calibrated by InterMet Systems), and three IST HYT-271 capacitive hygrometers (10 Hz sampling, 4-second RH time response, and 5-second Temperature time response). The temperature shown in the original manuscript is an average of the temperature of the 3 thermistors.*

*Regarding any time shifts associated with the data loggers, we can offer the following information:*
*        - Both K30 sensors, the three thermistors, and the three hygrometers share the same logging system.*
*        - The reference $CO_2$ sensor was logged separately.*

*Therefore, if any logging-related issue manifested itself in the data, such an issue could only occur between the test sensors and the reference sensors, and not between the test sensors and the temperature sensors. Although a time lag issue on the data seems an unlikely explanation for the result, an actual temperature time-response behavior on the K30 cannot be ruled out with the available data set. In light of the results from the repeated mesonet experiments and upon further analysis of the bench data, it seems more likely that any temperature impacts on the*

*K30s were obfuscated by the increase in $CO_2$ during the experiment. Therefore, we revised the manuscript and limited the interpretation of the bench experiments to indicate a potential time-scale temperature-impact limitation.*

- ○ Since this experiment does not control RH, it is difficult to use this for a impact analysis isolating the environmental parameters, which is the stated aim of the paper. This would need some further justification leveraging measured RH values during the experiment.

*We agree with the reviewer. We have limited the previous interpretation of this result as a rapid change experiment (emulating more realistic conditions encountered during UAS measurements, as pointed out by reviewer #2) with limited conclusions regarding the time scales of a temperature-induced response.*

*As mentioned previously, we have added the plots and basic statistics (min, max, avg, and std) for all three environmental variables during all experiments, as well as a comparison of the slope, $R^2$, and RMSE between the test sensors and the non-test variables. In this experiment, RH varied between 19 and 42 %RH (see discussion supplement figure 28 and table 27).*

*We are currently working on an analysis with the RH measurements and a comparison to the mesonet data set (a particular section of the time series, between the temperature and relative humidity experiments, where both variables change together). If successful this result may serve as a comparison to other non-isolated studies with the distinction of the more UAS-appropriate time scales.*

- Mesonet Relative Humidity Dependence Experiment
    - ○ The same comment regarding the external interference applies here (cf. Mesonet Temperature Dependence Experiment).
    - ○ The same comment regarding the CO2 data range (120-300 ppm) applies here (cf. Mesonet Temperature Dependence Experiment).

*Answered at the "cf. Mesonet Temperature Dependence Experiment".*

- Benchtop Relative Humidity Dependence Experiment
    - ○ Details of the humid air source are missing – is it guaranteed that temperature stays constant and only RH increases?

*The humidity source was a manual water spray. During this experiment, the minimum temperature was 22.30 and the maximum was 25.33 °C. The average was 23.75 with a 1.02 °C standard deviation.*

- L 188/189 "the correction needs to be based on the variation magnitude from the initial state"

- ○ Can you provide some physical justification for this assertion? This is not discussed for temperature or humidity and to my knowledge, it is not common practice.

*The highlighted assertion refers to the pressure correction method NOT a temperature of relative humidity correction. The assertion serves only to warn the reader about our assumption of the linearity of pressure's impact on the K30 between sea level and the test ranges. We stated it because we used ambient pressure as initial pressure ($P_0$) in the equation provided in Gaynullin et al. (2016) instead of sea-level pressure. As stated, we believe the assumption is supported by the results of the mesonet pressure experiment (correlation coefficient of 0.98 for both sensors in the range from 60000 to 105000 Pa).*

- L 202 "The results demonstrate four instances [...]"
  - ○ Where there further instances where this method was applied unsuccessfully?

*There were only these four tests, with two sensors each for a total of 8 successful applications of the method. The manuscript was adjusted to make the text clearer.*

- ○ Is this a robust method with respect to e.g. time and temperature?

*The method was not tested for either time or temperature corrections.*

- Fig 12: This dependence seems very linear, almost as if it were following the ideal gas law.
  - ○ Again, please add some comments regarding pressure correction being enabled or disabled in the Senseair K30 units.

*As mentioned previously, the K30 does not have an onboard pressure sensor or a configuration for onboard pressure corrections. The linearity found seems to be in accordance with the results presented by Martin et al. (2017) and the results for the HPP3.1 sensors in Arzoumanian et. al. (2019). The contribution of this paper regarding the pressure correction was to demonstrate that coefficients obtained with a low-cost test environment without traceable canisters (bench setup) can produce satisfactory results.*

- Fig 13: There seems to be some kind of change of pressure dependence with time (similar to hysteresis effect)
  - ○ Was this analysed further?
  - ○ Could this be caused by the large pressure range?

*The deviation of the sensors' behavior between the three distinct lines on the left panels of figure 13 (original manuscript) seems to be associated with the changes in $CO_2$ in the room during the experiment. Due to this change mid experiment, the same pressure does not return the same $CO_2$ level because the measured level changed. Due to this change in concentration, the mesonet pressure experiment was only used in the manuscript to demonstrate the magnitude of the pressure impacts on the sensors. All coefficient calculations were based on data from the bench experiments.*

- L 211 "Because the pressure chamber is completely isolated from the external environment [...]":
    - In the experiment before, this chamber was not isolated which is why you had the outside reference sensor. In L 207/208 you write "the pressure correction experiment setup [...] was used again" Is this not the same, non-isolated setup?

*The MESONET pressure chamber is NOT isolated. It uses two Thompson pumps to move air in and out of the chamber (increasing and decreasing pressure). Therefore, figures 12 and 13 (in the original manuscript) show the impact of a 50 ppm $CO_2$ variation in the laboratory on the experiment results. The BENCH pressure chamber (referred to in L211 and 207/208) is considered isolated during the experiment because once the chamber is sealed the air can only be removed from the chamber (only lowering pressure). In this case, if the room $CO_2$ changes during the experiment, the sensors inside the chamber are not affected by it. The pressure and the pressure time-response corrections were both performed with data from experiments using the BENCH pressure chamber.*

- L 222/223
    - Why did you use a different data source for estimating the exponential correction time constant and the time shift? Did you consider correcting the time shift first to then estimate the exponential correction time constant from this corrected data?

*The e-folding correction used [presented in Houston and Keeler (2018) and Miloshevich et al. (2004)] only takes into consideration the difference in $CO_2$ between the current and previous measurements, the time step between the two measurements, and a predetermined coefficient (tau). Therefore, applying the time-shift before or after the e-folding correction does not yield any difference. Cases 3 and 4 were used to determine the coefficient (tau) because they are easier to calculate the time taken to reach 1\*tau (63.2% of the final step change). Cases 1 and 2 were used to calculate the time shift because using the minimum pressure and $CO_2$ for each impulse it is easier to calculate the time shift. This case selection eliminates the need for the additional derivation (or integration) that could potentially introduce errors. Potential errors from using different datasets are mitigated by using 2 sets in each type of data set (i.e., cases 1 and 2 for impulses and cases 3 and 4 for step changes).*

- L 225 "[...] an idealized signal was created [...]"
    - How did you create this idealized signal? The experiments with this data are not shown in this paper. It would be important to at least add them to the supplement for the sake of documentation and repeatability.

*First, it is important to note that the referred idealized signal is an artificial signal with the pressure error but without the pressure time-response error. Therefore, ideal means the ideal impact of pressure (without pressure time-response error). It serves only as an evaluator of the pressure time-response algorithms. It should not be used for any pressure correction algorithms. The ideal signal was generated using the timestamps of the pressure step changes and the $CO_2$ values of the K30 after stabilization, after the step*

*change. A figure with the Pressure, ideal signal, original signal, and the corrected signal is shown in section 9, figure 43 of the attached discussion supplement.*

> ○ Why did you opt for creating an idealized signal rather than applying the pressure correction to the exponential and time shift corrected data? Is the purpose of the exponential- and time shift correction not to improve the pressure correction?

*By using the ideal signal, we were able to evaluate the effectiveness of the pressure time-response algorithm, without other potential sources of errors. Determining the effectiveness of the pressure time-response algorithm is crucial because, in a pressure correction algorithm that does not discard samples near pressure step changes (e.g., in a UAS flight under continuous ascent), the pressure time-response algorithm needs to be applied before the pressure correction algorithm. Otherwise, the pressure correction algorithm creates overcorrections near the pressure step changes (artificially increasing the reported $CO_2$ value).*

- L 231 "Unfortunately, the attempted correction was not as effective on the gradual pressure changes"
  - ○ Where do you see a difference in expected and real outcome? Did you quantify this?

*In our case, the expected was the idealized signal. The differences noted were the sources of the RMSE reported in the manuscript. They can be seen in the step changes for the test sensor 2, after 18:26, 18:28, and 18:30 (in figure 34 of the discussion supplement the expected is the blue line, the real outcome is the gray line, and the correction attempt is the orange line). In these examples, the corrected curve shows smaller errors against the original curve than the idealized curve.*

- L 234 "we recommend repeating these experiments on a better quality chamber"
  - ○ Why do you think the quality of the chamber is deficient? Why would smaller pressure changes help?

*The BACO Engineering 5-Gallon Vacuum Chamber Kit was not designed for this type of application. It requires an approximate 2500 Pa change to seal its lid. This large change does not allow this system to be used to study the impact of pressure changes near the ground (a known important region for studies of the impact of rotary-wing UAS on sensors). The step changes produced in this study were already at the limit of the system's vacuum pump, where the small 0.5-second pump activation period caused the pump to leak oil and probably reduced the lifespan of the system.*

Technical comments

- L 48 None of the sensors available in the market was designed for UAS-based deployment.

○ None of the sensors available **on** the market **were** designed for UAS-based deployment.

*Corrected.*

- Fig 1:
  ○ Naming the sensor "CO2 Independent Sensor" is misleading as one might understand the sensor to be independent of CO2 instead of temperature. I suggest amending the names.

  *We agree with the reviewer, the sensors were renamed reference sensors (Ref_IN and Ref_OUT)*

- Fig 2:
  ○ Line colors:
    ■ Please consider choosing colors which have more contrast. Without zooming in, it is difficult to differentiate the lines.
    ■ Why do test sensor 1 and 2 have different levels of saturation? Visually, it seems like sensor 2 is less important.

  *We agree with the reviewer, the plot colors were updated in the revised manuscript and discussion supplement using more contrasting color palettes (similar to Martin et al. 2017). The colors were tested using this tool: https://projects.susielu.com/viz-palette.*

  ○ Line types:
    ■ It is difficult to see the dashed and dash-dotted lines and doubly so to differentiate the two. Again, maybe rethink the figure layout.

  *We agree with the reviewer, the plot lines were changed to solid and colors adjusted (as shown in the discussion supplement).*

- Figure 3:
  ○ For scatter plots between two CO2 readings, always plot the 1:1 line, as the linear fit will be misinterpreted visually to be this 1:1 correlation. Alternatively, one can also restrict the line of the linear fit to the data and use the same x- as y-limits albeit for a larger span, ensuring the 1:1 line is at a 45 degree angle.

  *We agree with the reviewer and added to the discussion supplement a revised version of the plots.*

- Fig 3, 6, 8, 10:
  ○ Please consider using shared x- as well as shared y-axes. This makes such plots easier to read.

  *We attempted to implement a version of the reviewer's suggestion (available in the discussion supplement). Although it is worth highlighting that the scatter plots are*

*presented as a matrix where rows represent test sensors and columns represent test variables (e.g., Pressure, Temperature, RH, and Reference Sensors). Therefore, x-axes may not be shared between columns (as they may represent different units), and the y-axes may not be shared between rows (although, in practice, they are very similar). Nevertheless, we attempted to implement this suggestion on the plots between two CO2 readings.*

- Fig 16:
  - RMSEs by definition cannot be negative

  *Corrected.*

  - Please add Pressure readings as one aim of this figure is to show the quality of the pressure correction.

  *We agree with the reviewer, the change made in the revised manuscript.*

- L 196 "All cases emulate a typical UAS-based CO2 vertical profile" Did you mean flight profile?

  *Even though UAS-based atmospheric vertical profiles and horizontal transects are well-known flight patterns within the weather UAS community, we agree with the reviewer. The manuscript was adjusted to "UAS-based CO2 vertical sampling flight.*

- L 196/197 "[...] there is a dwell period (in this case, 1.5 minutes) to ensure samples from the previous altitude are discarded from the system after a change in altitude."
  - change altitude to pressure or use "simulated altitude"

  *Corrected to "simulated altitude".*

- L 207 "No mention of such affect was found [...]":
  - No mention of such **an effect** was found [...]

  *Corrected*

- L 233 "For those who"
  - For those, **for whom**

  *Corrected*

- L 257
  - Please consider adding the statements with regards to long-term stability also to section 1.1 or 2.

*We agree with the reviewer, the change made in the revised manuscript. A commentary was added to section 2.*

**Discussion Supplement for "The Impact of Environmental Variables on UAS-based Atmospheric Carbon Dioxide Measurements"**

Gustavo B. H. de Azevedo[1,2], Bill Doyle[2], Christopher A. Fiebrich[3], and David Schvartzman[1,4]

[1]Advanced Radar Research Center (ARRC) at The University of Oklahoma
[2]Center for Autonomous Sensing and Sampling (CASS) at The University of Oklahoma
[3]Oklahoma Mesonet, Oklahoma Climatological Survey at The University of Oklahoma
[4]University of Oklahoma School of Meteorology, Norman, Oklahoma

**Correspondence:** Gustavo B. H. de Azevedo (gust@ou.edu)

*Copyright statement.* Authors 2022

**1   Summary**

Following the reviewers' suggestions, we repeated the Mesonet temperature and relative humidity experiments under more controlled conditions, added more test sensors, and adjusted the result presentation. Therefore, this document is written in light of the following changes:

- The Li-820 sensor originally named CO2 Independent Sensor Outside is renamed Reference Sensor Out (Ref_OUT).

- The Li-840A sensor originally named CO2 Independent Sensor Inside is renamed Reference Sensor In (Ref_IN).

- The Senseair K30 sensors originally named CO2 Test Sensors 1 and 2 are renamed K30_##, where the first digit refers to its attached test system and the second digit is its identification. This way the first Senseair K30 sensor of test system 2 is named K30_21.

    - Test systems (labeled T_#) has its own pressure (two MS5611), temperature (three PT-100 bead thermistors), relative humidity (three IST HYT-271 hygrometer), and Carbon Dioxide (two Senseair K30).

- Environmental conditions during the experiments are labeled by chamber, reference sensor, and the test system they are associated with.

- The Mesonet temperature and relative humidity experiments now have a "run" identification (i.e., "Mesonet Run 1 Temperature", "Mesonet Run 2 Temperature", "Mesonet Run 1 Relative humidity", etc), where Run 1 is the data originally presented in the manuscript and Runs 2 and 3 are the repetition runs.

The following sections provide supporting material to explain the low CO2 values seen in the Mesonet Experiment (Run 1). The results of the correction application on the Mesonet T/RH Experiment (Run 1) dataset, followed by the results for the Mesonet T/RH Experiment (Run 2 and 3), and the adjusted Bench temperature and relative humidity Experiments. The document ends with the supporting plot for the Pressure time-response correction ("ideal signal").

Please note that even though the results for the correction application on the Mesonet T/RH Experiment (Run 1) dataset are presented in the attached "Discussion Supplement", they serve only as a comparison of the correction method. Following the reviewer's suggestion, the Mesonet T/RH Experiment (Run 1) dataset will be discarded. Only Mesonet T/RH Experiment (Run 2 and 3) will be analyzed in the revised manuscript.

**1.1 Experiments summary**

The following tables cross lists all the experiments performed and their sensors. The sensor intercomparison experiments are not listed on the table. They were preformed before the first run of the Mesonet experiments, before the second run of the Mesonet experiments, in between the second and third runs, and after the third run of the Mesonet experiments.

| Location | Name | Reference Sensors | Test Sensor |
|---|---|---|---|
| Mesonet | Run 1 Pressure | Ref_IN | K30_11, K30_12 |
| | Run 1 Temperature | Ref_IN, Ref_OUT | K30_11, K30_12 |
| | Run 1 Relative Humidity | Ref_IN, Ref_OUT | K30_11, K30_12 |
| | Run 2 Temperature | Ref_IN, Ref_OUT | K30_21, K30_22, K30_31, K30_32 |
| | Run 2 Relative Humidity | Ref_IN, Ref_OUT | K30_21, K30_22, K30_31, K30_32 |
| | Run 3 Temperature | Ref_IN, Ref_OUT | K30_21, K30_22, K30_31, K30_32, K30_13, K30_14 |
| | Run 3 Relative Humidity | Ref_IN, Ref_OUT | K30_21, K30_22, K30_31, K30_32, K30_13, K30_14 |
| Bench | Run 1 Pressure Correction | Ref_IN | K30_21, K30_22 |
| | Run 2 Pressure Correction | Ref_IN | K30_21, K30_22 |
| | Run 3 Pressure Correction | Ref_IN | K30_21, K30_22 |
| | Run 4 Pressure Correction | Ref_IN | K30_21, K30_22 |
| | Pressure Time-response Learn 1 | Ref_IN | K30_21, K30_22, K30_31, K30_32 |
| | Pressure Time-response Learn 2 | Ref_IN | K30_21, K30_22, K30_31, K30_32 |
| | Pressure Time-response Test 1 | Ref_IN | K30_21, K30_22, K30_31, K30_32 |
| | Pressure Time-response Test 2 | Ref_IN | K30_21, K30_22, K30_31, K30_32 |
| | Run 1 Temperature | Ref_IN | K30_21, K30_22 |
| | Run 1 Relative Humidity | Ref_IN | K30_21, K30_22 |
| | Run 2 Temperature | Ref_IN | K30_21, K30_22 |
| | Run 2 Relative Humidity | Ref_IN | K30_21, K30_22 |
| | Run 3 Relative Humidity | Ref_IN | K30_21, K30_22 |

[Figure]

**Figure 1.** Data showing the impact of the Thunder Scientific 2500 chamber on the reported $CO_2$ values. The chamber was turned on at 01:29:37 and turned off at 08:24:43.

[Figure]

**Figure 2.** Environmental conditions during the second run of the Mesonet Temperature and Relative Humidity experiments. The chamber was turned on at 01:29:37 and turned off at 08:24:43. During this period the pressure (all sensors) and internal temperature (reference sensors) does not vary greatly.

**Table 1.** Metrics for the experimental conditions for the complete second run of the Mesonet Experiments.

| Variable | Sensor | Minimum | Maximum | Average | Standard deviation |
|---|---|---|---|---|---|
| Pressure [Pa] | Ref_OUT | 96910 | 97500 | 97262.68 | 105.94 |
| | Ref_IN | 96540 | 97160 | 96902.5 | 107.37 |
| | T_2 | 97066.57 | 97669.28 | 97430.29 | 106.42 |
| | T_3 | 97039.22 | 97665.22 | 97427.47 | 106.54 |
| Temperature [°C] | Ref_OUT | 50.91 | 50.91 | 50.91 | **0**[*] |
| | Ref_IN | 51.17 | 51.28 | 51.23 | 0.01 |
| | Chamber | 10.23 | 40.26 | - | - |
| Relative Humidity [%] | Chamber | 2.06 | 88.03 | - | - |

[*]**Please see section 10 of this document for explanation about this zero deviation.**

[Figure]

**Figure 3.** Comparison of the reference offset strategies. On the left the all sensors are brought to the level of the Ref_IN sensor, on the right, to the Ref_OUT sensor.

**3 Mesonet Experiment 1**

**3.1 Reference Intercomparison**

[Figure]

**Figure 4.** Environmental conditions during the intercomparison before the first Mesonet experiments.

**Table 2.** Metrics for the experimental conditions for the intercomparison before first Mesonet experiments.

| Variable | Sensor | Minimum | Maximum | Average | Standard deviation |
|---|---|---|---|---|---|
| Pressure [Pa] | Ref_OUT | 97700 | 98000 | 97819.24 | 87.62 |
| | Ref_IN | 97380 | 97740 | 97526.47 | 88.92 |
| Temperature [°C] | Ref_OUT | 50.91 | 50.91 | 50.91 | **0**[*] |
| | Ref_IN | 51.17 | 51.25 | 51.22 | 0.02 |
| $H_2O$ [ppt] | Ref_IN | 3.38 | 4.3 | 3.57 | 0.18 |

[*]**Please see section 10 of this document for explanation about this zero deviation.**

[Figure]

**Figure 5.** Results for the intercomparison before the first Mesonet experiment. At this date, the two reference presented a constant $54.75$ ppm offset. After correcting this offset, the RMSE was $1.63 * 10^{-14}$ ppm.

**Table 3.** Metrics for the intercomparison before the first Mesonet experiments.

| Variable | Sensor | Minimum | Maximum | Average | Standard deviation |
|----------|--------|---------|---------|---------|--------------------|
| | Ref_OUT | 413.91 | 443.85 | 424.57 | 6.51 |
| $CO_2$ [ppm] | Ref_IN | 360.24 | 387.49 | 369.82 | 6.14 |
| | ARM SGP | 412.1 | 464.98 | 429.74 | 11.85 |

[Figure]

**Figure 6.** Experimental conditions during the first Mesonet Temperature run.

**Table 4.** Metrics for the experimental conditions during the first Mesonet Temperature run.

| Variable | Sensor | Minimum | Maximum | Average | Standard deviation |
|---|---|---|---|---|---|
| | Ref_OUT | 98860 | 99290 | 99170.04 | 128.85 |
| Pressure [Pa] | Ref_IN | 98460 | 98970 | 98825.68 | 134.19 |
| | T_1 | 99133.8 | 99565.99 | 99445.19 | 127.6 |
| | Ref_OUT | 50.91 | 50.91 | 50.91 | **0**[*] |
| Temperature [°C] | Ref_IN | 51.17 | 51.28 | 51.23 | 0.01 |
| | Chamber | 10.74 | 40.32 | - | - |
| Relative Humidity [%] | Chamber | 43 | 47.09 | 45.64 | 0.98 |

[*]**Please see section 10 of this document for explanation about this zero deviation.**

**Figure 7.** Results for the first Mesonet Temperature run.

**Table 5.** Carbon Dioxide metrics for the first Mesonet Temperature run.

| Variable | Sensor | Minimum | Maximum | Average | Standard deviation |
|---|---|---|---|---|---|
| | Ref_OUT | 425.48 | 581.55 | 484.34 | 45.08 |
| | Ref_IN | 435.11 | 479.91 | 456.2 | 12.7 |
| $CO_2$ [ppm] | K30_11 | 435.35 | 501.35 | 464.47 | 18.86 |
| | K30_12 | 442.05 | 509.72 | 468.27 | 21.15 |

**Correlation for the Mesonet Temperature 1**

**Figure 8.** Scatter plots for the first Mesonet Temperature run.

**Table 6.** Linear fit metrics for the first Mesonet Temperature run.

| Sensor | Predictor | Slope | Y-Intercept | $R^2$ | RMSE |
|--------|-----------|-------|-------------|-------|------|
| Ref_IN | Pressure | -0.08 | 8212.38 | 0.68 | 7.19 |
| | Temperature | 0.6 | 440.47 | 0.25 | 11 |
| | Relative Humidity | 7.42 | 117.68 | 0.33 | 10.43 |
| | Ref_OUT | 0.25 | 337.26 | 0.76 | 6.22 |
| K30_11 | Pressure | -0.14 | 13911.4 | 0.84 | 7.61 |
| | Temperature | 1.32 | 430.2 | 0.54 | 12.79 |
| | Relative Humidity | 9.19 | 44.89 | 0.23 | 16.57 |
| | Ref_IN | 1.41 | -178.06 | 0.9 | 5.98 |
| | Ref_OUT | 0.4 | 270.02 | 0.92 | 5.31 |
| K30_12 | Pressure | -0.16 | 15918.25 | 0.88 | 7.38 |
| | Temperature | 1.62 | 426.07 | 0.65 | 12.5 |
| | Relative Humidity | 8.46 | 82.31 | 0.15 | 19.46 |
| | Ref_IN | 1.5 | -218.19 | 0.82 | 9.08 |
| | Ref_OUT | 0.45 | 248.4 | 0.94 | 5.37 |

**3.3 Relative Humidity**

[Figure]

**Figure 9.** Experimental conditions during the first Mesonet Relative Humidity run.

**Table 7.** Metrics for the experimental conditions during the first Mesonet Relative Humidity run.

| Variable | Sensor | Minimum | Maximum | Average | Standard deviation |
|---|---|---|---|---|---|
| | Ref_OUT | 97950 | 98410 | 98264.73 | 121.25 |
| Pressure [Pa] | Ref_IN | 97590 | 98100 | 97930.66 | 121.5 |
| | T_1 | 98225.48 | 98674.64 | 98537.13 | 115.1 |
| | Ref_OUT | 50.91 | 50.91 | 50.91 | 0 |
| Temperature [°C] | Ref_IN | 51.17 | 51.28 | 51.23 | 0.01 |
| | Chamber | 25.47 | 25.5 | 25.48 | 0.01 |
| Relative Humidity [%] | Chamber | 14.97 | 95.48 | - | - |

[Figure]

**Figure 10.** Results for the first Mesonet Relative Humidity run.

**Table 8.** Carbon Dioxide metrics for the first Mesonet Relative Humidity run.

| Variable | Sensor | Minimum | Maximum | Average | Standard deviation |
|---|---|---|---|---|---|
| | Ref_OUT | 428.93 | 593.21 | 483 | 52.16 |
| | Ref_IN | 447.85 | 510.86 | 465.9 | 19.74 |
| $CO_2$ [ppm] | K30_11 | 449.41 | 532.41 | 476.07 | 24.97 |
| | K30_12 | 449.43 | 532.8 | 476.58 | 24.91 |

**Correlation for the Mesonet Relative Humidity 1**

[Figure]

**Figure 11.** Scatter plots for the first Mesonet Relative Humidity run.

**Table 9.** Linear fit metrics for the first Mesonet Relative Humidity run.

| Sensor | Predictor | Slope | Y-Intercept | $R^2$ | RMSE |
|--------|-----------|-------|-------------|-------|------|
| | Pressure | 0.14 | -13756.32 | 0.44 | 18.64 |
| | Temperature | 1747.35 | -44046.02 | 0.19 | 22.52 |
| K30_11 | Relative Humidity | 0.79 | 427.22 | 0.69 | 13.91 |
| | Ref_IN | 1.25 | -107.72 | 0.98 | 3.48 |
| | Ref_OUT | 0.47 | 249.43 | 0.96 | 4.96 |
| | Pressure | 0.15 | -14021.01 | 0.46 | 18.27 |
| | Temperature | 1765.67 | -44512.15 | 0.19 | 22.4 |
| K30_12 | Relative Humidity | 0.8 | 427.3 | 0.71 | 13.52 |
| | Ref_IN | 1.25 | -104.56 | 0.98 | 3.81 |
| | Ref_OUT | 0.47 | 250.86 | 0.96 | 5.13 |

**4 Mesonet Experiment 2**

**4.1 Reference Sensors**

[Figure]

**Figure 12.** Experimental conditions for the intercomparison before second Mesonet experiments.

**Table 10.** Metrics for the experimental conditions for the intercomparison before second Mesonet experiments.

| Variable | Sensor | Minimum | Maximum | Average | Standard deviation |
|---|---|---|---|---|---|
| | Ref_OUT | 97220 | 97320 | 97261.68 | 23.66 |
| | Ref_IN | 96830 | 96980 | 96904.07 | 26.97 |
| Pressure [Pa] | T_1 | 97395.68 | 97470.13 | 97427.53 | 20.73 |
| | T_2 | 97393.41 | 97483.26 | 97430.47 | 25.02 |
| Temperature [°C] | Ref_OUT | 50.91 | 50.91 | 50.91 | 0 |
| | Ref_IN | 51.2 | 51.28 | 51.23 | 0.01 |
| $H_2O$ [ppt] | Ref_IN | 13.57 | 13.75 | 13.65 | 0.03 |

[Figure]

**Figure 13.** Results for the intercomparison before the second Mesonet experiment. At this date, the two reference presented a constant 18.23 ppm offset. After correcting this offset, the RMSE was $1.53 * 10^{-14}$ ppm.

**Table 11.** Metrics for the intercomparison before the second Mesonet experiments.

| Variable | Sensor | Minimum | Maximum | Average | Standard deviation |
|---|---|---|---|---|---|
| | Ref_OUT | 465.75 | 520.45 | 490.28 | 13.02 |
| | Ref_IN | 484.46 | 538.11 | 508.51 | 13.17 |
| | ARM SGP | 413.32 | 420.2 | 417.05 | 1.42 |
| $CO_2$ [ppm] | K30_21 | 425.5 | 481.5 | 449.68 | 13.55 |
| | K30_22 | 482 | 539 | 508.29 | 13.93 |
| | K30_31 | 529 | 585 | 552.27 | 13.48 |
| | K30_32 | 441 | 497 | 467.25 | 14.28 |

[Figure]

**Figure 14.** Experimental conditions during the second Mesonet Temperature run.

**Table 12.** Metrics for the experimental conditions during the second Mesonet Temperature run.

| Variable | Sensor | Minimum | Maximum | Average | Standard deviation |
|----------|--------|---------|---------|---------|--------------------|
| Pressure [Pa] | Ref_OUT | 97210 | 97400 | 97304.12 | 47.69 |
| | Ref_IN | 96830 | 97050 | 96944.33 | 45.83 |
| | T_2 | 97385.18 | 97565.27 | 97470.79 | 48.26 |
| | T_3 | 97385.21 | 97563.27 | 97468.38 | 47.66 |
| Temperature [°C] | Ref_OUT | 50.91 | 50.91 | 50.91 | 0 |
| | Ref_IN | 51.17 | 51.28 | 51.23 | 0.01 |
| | Chamber | 10.28 | 40.26 | - | - |
| Relative Humidity [%] | Chamber | 43.48 | 47.81 | 45.17 | 1.23 |

[Figure]

**Figure 15.** Results for the second Mesonet Temperature run.

**Table 13.** Carbon Dioxide metrics for the second Mesonet Temperature run.

| Variable | Sensor | Minimum | Maximum | Average | Standard deviation |
|---|---|---|---|---|---|
| | Ref_OUT | 430.39 | 444.43 | 434.57 | 3.68 |
| | Ref_IN | 434.32 | 446.82 | 438.07 | 2.47 |
| | K30_21 | 434.62 | 477.32 | 454.36 | 11.88 |
| $CO_2$ [ppm] | K30_22 | 436.42 | 477.51 | 453.11 | 11.29 |
| | K30_31 | 427.33 | 448.05 | 439.02 | 5.08 |
| | K30_32 | 435.23 | 471.76 | 451.68 | 10.29 |

**Correlation for the Mesonet Temperature 2**

[Figure]

**Figure 16.** Scatter plots for the second Mesonet Temperature run.

**Table 14.** Linear fit metrics for the second Mesonet Temperature run.

| Sensor | Predictor | Slope | Y-Intercept | $R^2$ | RMSE |
|--------|-----------|-------|-------------|-------|------|
| Ref_IN | Pressure | -0.02 | 2538.79 | 0.15 | 2.26 |
|  | Temperature | -0.17 | 442.25 | 0.42 | 1.87 |
|  | Relative Humidity | 0.68 | 407.18 | 0.12 | 2.31 |
|  | Ref_OUT | 0.44 | 245.07 | 0.44 | 1.83 |
| K30_21 | Pressure | 0.2 | -18931.25 | 0.65 | 6.99 |
|  | Temperature | 1.25 | 423.91 | 0.96 | 2.35 |
|  | Relative Humidity | -0.81 | 490.97 | 0.01 | 11.83 |
|  | Ref_IN | -2.47 | 1536.34 | 0.26 | 10.21 |
|  | Ref_OUT | -2.09 | 1363.93 | 0.42 | 9.03 |
| K30_22 | Pressure | 0.18 | -17025.1 | 0.59 | 7.24 |
|  | Temperature | 0.93 | 430.45 | 0.59 | 7.23 |
|  | Relative Humidity | 4.22 | 262.52 | 0.21 | 10.03 |
|  | Ref_IN | -1.05 | 912.85 | 0.05 | 10.98 |
|  | Ref_OUT | -1.82 | 1242.14 | 0.35 | 9.08 |
| K30_31 | Pressure | 0.01 | -235.81 | 0 | 5.05 |
|  | Temperature | 0.1 | 436.62 | 0.03 | 4.98 |
|  | Relative Humidity | -3.61 | 602.12 | 0.76 | 2.47 |
|  | Ref_IN | -0.15 | 503.29 | 0.01 | 5.05 |
|  | Ref_OUT | 0.24 | 335.75 | 0.03 | 4.99 |
| K30_32 | Pressure | 0.18 | -16989.37 | 0.69 | 5.75 |
|  | Temperature | 1.08 | 425.42 | 0.95 | 2.24 |
|  | Relative Humidity | -0.48 | 473.53 | 0 | 10.27 |
|  | Ref_IN | -2.08 | 1361.23 | 0.25 | 8.93 |
|  | Ref_OUT | -1.82 | 1243.47 | 0.43 | 7.79 |

**4.3 Relative Humidity**

[Figure]

**Figure 17.** Experimental conditions during the second Mesonet Relative Humidity run.

**Table 15.** Metrics for the experimental conditions during the second Mesonet Relative Humidity run.

| Variable | Sensor | Minimum | Maximum | Average | Standard deviation |
|---|---|---|---|---|---|
| Pressure [Pa] | Ref_OUT | 97100 | 97500 | 97342.03 | 111.34 |
| | Ref_IN | 96720 | 97160 | 96980.96 | 111 |
| | T_2 | 97274.8 | 97669.28 | 97511.81 | 112.58 |
| | T_3 | 97270.3 | 97665.22 | 97508.13 | 112.64 |
| Temperature [°C] | Ref_OUT | 50.91 | 50.91 | 50.91 | 0 |
| | Ref_IN | 51.2 | 51.28 | 51.23 | 0.01 |
| | Chamber | 21.95 | 26.76 | 26.2 | 0.95 |
| Relative Humidity [%] | Chamber | 14.87 | 88.03 | - | - |

[Figure]

**Figure 18.** Results for the second Mesonet Relative Humidity run.

**Table 16.** Carbon Dioxide metrics for the second Mesonet Relative Humidity run.

| Variable | Sensor | Minimum | Maximum | Average | Standard deviation |
|---|---|---|---|---|---|
| | Ref_OUT | 432.36 | 434.37 | 433.36 | 0.57 |
| | Ref_IN | 427.27 | 439.11 | 430.78 | 2.54 |
| | K30_21 | 429.1 | 443.48 | 438.27 | 3.27 |
| $CO_2$ [ppm] | K30_22 | 429.23 | 445.32 | 439.59 | 3.75 |
| | K30_31 | 421.89 | 439.33 | 434.08 | 3.09 |
| | K30_32 | 429.27 | 451.75 | 444.42 | 6.1 |

**Correlation for the Mesonet Relative Humidity 2**

**Figure 19.** Scatter plots for the second Mesonet Relative Humidity run.

**Table 17.** Linear fit metrics for the second Mesonet Relative Humidity run.

| Sensor | Predictor | Slope | Y-Intercept | $R^2$ | RMSE |
|---|---|---|---|---|---|
| | Pressure | -0.01 | 1489.63 | 0.14 | 3.02 |
| | Temperature | 2.65 | 368.72 | 0.6 | 2.06 |
| K30_21 | Relative Humidity | 0.1 | 432.97 | 0.65 | 1.92 |
| | Ref_IN | -0.25 | 544.31 | 0.04 | 3.19 |
| | Ref_OUT | 3.42 | -1043.98 | 0.35 | 2.62 |
| | Pressure | -0.02 | 2062.65 | 0.25 | 3.24 |
| | Temperature | 3.22 | 355.25 | 0.67 | 2.16 |
| K30_22 | Relative Humidity | 0.11 | 433.84 | 0.58 | 2.43 |
| | Ref_IN | -0.18 | 519.04 | 0.02 | 3.71 |
| | Ref_OUT | 4.7 | -1597.44 | 0.5 | 2.64 |
| | Pressure | 0 | 128.65 | 0.01 | 3.05 |
| | Temperature | 0.84 | 412.08 | 0.07 | 2.96 |
| K30_31 | Relative Humidity | 0.08 | 429.5 | 0.55 | 2.07 |
| | Ref_IN | -0.36 | 589.56 | 0.09 | 2.93 |
| | Ref_OUT | 0.23 | 335.68 | 0 | 3.07 |
| | Pressure | -0.02 | 2753.61 | 0.19 | 5.48 |
| | Temperature | 5.39 | 303.33 | 0.7 | 3.32 |
| K30_32 | Relative Humidity | 0.18 | 434.63 | 0.63 | 3.69 |
| | Ref_IN | -0.52 | 668.32 | 0.05 | 5.95 |
| | Ref_OUT | 7.6 | -2850.09 | 0.5 | 4.32 |

**5 Mesonet Experiment 3**

**5.1 Temperature**

[Figure]

**Figure 20.** Experimental conditions during the third Mesonet Temperature run.

**Table 18.** Metrics for the experimental conditions during the third Mesonet Temperature run.

| Variable | Sensor | Minimum | Maximum | Average | Standard deviation |
|---|---|---|---|---|---|
| | Ref_OUT | 96730 | 97170 | 96886.92 | 133.07 |
| | Ref_IN | 96350 | 96820 | 96517.46 | 131.51 |
| Pressure [Pa] | T_1 | 96893.91 | 97325.16 | 97034.12 | 135.93 |
| | T_2 | 96909.97 | 97343.58 | 97050.6 | 134.92 |
| | T_3 | 96895.46 | 97326.49 | 97034.11 | 134.52 |
| | Ref_OUT | 50.91 | 50.91 | 50.91 | 0 |
| Temperature [°C] | Ref_IN | 51.17 | 51.28 | 51.23 | 0.01 |
| | Chamber | 10.44 | 40.98 | - | - |
| Relative Humidity [%] | Chamber | 43.78 | 49.27 | 45.56 | 1.05 |

[Figure]

**Figure 21.** Results for the third Mesonet Temperature run.

**Table 19.** Carbon Dioxide metrics for the third Mesonet Temperature run.

| Variable | Sensor | Minimum | Maximum | Average | Standard deviation |
|---|---|---|---|---|---|
| | Ref_OUT | 436.43 | 465.98 | 449.53 | 8.36 |
| | Ref_IN | 451.11 | 467.77 | 456.67 | 4.03 |
| | K30_13 | 454.01 | 486.43 | 468.41 | 8.42 |
| | K30_14 | 452.86 | 485.25 | 467.83 | 7.9 |
| $CO_2$ [ppm] | K30_21 | 452.92 | 499.52 | 475.51 | 12.3 |
| | K30_22 | 446.99 | 493.18 | 471.16 | 11.07 |
| | K30_31 | 446.81 | 469.85 | 458.34 | 4.97 |
| | K30_32 | 452.9 | 492.42 | 472.37 | 10.47 |

**Correlation for the Mesonet Temperature 3**

[Figure]

**Figure 22.** Scatter plots for the third Mesonet Temperature run.

**Table 20.** Linear fit metrics for the third Mesonet Temperature run.

| Sensor | Predictor | Slope | Y-Intercept | $R^2$ | RMSE |
|---|---|---|---|---|---|
| Ref_IN | Pressure | -0.02 | 2078.44 | 0.3 | 3.37 |
| | Temperature | -0.3 | 464.16 | 0.51 | 2.81 |
| | Relative Humidity | 0.09 | 452.78 | 0 | 4.02 |
| | Ref_OUT | 0.32 | 313.36 | 0.44 | 3.01 |
| K30_13 | Pressure | 0.04 | -3669.06 | 0.47 | 6.1 |
| | Temperature | 0.78 | 448.64 | 0.81 | 3.64 |
| | Relative Humidity | -1.81 | 550.89 | 0.05 | 8.2 |
| | Ref_IN | -0.82 | 842.92 | 0.15 | 7.74 |
| | Ref_OUT | -0.21 | 562.65 | 0.04 | 8.23 |
| K30_14 | Pressure | 0.04 | -3251.44 | 0.44 | 5.93 |
| | Temperature | 0.7 | 450.08 | 0.75 | 3.98 |
| | Relative Humidity | -1.03 | 514.56 | 0.02 | 7.82 |
| | Ref_IN | -0.67 | 772.65 | 0.12 | 7.42 |
| | Ref_OUT | -0.24 | 574.91 | 0.06 | 7.64 |
| K30_21 | Pressure | 0.06 | -5286.77 | 0.42 | 9.33 |
| | Temperature | 1.21 | 444.97 | 0.91 | 3.71 |
| | Relative Humidity | -3.32 | 626.54 | 0.08 | 11.8 |
| | Ref_IN | -1.53 | 1175.94 | 0.25 | 10.64 |
| | Ref_OUT | -0.36 | 637.62 | 0.06 | 11.92 |
| K30_22 | Pressure | 0.05 | -4347.35 | 0.37 | 8.81 |
| | Temperature | 1.05 | 444.56 | 0.85 | 4.25 |
| | Relative Humidity | -2.07 | 565.27 | 0.04 | 10.85 |
| | Ref_IN | -1.37 | 1094.61 | 0.25 | 9.6 |
| | Ref_OUT | -0.33 | 619.19 | 0.06 | 10.72 |
| K30_31 | Pressure | 0 | 321.94 | 0 | 4.96 |
| | Temperature | -0.1 | 460.86 | 0.04 | 4.87 |
| | Relative Humidity | -2.6 | 576.97 | 0.3 | 4.15 |
| | Ref_IN | 0.76 | 109.91 | 0.38 | 3.9 |
| | Ref_OUT | 0.44 | 262.08 | 0.54 | 3.36 |
| K30_32 | Pressure | 0.06 | -4964.63 | 0.52 | 7.27 |
| | Temperature | 1.01 | 446.77 | 0.88 | 3.59 |
| | Relative Humidity | -4.11 | 659.47 | 0.17 | 9.55 |
| | Ref_IN | -1.23 | 1032.26 | 0.22 | 9.24 |
| | Ref_OUT | -0.16 | 546.35 | 0.02 | 10.38 |

**5.2 Relative Humidity**

[Figure]

**Figure 23.** Experimental conditions during the third Mesonet Relative Humidity run.

**Table 21.** Metrics for the experimental conditions during the third Mesonet Relative Humidity run.

| Variable | Sensor | Minimum | Maximum | Average | Standard deviation |
|---|---|---|---|---|---|
| | Ref_OUT | 96660 | 97020 | 96787.52 | 101.93 |
| | Ref_IN | 96250 | 96680 | 96413.49 | 103.39 |
| Pressure [Pa] | T_1 | 96805.94 | 97161.59 | 96924.18 | 98.27 |
| | T_2 | 96821.58 | 96951.79 | 96908.85 | 36.22 |
| | T_3 | 96803.85 | 97169.14 | 96932.27 | 100.46 |
| | Ref_OUT | 50.91 | 50.91 | 50.91 | 0 |
| Temperature [°C] | Ref_IN | 51.17 | 51.28 | 51.23 | 0.01 |
| | Chamber | 25.83 | 27.61 | 27.08 | 0.33 |
| Relative Humidity [%] | Chamber | 15.1 | 85.4 | - | - |

[Figure]

**Figure 24.** Results for the third Mesonet Relative Humidity run.

**Table 22.** Carbon Dioxide metrics for the third Mesonet Relative Humidity run.

| Variable | Sensor | Minimum | Maximum | Average | Standard deviation |
|---|---|---|---|---|---|
| | Ref_OUT | 426.49 | 431.72 | 428.72 | 1.45 |
| | Ref_IN | 424.45 | 437.28 | 428.66 | 2.46 |
| | K30_13 | 429.9 | 442.38 | 436.38 | 3.13 |
| | K30_14 | 429.07 | 447.24 | 439.24 | 4.75 |
| $CO_2$ [ppm] | K30_21 | 429.52 | 441.41 | 436.28 | 2.66 |
| | K30_22 | 429.48 | 447.17 | 439.36 | 5.56 |
| | K30_31 | 428.83 | 442.83 | 436.88 | 3.59 |
| | K30_32 | 429.72 | 440.2 | 435.62 | 2.63 |

**Correlation for the Mesonet Relative Humidity 3**

[Figure]

**Figure 25.** Scatter plots for the third Mesonet Relative Humidity run.

**Table 23.** Linear fit metrics for the third Mesonet Relative Humidity run.

| Sensor | Predictor | Slope | Y-Intercept | $R^2$ | RMSE |
|--------|-----------|-------|-------------|-------|------|
| | Pressure | 0.01 | -92.64 | 0.03 | 3.08 |
| | Temperature | 6.31 | 265.53 | 0.45 | 2.33 |
| K30_13 | Relative Humidity | 0.1 | 430.8 | 0.75 | 1.55 |
| | Ref_IN | -0.75 | 756.54 | 0.34 | 2.53 |
| | Ref_OUT | -1.27 | 981.63 | 0.35 | 2.52 |
| | Pressure | 0.01 | -236.3 | 0.02 | 4.68 |
| | Temperature | 9.54 | 180.9 | 0.44 | 3.53 |
| K30_14 | Relative Humidity | 0.16 | 430.51 | 0.8 | 2.1 |
| | Ref_IN | -1.2 | 955.06 | 0.39 | 3.71 |
| | Ref_OUT | -1.89 | 1249.99 | 0.33 | 3.86 |
| | Pressure | 0.06 | -5512.96 | 0.7 | 1.45 |
| | Temperature | 7.11 | 243.81 | 0.79 | 1.23 |
| K30_21 | Relative Humidity | 0.06 | 432.81 | 0.41 | 2.04 |
| | Ref_IN | -0.31 | 569.21 | 0.08 | 2.54 |
| | Ref_OUT | -1.59 | 1117.18 | 0.75 | 1.32 |
| | Pressure | 0.1 | -9486.42 | 0.45 | 4.13 |
| | Temperature | 12.72 | 94.92 | 0.57 | 3.62 |
| K30_22 | Relative Humidity | 0.08 | 435.27 | 0.13 | 5.18 |
| | Ref_IN | -0.39 | 604.51 | 0.03 | 5.46 |
| | Ref_OUT | -3.61 | 1988.63 | 0.89 | 1.84 |
| | Pressure | 0 | 18.48 | 0.01 | 3.55 |
| | Temperature | 6.75 | 254.08 | 0.39 | 2.8 |
| K30_31 | Relative Humidity | 0.12 | 430.45 | 0.76 | 1.75 |
| | Ref_IN | -0.73 | 748.53 | 0.25 | 3.11 |
| | Ref_OUT | -1.28 | 983.93 | 0.27 | 3.06 |
| | Pressure | 0.01 | -242.33 | 0.07 | 2.52 |
| | Temperature | 5.56 | 285.04 | 0.49 | 1.86 |
| K30_32 | Relative Humidity | 0.08 | 431.25 | 0.66 | 1.52 |
| | Ref_IN | -0.44 | 624.66 | 0.17 | 2.38 |
| | Ref_OUT | -1.14 | 923.1 | 0.4 | 2.03 |

[Figure]

**Figure 26.** Experimental conditions for the intercomparison after third Mesonet experiments.

**Table 24.** Metrics for the experimental conditions for the intercomparison after third Mesonet experiments.

| Variable | Sensor | Minimum | Maximum | Average | Standard deviation |
|---|---|---|---|---|---|
| Pressure [Pa] | Ref_OUT | 96030 | 96120 | 96075.97 | 22.5 |
| | Ref_IN | 95850 | 95980 | 95921.67 | 24.51 |
| Temperature [°C] | Ref_OUT | 50.91 | 50.91 | 50.91 | 0 |
| | Ref_IN | 51.17 | 51.25 | 51.23 | 0.01 |
| $H_2O$ [ppt] | Ref_IN | 18.03 | 20.88 | 18.87 | 0.61 |

[Figure]

**Figure 27.** Results for the intercomparison after the third Mesonet experiment. At this date, the two reference presented a constant $17.24$ ppm offset. After correcting this offset, the RMSE was $1.12 * 10^{-14}$ ppm.

**Table 25.** Metrics for the intercomparison after the third Mesonet experiments.

| Variable | Sensor | Minimum | Maximum | Average | Standard deviation |
|----------|--------|---------|---------|---------|--------------------|
|          | Ref_OUT | 398.62 | 437.86 | 413.05 | 8.74 |
| $CO_2$ [ppm] | Ref_IN | 417.4 | 453.29 | 430.29 | 8.27 |
|          | ARM SGP | 415.3 | 420.81 | 418.31 | 1.15 |

**6 Benchtop Experiments 1**

**6.1 Temperature**

[Figure]

**Figure 28.** Experimental conditions during the first Bench Temperature run.

**Table 26.** Metrics for the experimental conditions during the first Bench Temperature run.

| Variable | Sensor | Minimum | Maximum | Average | Standard deviation |
|---|---|---|---|---|---|
| Pressure [Pa] | Ref_IN | 97300 | 97370 | 97342.1 | 13.57 |
| | T_2 | 97617.75 | 97634.8 | 97626.68 | 2.82 |
| Temperature [°C] | Ref_IN | 51.17 | 51.25 | 51.23 | 0.01 |
| | T_2 | 20.75 | 53.41 | - | - |
| Relative Humidity [%] | T_2 | 10.62 | 50.09 | 28.31 | 11.43 |

[Figure]

**Figure 29.** Results for the first Bench Temperature run.

**Table 27.** Carbon Dioxide metrics for the first Bench Temperature run.

| Variable | Sensor | Minimum | Maximum | Average | Standard deviation |
|----------|--------|---------|---------|---------|--------------------|
|          | Ref_IN | 459.04  | 545.15  | 492.49  | 23.76              |
| $CO_2$ [ppm] | K30_21 | 497.84 | 559.03 | 525.9 | 17.94 |
|          | K30_22 | 483.39  | 558.39  | 518.05  | 23.47              |

**Correlation for the Bench Temperature 1**

[Figure]

**Figure 30.** Scatter plots for the first Bench Temperature run.

**Table 28.** Linear fit metrics for the first Bench Temperature run.

| Sensor | Predictor | Slope | Y-Intercept | $R^2$ | RMSE |
|--------|-----------|-------|-------------|-------|------|
| K30_21 | Pressure | -3.01 | 294698.42 | 0.22 | 15.79 |
|        | Temperature | 0.22 | 518.82 | 0.01 | 17.82 |
|        | Relative Humidity | 0.28 | 517.94 | 0.03 | 17.63 |
|        | Ref_IN | 0.36 | 346.62 | 0.23 | 15.7 |
| K30_22 | Pressure | -4.04 | 394541.7 | 0.23 | 20.51 |
|        | Temperature | 0.1 | 514.98 | 0 | 23.43 |
|        | Relative Humidity | 0.54 | 502.84 | 0.07 | 22.63 |
|        | Ref_IN | 0.53 | 256.9 | 0.29 | 19.78 |

**6.2 Relative Humidity**

[Figure]

**Figure 31.** Experimental conditions during the first Bench Relative Humidity run.

**Table 29.** Metrics for the experimental conditions during the first Bench Relative Humidity run.

| Variable | Sensor | Minimum | Maximum | Average | Standard deviation |
|---|---|---|---|---|---|
| Pressure [Pa] | Ref_IN | 97320 | 97390 | 97353 | 13.16 |
| | T_2 | 97627.66 | 97659.9 | 97640.33 | 5.71 |
| Temperature [°C] | Ref_IN | 51.17 | 51.25 | 51.23 | 0.01 |
| | T_2 | 20.9 | 22.79 | 21.81 | 0.58 |
| Relative Humidity [%] | T_2 | 48.27 | 74.04 | - | - |

[Figure]

**Figure 32.** Results for the first Bench Relative Humidity run.

**Table 30.** Carbon Dioxide metrics for the first Bench Relative Humidity run.

| Variable | Sensor | Minimum | Maximum | Average | Standard deviation |
|---|---|---|---|---|---|
| | Ref_IN | 415.71 | 499.36 | 443.18 | 16.68 |
| $CO_2$ [ppm] | K30_21 | 429.07 | 440.17 | 434 | 2.74 |
| | K30_22 | 427.3 | 438.15 | 433.97 | 2.85 |

**Correlation for the Bench Relative Humidity 1**

[Figure]

**Figure 33.** Scatter plots for the first Bench Relative Humidity run.

**Table 31.** Linear fit metrics for the first Bench Relative Humidity run.

| Sensor | Predictor | Slope | Y-Intercept | $R^2$ | RMSE |
|---|---|---|---|---|---|
| K30_21 | Pressure | -0.28 | 27598.92 | 0.34 | 2.23 |
| | Temperature | -2.36 | 485.44 | 0.25 | 2.37 |
| | Relative Humidity | -0.04 | 436.72 | 0.02 | 2.71 |
| | Ref_IN | 0.06 | 406.28 | 0.14 | 2.54 |
| K30_22 | Pressure | -0.29 | 28917.54 | 0.34 | 2.31 |
| | Temperature | -2.99 | 499.14 | 0.37 | 2.26 |
| | Relative Humidity | -0.02 | 435.4 | 0.01 | 2.84 |
| | Ref_IN | 0.04 | 418.13 | 0.04 | 2.79 |

**7 Benchtop Experiments 2**

**7.1 Temperature**

[Figure]

**Figure 34.** Experimental conditions during the second Bench Temperature run.

**Table 32.** Metrics for the experimental conditions during the second Bench Temperature run.

| Variable | Sensor | Minimum | Maximum | Average | Standard deviation |
|---|---|---|---|---|---|
| Pressure [Pa] | Ref_IN | 97440 | 97530 | 97482.13 | 16.39 |
| | T_2 | 97754.06 | 97773.67 | 97762.57 | 5.1 |
| Temperature [°C] | Ref_IN | 51.17 | 51.25 | 51.23 | 0.01 |
| | T_2 | 21.51 | 39.81 | - | - |
| Relative Humidity [%] | T_2 | 19.61 | 41.42 | 33.56 | 6.33 |

[Figure]

**Figure 35.** Results for the second Bench Temperature run.

**Table 33.** Carbon Dioxide metrics for the second Bench Temperature run.

| Variable | Sensor | Minimum | Maximum | Average | Standard deviation |
|---|---|---|---|---|---|
| | Ref_IN | 406.13 | 427.64 | 413.68 | 6.41 |
| CO$_2$ [ppm] | K30_21 | 403.99 | 423.49 | 412.55 | 6.05 |
| | K30_22 | 397.77 | 416.77 | 407.28 | 4.96 |

**Correlation for the Bench Temperature 2**

[Figure]

**Figure 36.** Scatter plots for the second Bench Temperature run.

**Table 34.** Linear fit metrics for the second Bench Temperature run.

| Sensor | Predictor | Slope | Y-Intercept | $R^2$ | RMSE |
|--------|-----------|-------|-------------|-------|------|
| K30_21 | Pressure | -0.62 | 61137.65 | 0.27 | 5.15 |
| | Temperature | -0.03 | 413.39 | 0 | 6.04 |
| | Relative Humidity | -0.18 | 418.56 | 0.04 | 5.94 |
| | Ref_IN | 0.73 | 112.51 | 0.59 | 3.87 |
| K30_22 | Pressure | -0.05 | 5327.96 | 0 | 4.95 |
| | Temperature | -0.23 | 413.29 | 0.04 | 4.86 |
| | Relative Humidity | 0.02 | 406.54 | 0 | 4.96 |
| | Ref_IN | 0.41 | 238.09 | 0.28 | 4.21 |

**7.2 Relative Humidity**

[Figure]

**Figure 37.** Experimental conditions during the second Bench Relative Humidity run.

**Table 35.** Metrics for the experimental conditions during the second Bench Relative Humidity run.

| Variable | Sensor | Minimum | Maximum | Average | Standard deviation |
|---|---|---|---|---|---|
| Pressure [Pa] | Ref_IN | 97400 | 97470 | 97439.71 | 15.51 |
| | T_2 | 97707.94 | 97724.89 | 97717.96 | 4.32 |
| Temperature [°C] | Ref_IN | 51.17 | 51.25 | 51.22 | 0.01 |
| | T_2 | 19.8 | 22.3 | 21.44 | 0.69 |
| Relative Humidity [%] | T_2 | 38.9 | 57.85 | 45.78 | 5.99 |

[Figure]

**Figure 38.** Results for the second Bench Relative Humidity run.

**Table 36.** Carbon Dioxide metrics for the second Bench Relative Humidity run.

| Variable | Sensor | Minimum | Maximum | Average | Standard deviation |
|---|---|---|---|---|---|
| | Ref_IN | 400.31 | 403.6 | 401.63 | 1.04 |
| $CO_2$ [ppm] | K30_21 | 399.36 | 402.36 | 400.68 | 0.79 |
| | K30_22 | 400.24 | 403.09 | 401.65 | 0.75 |

**Correlation for the Bench Relative Humidity 2**

[Figure]

**Figure 39.** Scatter plots for the second Bench Relative Humidity run.

**Table 37.** Linear fit metrics for the second Bench Relative Humidity run.

| Sensor | Predictor | Slope | Y-Intercept | $R^2$ | RMSE |
|--------|-----------|-------|-------------|-------|------|
| K30_21 | Pressure | -0.18 | 18403.68 | 0.59 | 0.67 |
| | Temperature | -0.57 | 413.78 | 0.14 | 0.96 |
| | Relative Humidity | 0.06 | 398.89 | 0.12 | 0.97 |
| | Ref_IN | 1.14 | -57.65 | 0.68 | 0.59 |
| K30_22 | Pressure | -0.04 | 4655.61 | 0.06 | 0.76 |
| | Temperature | -0.23 | 405.59 | 0.04 | 0.77 |
| | Relative Humidity | 0.03 | 399.45 | 0.04 | 0.77 |
| | Ref_IN | 0.55 | 180.63 | 0.27 | 0.67 |

**8 Benchtop Experiments 3**

**8.1 Relative Humidity**

[Figure]

**Figure 40.** Experimental conditions during the third Bench Relative Humidity run.

**Table 38.** Metrics for the experimental conditions during the third Bench Relative Humidity run.

| Variable | Sensor | Minimum | Maximum | Average | Standard deviation |
|---|---|---|---|---|---|
| Pressure [Pa] | Ref_IN | 97220 | 97290 | 97259.79 | 13.42 |
| | T_2 | 97526.79 | 97542.3 | 97533.86 | 3.63 |
| Temperature [°C] | Ref_IN | 51.17 | 51.25 | 51.23 | 0.01 |
| | T_2 | 22.38 | 25.39 | 23.82 | 1.02 |
| Relative Humidity [%] | T_2 | 38.16 | 80.56 | - | - |

[Figure]

**Figure 41.** Results for the third Bench Relative Humidity run.

**Table 39.** Carbon Dioxide metrics for the third Bench Relative Humidity run.

| Variable | Sensor | Minimum | Maximum | Average | Standard deviation |
|---|---|---|---|---|---|
| | Ref_IN | 394.42 | 403.21 | 397.29 | 2.59 |
| $CO_2$ [ppm] | K30_21 | 393.69 | 400.28 | 396.63 | 1.79 |
| | K30_22 | 393.31 | 400.31 | 395.75 | 1.55 |

**Correlation for the Bench Relative Humidity 3**

[Figure]

**Figure 42.** Scatter plots for the third Bench Relative Humidity run.

**Table 40.** Linear fit metrics for the third Bench Relative Humidity run.

| Sensor | Predictor | Slope | Y-Intercept | $R^2$ | RMSE |
|---|---|---|---|---|---|
| K30_21 | Pressure | -0.28 | 27483.54 | 0.32 | 1.48 |
| | Temperature | -0.95 | 419.17 | 0.29 | 1.51 |
| | Relative Humidity | 0.02 | 395.46 | 0.02 | 1.77 |
| | Ref_IN | 0.59 | 161.25 | 0.73 | 0.92 |
| K30_22 | Pressure | -0.08 | 7801.26 | 0.03 | 1.52 |
| | Temperature | -0.68 | 411.93 | 0.2 | 1.39 |
| | Relative Humidity | 0.05 | 393.19 | 0.15 | 1.43 |
| | Ref_IN | 0.38 | 246.1 | 0.4 | 1.2 |

**9 Ideal Pressure Time Response**

[Figure]

**Figure 43.** Example of idealized signal for pressure time-response correction. It is important to note that the referred idealized signal is an artificial signal with the pressure error but without the pressure time-response error. Therefore, ideal means the ideal impact of pressure (without pressure time-response error). It serves only as an evaluator of the pressure time-response algorithms. It should not be used for any pressure correction algorithms. The ideal signal was generated using the timestamps of the pressure step changes and the $CO_2$ values of the K30 after stabilization, after the step change.

**10 Temperature Li-820 (Ref_OUT)**

In all experimental condition tables in this document, the reported temperature for the optical chamber of the Li-820 (a.k.a. Ref_OUT) is 50.91 degrees Celsius with its standard deviation equal to zero. This may appear to be a manuscript preparation error (e.g., a copy and paste error), but it is not. To investigate the matter we first evaluated if our dataset ever showed any temperature different than 50.91 for this sensor. At beginning of all experiments our loggers recorded the warm-up ramp of this sensor with temperatures below 50.91 (as can be seen in figures 44 and 45). However, after this warm-up period the sensor does not report a value different than 50.91. Analyzing the temperatures reported by the Li-840A (a.k.a. Ref_IN), the largest deviation reported by this sensor for all experiments was 0.02 degrees Celsius. Therefore it is possible that the analog to digital converter (ADC) in the Li-820 is not capable of detecting these small fluctuations. Another contributing factor to the standard

deviation equal to zero, is our data trimming strategy. We only calculates the deviation for the test periods. At the beginning of each test period, the Li-820's heater has had at least one hour to stabilize temperature of the sensor's optical chamber. At this point we do not have any reason to believe any malfunction on the sensor.

[Figure]

**Figure 44.** Complete data series for the experimental conditions of the second run of the Mesonet Temperature and Relative humidity experiments showing temperatures lower than 50.91 for a minutes.

[Figure]

**Figure 45.** Zoomed view of figure 44 showing a temperature variation of the Li-820 (a.k.a. Ref_OUT) optical chamber.

---

## Author Comment (AC2)

This document contains item-by-item responses to the reviewer's comments. The reviewer's comments are in black, non-italicized, regular fonts. Author responses are in *blue, italicized* fonts. Changes to the manuscript will be provided during revised submission, per instructions of the Copernicus editorial team (i.e., revised manuscript and diff file). Nonetheless, examples of the corrections and a detailed view of each experiment are provided in the attached discussion supplement.

General comments:

We thank Reviewer #2 for his comments and suggestions. Following his (and reviewer's #1) suggestions, we repeated the Mesonet temperature and relative humidity experiments under more controlled conditions, added more test sensors, and adjusted the result presentation. Therefore, the final comment below is written in light of the following changes:

- The Li-820 sensor originally named $CO_2$ Independent Sensor Outside is renamed Reference Sensor Out (Ref_OUT).
- The Li-840A sensor originally named $CO_2$ Independent Sensor Inside is renamed Reference Sensor In (Ref_IN).
- The Senseair K30 sensors originally named $CO_2$ Test Sensors 1 and 2 are renamed K30_##, where the first digit refers to its attached system (i.e. logger, temperature, and relative humidity sensors) and the second digit is its identification. This way the first Senseair K30 sensor of test system 2 is named K30_21.
- The Mesonet temperature and relative humidity experiments now have a "run" identification (i.e., "Mesunet Run 1 - Temperature", "Mesunet Run 2 - Temperature", "Mesunet Run 1 - Relative humidity", etc), where Run 1 is the data originally presented in the manuscript and Runs 2 and 3 are the repetition runs.

A "Discussion Supplement" (appended to the end of this reply) shows a summary of experiments and sensors used following the model in Arzoumanian (2019; doi:10.5194/amt-12-2665-2019), supporting material to explain the low $CO_2$ values seen in the Mesonet Experiment (Run 1). The results of the correction application on the Mesonet T/RH Experiment (Run 1) dataset, followed by the results for the Mesonet T/RH Experiment (Run 2 and 3), and the adjusted Bench temperature and relative humidity Experiments. The document ends with the supporting plot for the Pressure time-response correction ("ideal signal") and a brief discussion about the reported temperature from the Li-820 reference sensor.

Please note that even though the results for the correction application on the Mesonet T/RH Experiment (Run 1) dataset are presented in the attached "Discussion Supplement", they serve only as a comparison of the correction method. Following the reviewer's suggestion, the Mesonet T/RH Experiment (Run 1) dataset will be discarded. Only Mesonet T/RH Experiment (Run 2 and 3) will be analyzed in the revised manuscript.

—------------------------—----------------------—-------------------------------—--------------------

*Reply for* Anonymous Referee #2

General comments:

The first major concern is the lack of a calibrated and reliable reference dataset for ambient air CO2 dry air mole fractions (or as referred to in manuscript: the CO2 concentration). The authors LiCOR LI-840 and LI-820 systems as a reference. However, it is obvious that neither of the two instruments was appropriately calibrated. An indoor air concentration of 200-300ppm CO2 as reported in Figure 3 is completely unrealistic.

Ambient clean air data from NOAA can be found here: Global Monitoring Laboratory - Carbon Cycle Greenhouse Gases (noaa.gov) demonstrating that 200-300 ppm is not possible unless in artificial gas mixtures or environments. Given that the LiCORs are not calibrated it is also unlikely that they were properly tested for cross-sensitivities, offsets or non-linearity.

*We agree with the reviewer. The 200-300 ppm values shown are not realistic for indoor concentrations. However, it is not an artifact of unreliable reference sensors. This considerable reduction in $CO_2$ concentration is caused by the Thunder Scientific 2500 chamber. Figure 1 in the discussion supplement shows the raw ppm values for the Li-840A (Ref_IN), the Li-820 (Ref_OUT), and four K30 sensors, before, during, and after the second run of the mesonet temp/RH experiments. In this figure, the Ref_IN and all four K30 sensors drop their reported values from ~500 ppm to ~250 ppm after the start of the experiment and return to ~450 ppm after the experiment ends. For the same time periods, the outside reference (Ref_OUT) is not affected.*

*Looking at the documentation of the Thunder Scientific 2500 chamber, there is mention of the use of Nitrogen to control pressure, temperature, and humidity. However, the chamber at the Mesonet Calibration Laboratory does not use this feature. In their configuration, the chamber uses water, a series of compressors, and pre-chambers to generate standard test conditions. More information about the inner workings of this chamber can be found at https://www.thunderscientific.com/humidity_equipment/model_2500.html.*

*Searching for an explanation for this $CO_2$ reduction we also investigated the behavior of the other variables during the experiment. The data in figure 2 and table 1, in the discussion supplement, show that none of the pressure sensors varied greatly (all sensors show a standard deviation of approximately 106 Pa). The data also shows this effect happens before large temperature and humidity changes and is also seen on the Ref_IN control sensor (which is independent of temperature and humidity changes in the test ranges).*

*After our thorough review of the documentation, it was not clear the exact source of this effect. Nevertheless, it is apparent when analyzing the data that this effect has a near-constant behavior throughout the experiments. Therefore, an offset correction can be applied without loss of generality.*

*In the data presented in the manuscript, we opted to bring all data to the level of the Ref_IN sensor (Li-840A). This decision was based on the robustness of the Li-840A when compared to the Li-820 (Ref_OUT), and our wish to analyze the data at the relative reference point of the chamber's environment. In hindsight, we understand how the unrealistic data presented with a lack of an explanation would confuse readers. Figure 3 of the discussion supplement illustrates the impact of these two different correction strategies. In the left panel, the result of correcting the data to the Ref_IN sensor, and in the right panel the result of correcting the data to the Ref_OUT sensor.*

*In the revised manuscript we have used the Ref_OUT sensor to correct the data from the mesonet experiments. we used the 60 minutes prior to the experiment (e.g., from 02:00 to 03:00 in figures 1 and 2) where the conditions are stable, to find an average offset for each sensor to the Ref_OUT sensor. This offset was then applied to the entire time series. An explanation of this strategy was also added in the revised manuscript.*

*To support our claims about the quality of the measurements of the Li-840A (Ref_IN) and the Li-820 (Ref_OUT), we have added plots and tables in the discussion supplement showing intercomparisons between the reference and test sensors, before and after each mesonet experiment. These comparisons support our strategy to use the Ref_OUT sensor to correct the data from the mesonet experiments. In these plots, the $CO_2$ concentration measured by the Atmospheric Radiation Measurement - Southern Great Plains (ARM SGP) reference tower was added to show these are reasonable atmospheric values for Oklahoma.* **We understand the distance between Norman and the ARM-SGP tower does not allow for a direct comparison of the data.** *Nevertheless, it is presented here to demonstrate to the reviewers that the two reference sensors were measuring values within reasonable expectations. A version of these additional plots and tables with basic statistical metrics (without the ARM-SGP data), was added to the revised manuscript supplement.*

*We also add here that the Li-840A was sent back to LiCOR for calibration on Jun/06/2020. The report provided by the company indicates the sensor had an offset of 1.098 ppm to Zero ppm and offsets of 0.9809 and 0.0148 to the two Span references.*

The LI-820 and LI-40 have known temperature dependent drifts. According to the LI-840 manual this the calibration drift is <0.4ppm per degree C 840A_Manual_10690.pdf | Powered by Box (boxenterprise.net)

How can the reader be sure that there isn't a residual drift in the reference data?

*The Li-840A and Li-820 have internal heaters that elevate their sampling chambers to a temperature of ~51 °C (reported in the discussion supplement submitted with this response), which is above the temperatures tested in our experiments (max:~40 °C). This characteristic eliminates their dependence within the range tested. These assumptions are validated by the new plots and tables for experimental conditions (e.g., discussion supplement figure 14 and table 13) and the results from the mesonet runs 2 and 3 (e.g. discussion supplement tables 14 and 20, and figures 14, 15, 20, and 21). In addition, the discussion supplement tables 15 and 21 show the slope, y-intercept, $R^2$, and RMSE for the Li-840A against pressure, temperature, relative humidity, and Ref_OUT measurements*

*during mesonet temperature experiments 2 and 3. Comparing the $R^2$ estimates for temperature and Ref_OUT (a.k.a. the Li-820), we can conclude that temperature measurements are approximately equal or worse at predicting the behavior of the Li-840A.*

*Besides our analysis, we also forwarded this question to Li-cor's technical support team. The answer received was the following:*

*"Thank you for your email. If the heater is turned on, you should not see any significant drift in the response over the entire specified range of the LI-840A."*

2.) The range for the temperature calibration is too small and only a single test (at only one RH level) was conducted. Atmospheric temperatures outside the tropics frequently reach values below $10_0$C, which seems to have been the lowest temperature setting tested in section 3. Also, the chamber experiment holds the temperature stable for multiple hours. Is this really a realistic temperature profile for a drone flight?

*We agree with the reviewer, there are many teams that operate these sensors below 10 °C. However, the operational configuration of the Thunder Scientific 2500 chamber at the Mesonet Calibration Laboratory does not produce reliable test conditions below 10 °C. Therefore, this is a limitation of this study. Nevertheless, when compared with other studies in the literature, the presented manuscript does expand the results available. For example, the results from Martin et. al (2017) are limited to 16-24 °C and Arzoumanian et. al. (2019) are limited to 16-32 °C.*

*Regarding the time scales of the experiments, the goal of the long intervals was to study the general behavior of the sensors and create a comparable dataset to other results in the literature (with the expansion of the temperature and RH test ranges). This type of experiment is important because these sensors could have presented temperature time-response issues (i.e., lags) on the scales of tens of minutes. Therefore, the long dwell (Mesonet) and the short impulse (Bench) tests complement each other.*

*Regarding the number of tests, as mentioned above, two more runs of each test (temperature and relative humidity) were added to the study. One more bench test of each variable (T/RH) was also added (see experiment summary in the discussion supplement).*

3.) It is unclear how/if the lab bench setup described in figure 4 was able to provide a homogeneously heated air-stream to all instruments. It would be necessary have many more temperature sensors placed around the 2xK30 and the Li-COR to be sure they measured the same (temperature) air. Furthermore, the lab bench tests measured a response to a short-term temperature change within a few minutes, while the chamber test duration was over 6 hours with 2 hours time for instruments to equilibrate. How can those to experiments be compared? The low correlation seen in Figure 6 could well be related to the change in time scale of the experiment.

*We agree with the reviewer. Our ability to ensure temperature homogeneity during the experiment was not clear in the original manuscript. Besides the mixing fan depicted in figure 4 (original manuscript), the diaphragm pump intakes for the $CO_2$ sensors (one for the*

*reference sensor and one for the two K30 sensors) were placed immediately after three PT-100 bead thermistors (10 Hz sampling, 1 Hz time response, sold and calibrated by InterMet Systems), and three IST HYT-271 capacitive hygrometers (10 Hz sampling, 4-second RH time response, and 5-second Temperature time response). The temperature shown in the original manuscript is an average of the temperature of the 3 thermistors. The placement of the thermistors and hygrometers as well as the plumbing of the $CO_2$ sensors was added to the experiment schematics in the revised manuscript.*

Specific comments:

L1: Suggestion to mention that this study focusses on (lower-cost) NDIR sensors

*We agree with the reviewer. A comment was added to the revised manuscript indicating this study is particularly interested in low-cost, weight, size, and power systems. We also provided a statement about our understanding of the term low-cost (under US$300 for the total sensor package).*

L14: Please clarify: what does "mentioned measurement systems" refers to. Also please add a citation of studies that demonstrated the claim that no suitable measurement systems existed for local and regional scale work. Since the 2010s cavity ring down spectroscopy (CRDS) and integrated cavity output spectroscopy (ICOS) systems have been in regular use for atmospheric CO2 measurements and have allowed high-resolution and accurate measurements, even on mobile platforms (e.g. Chen at al. 2010, AMT - High- accuracy continuous airborne measurements of greenhouse gases (CO2 and CH4) using the cavity ring-down spectroscopy (CRDS) technique (copernicus.org)).

*The statement "mentioned measurement systems" refers to the "instrumented towers, satellites, and manned aircraft", mentioned in L10.*

*The manuscript does not claim there are "no suitable measurement systems for local and regional scales". What the statement in L14/15 indicates is that instrumented towers, satellites, and manned aircraft "... do not always support fast and comprehensive data collection near regional and local phenomena." For example, the Atmospheric Radiation Measurement - Southern Great Plains (ARM SGP) reference tower in Billings (OK) may not capture the nuances of a 40-minute traffic jam in Norman (OK) before an OU football game. Therefore, low-cost sensors may be an initial solution for initial exploratory studies. Such studies are also important because they help justify funding requisitions for more rigorous studies. Furthermore, outside developed countries (such as the US) the coverage of instrumented towers and manned aircraft is lower, and access to research funding is also lower. Thus increasing the need for low-cost tools to investigate local phenomena. As stated in L15/16, UAS-based measurement is a "... complementary in-situ observation tool for local atmospheric $CO_2$ profiles (Villa et al., 2016)."*

L85: You mention the need for instrument specific correction coefficients, but only decided to measure 2 instruments. How representative are two units? Martin et al. 2017 (AMT -

Evaluation and environmental correction of ambient CO2 measurements from a low-cost NDIR sensor (copernicus.org)) tested at least 6 units of the K30 series.

*We agree with the reviewer, two units may not be representative. As mentioned above we repeated the experiments with more units. A summary of the experiments and units tested is provided in table 1 of the discussion supplement. This table was also added to the revised manuscript.*

Figure 2. It is very difficult to distinguish the time series of the different instruments.

*We agree with the reviewer, the plot colors were updated in the revised manuscript and discussion supplement using more contrasting color palettes (similar to Martin et al. 2017). The colors were tested using this tool: https://projects.susielu.com/viz-palette.*

L144, Figure 8: A linear fit does not seem appropriate for the left panels. Did you consider a non-linear instrument response?

*We had not considered a non-linear fit because we had not found literature to support non-linear behavior. In fact, there is very little literature on the impact of humidity on low-cost NDIR sensors. Most studies found report the use of desiccants to eliminate humidity from the atmospheric samples. However, using desiccants in small UAS is not always possible (either due to cost, weight, or even fuselage access limitations), as desiccants need to be replaced frequently. Therefore, we understood the poor fit as an indication of low impact from the variable.*

*The new dataset for the repeated Mesonet Experiments (run 2 and 3) did not show the same behavior for the test sensors against relative humidity. In fact, figures 19 and 25 of the discussion supplement show a more linear behavior. Nonetheless, your suggestion is very interesting because the ~2 °C variation during both relative humidity experiments appears to be non-linear. At the time this response is being written we are analyzing the possibility of adding a complementary joint variation (T/RH) case to the revised manuscript. In this case, additional nonlinear analyses may be beneficial.*

L170: The temperature experiment, especially Figure4 clearly show that the concentration inside and outside the chamber can (and do) differ. Why do you consider the LI-840 on the outside as a reliable reference here, especially after the potential 'unknown external interference'?

*It is not clear to us what the reviewer is referring to in this comment. L170 in the manuscript offers a description of the Mesonet Pressure Dependence Experiment and Figure 4 illustrates the arrangement of the Bench Temperature and Relative Humidity experiments. The "unknown external interferences" mentioned in the manuscript are associated with the Mesonet Temperature and Relative Humidity experiments. Therefore, we will attempt to respond to the best of our understanding **assuming the question is about the experiment arrangement detailed in L170.***

*The Mesonet Pressure Experiment uses a different chamber than the Temp/RH experiments (as stated in L162/163). The pressure chamber (Cincinnati Sub-Zero Z16 with the custom gasket-based vacuum and compression system) uses two Thompson pumps to move air in and out of the chamber to raise and lower pressure. The air moved in and out of the chamber comes from the laboratory. Therefore, we placed the intake plumbing for the diaphragm pump of the Li-840A within 1 cm of the chamber's pump intake and exhaust to monitor the $CO_2$ of the air coming in and out of the chamber.*

- *Attempt to respond to the best of our understanding **assuming the question is about the experimental setup for the Mesonet Temperature and Relative Humidity experiments:***

*As detailed in L95 of the original manuscript, the temperature and humidity chamber (Thunder Scientific 2500) is not sealed. Therefore, there is exchange with air in the laboratory. Therefore our experiment design is dependent on a low variation of $CO_2$ in the laboratory to isolate the impact of T/RH on the K30. Under this assumption, if both reference sensors (IN and OUT) showed low variation and the test sensors showed high variation, we could use the dataset to study the impact of the test variable. These conditions were achieved when we repeated the experiments (see sections 4 and 5 of the attached discussion supplement).*

L214: This is a major concern: Can the results reported here be useful to other researchers, If the time delayed response to pressure changes is specific to the inlet and housing design?

*As stated by Gaynullin et. al. (2016), Martin et. al. (2017), and stated in the original manuscript, all the calibrations and coefficients shown are unit specific. Therefore, this study focused on demonstrating a low-cost repeatable method to determine these coefficients. Thus, **other researchers can use the methods shown in this study** to determine the time constants of each of their specific systems. Furthermore, as stated in L254 - L258 (original manuscript), the referred researchers will need to repeat the presented methods over time to recalibrate and re-evaluate their systems to "account for temporal drift and sensor decay".*

L240: The accuracy of the instruments has not been investigated at all. No gas standards from NOAA or NIST was used here, neither were calibrated reference instruments.

The authors did demonstrate that they can reproduce measurements within 2.5ppm for same air sampling of another optical instruments under certain conditions.

*We agree with the reviewer. Even though the comments in this document and the additional plots and tables provided in the discussion supplement indicate the reliability of the reference sensors, no gas standards were used. Therefore, we have added to the revised manuscript an explicit indication that our results are relative to the reference sensors.*

**Discussion Supplement for "The Impact of Environmental Variables on UAS-based Atmospheric Carbon Dioxide Measurements"**

Gustavo B. H. de Azevedo[1,2], Bill Doyle[2], Christopher A. Fiebrich[3], and David Schvartzman[1,4]

[1]Advanced Radar Research Center (ARRC) at The University of Oklahoma
[2]Center for Autonomous Sensing and Sampling (CASS) at The University of Oklahoma
[3]Oklahoma Mesonet, Oklahoma Climatological Survey at The University of Oklahoma
[4]University of Oklahoma School of Meteorology, Norman, Oklahoma

**Correspondence:** Gustavo B. H. de Azevedo (gust@ou.edu)

*Copyright statement.* Authors 2022

**1 Summary**

Following the reviewers' suggestions, we repeated the Mesonet temperature and relative humidity experiments under more controlled conditions, added more test sensors, and adjusted the result presentation. Therefore, this document is written in light of the following changes:

– The Li-820 sensor originally named CO2 Independent Sensor Outside is renamed Reference Sensor Out (Ref_OUT).

– The Li-840A sensor originally named CO2 Independent Sensor Inside is renamed Reference Sensor In (Ref_IN).

– The Senseair K30 sensors originally named CO2 Test Sensors 1 and 2 are renamed K30_##, where the first digit refers to its attached test system and the second digit is its identification. This way the first Senseair K30 sensor of test system 2 is named K30_21.

  – Test systems (labeled T_#) has its own pressure (two MS5611), temperature (three PT-100 bead thermistors), relative humidity (three IST HYT-271 hygrometer), and Carbon Dioxide (two Senseair K30).

– Environmental conditions during the experiments are labeled by chamber, reference sensor, and the test system they are associated with.

– The Mesonet temperature and relative humidity experiments now have a "run" identification (i.e., "Mesonet Run 1 Temperature", "Mesonet Run 2 Temperature", "Mesonet Run 1 Relative humidity", etc), where Run 1 is the data originally presented in the manuscript and Runs 2 and 3 are the repetition runs.

The following sections provide supporting material to explain the low CO2 values seen in the Mesonet Experiment (Run 1). The results of the correction application on the Mesonet T/RH Experiment (Run 1) dataset, followed by the results for the Mesonet T/RH Experiment (Run 2 and 3), and the adjusted Bench temperature and relative humidity Experiments. The document ends with the supporting plot for the Pressure time-response correction ("ideal signal").

Please note that even though the results for the correction application on the Mesonet T/RH Experiment (Run 1) dataset are presented in the attached "Discussion Supplement", they serve only as a comparison of the correction method. Following the reviewer's suggestion, the Mesonet T/RH Experiment (Run 1) dataset will be discarded. Only Mesonet T/RH Experiment (Run 2 and 3) will be analyzed in the revised manuscript.

**1.1 Experiments summary**

The following tables cross lists all the experiments performed and their sensors. The sensor intercomparison experiments are not listed on the table. They were preformed before the first run of the Mesonet experiments, before the second run of the Mesonet experiments, in between the second and third runs, and after the third run of the Mesonet experiments.

| Location | Name | Reference Sensors | Test Sensor |
|---|---|---|---|
| Mesonet | Run 1 Pressure | Ref_IN | K30_11, K30_12 |
| | Run 1 Temperature | Ref_IN, Ref_OUT | K30_11, K30_12 |
| | Run 1 Relative Humidity | Ref_IN, Ref_OUT | K30_11, K30_12 |
| | Run 2 Temperature | Ref_IN, Ref_OUT | K30_21, K30_22, K30_31, K30_32 |
| | Run 2 Relative Humidity | Ref_IN, Ref_OUT | K30_21, K30_22, K30_31, K30_32 |
| | Run 3 Temperature | Ref_IN, Ref_OUT | K30_21, K30_22, K30_31, K30_32, K30_13, K30_14 |
| | Run 3 Relative Humidity | Ref_IN, Ref_OUT | K30_21, K30_22, K30_31, K30_32, K30_13, K30_14 |
| Bench | Run 1 Pressure Correction | Ref_IN | K30_21, K30_22 |
| | Run 2 Pressure Correction | Ref_IN | K30_21, K30_22 |
| | Run 3 Pressure Correction | Ref_IN | K30_21, K30_22 |
| | Run 4 Pressure Correction | Ref_IN | K30_21, K30_22 |
| | Pressure Time-response Learn 1 | Ref_IN | K30_21, K30_22, K30_31, K30_32 |
| | Pressure Time-response Learn 2 | Ref_IN | K30_21, K30_22, K30_31, K30_32 |
| | Pressure Time-response Test 1 | Ref_IN | K30_21, K30_22, K30_31, K30_32 |
| | Pressure Time-response Test 2 | Ref_IN | K30_21, K30_22, K30_31, K30_32 |
| | Run 1 Temperature | Ref_IN | K30_21, K30_22 |
| | Run 1 Relative Humidity | Ref_IN | K30_21, K30_22 |
| | Run 2 Temperature | Ref_IN | K30_21, K30_22 |
| | Run 2 Relative Humidity | Ref_IN | K30_21, K30_22 |
| | Run 3 Relative Humidity | Ref_IN | K30_21, K30_22 |

[Figure]

**Figure 1.** Data showing the impact of the Thunder Scientific 2500 chamber on the reported $CO_2$ values. The chamber was turned on at 01:29:37 and turned off at 08:24:43.

[Figure]

**Figure 2.** Environmental conditions during the second run of the Mesonet Temperature and Relative Humidity experiments. The chamber was turned on at 01:29:37 and turned off at 08:24:43. During this period the pressure (all sensors) and internal temperature (reference sensors) does not vary greatly.

**Table 1.** Metrics for the experimental conditions for the complete second run of the Mesonet Experiments.

| Variable | Sensor | Minimum | Maximum | Average | Standard deviation |
|---|---|---|---|---|---|
| | Ref_OUT | 96910 | 97500 | 97262.68 | 105.94 |
| | Ref_IN | 96540 | 97160 | 96902.5 | 107.37 |
| Pressure [Pa] | T_2 | 97066.57 | 97669.28 | 97430.29 | 106.42 |
| | T_3 | 97039.22 | 97665.22 | 97427.47 | 106.54 |
| | Ref_OUT | 50.91 | 50.91 | 50.91 | **0**[*] |
| Temperature [°C] | Ref_IN | 51.17 | 51.28 | 51.23 | 0.01 |
| | Chamber | 10.23 | 40.26 | - | - |
| Relative Humidity [%] | Chamber | 2.06 | 88.03 | - | - |

[*]**Please see section 10 of this document for explanation about this zero deviation.**

[Figure]

**Figure 3.** Comparison of the reference offset strategies. On the left the all sensors are brought to the level of the Ref_IN sensor, on the right, to the Ref_OUT sensor.

**3 Mesonet Experiment 1**

**3.1 Reference Intercomparison**

[Figure]

**Figure 4.** Environmental conditions during the intercomparison before the first Mesonet experiments.

**Table 2.** Metrics for the experimental conditions for the intercomparison before first Mesonet experiments.

| Variable | Sensor | Minimum | Maximum | Average | Standard deviation |
|---|---|---|---|---|---|
| Pressure [Pa] | Ref_OUT | 97700 | 98000 | 97819.24 | 87.62 |
| | Ref_IN | 97380 | 97740 | 97526.47 | 88.92 |
| Temperature [°C] | Ref_OUT | 50.91 | 50.91 | 50.91 | **0**[*] |
| | Ref_IN | 51.17 | 51.25 | 51.22 | 0.02 |
| $H_2O$ [ppt] | Ref_IN | 3.38 | 4.3 | 3.57 | 0.18 |

[*]**Please see section 10 of this document for explanation about this zero deviation.**

[Figure]

**Figure 5.** Results for the intercomparison before the first Mesonet experiment. At this date, the two reference presented a constant $54.75$ ppm offset. After correcting this offset, the RMSE was $1.63 * 10^{-14}$ ppm.

**Table 3.** Metrics for the intercomparison before the first Mesonet experiments.

| Variable | Sensor | Minimum | Maximum | Average | Standard deviation |
|----------|--------|---------|---------|---------|--------------------|
| $CO_2$ [ppm] | Ref_OUT | 413.91 | 443.85 | 424.57 | 6.51 |
| | Ref_IN | 360.24 | 387.49 | 369.82 | 6.14 |
| | ARM SGP | 412.1 | 464.98 | 429.74 | 11.85 |

[Figure]

**Figure 6.** Experimental conditions during the first Mesonet Temperature run.

**Table 4.** Metrics for the experimental conditions during the first Mesonet Temperature run.

| Variable | Sensor | Minimum | Maximum | Average | Standard deviation |
|---|---|---|---|---|---|
| | Ref_OUT | 98860 | 99290 | 99170.04 | 128.85 |
| Pressure [Pa] | Ref_IN | 98460 | 98970 | 98825.68 | 134.19 |
| | T_1 | 99133.8 | 99565.99 | 99445.19 | 127.6 |
| | Ref_OUT | 50.91 | 50.91 | 50.91 | **0**[*] |
| Temperature [°C] | Ref_IN | 51.17 | 51.28 | 51.23 | 0.01 |
| | Chamber | 10.74 | 40.32 | - | - |
| Relative Humidity [%] | Chamber | 43 | 47.09 | 45.64 | 0.98 |

[*]**Please see section 10 of this document for explanation about this zero deviation.**

**Figure 7.** Results for the first Mesonet Temperature run.

**Table 5.** Carbon Dioxide metrics for the first Mesonet Temperature run.

| Variable | Sensor | Minimum | Maximum | Average | Standard deviation |
|----------|--------|---------|---------|---------|--------------------|
| | Ref_OUT | 425.48 | 581.55 | 484.34 | 45.08 |
| | Ref_IN | 435.11 | 479.91 | 456.2 | 12.7 |
| $CO_2$ [ppm] | K30_11 | 435.35 | 501.35 | 464.47 | 18.86 |
| | K30_12 | 442.05 | 509.72 | 468.27 | 21.15 |

**Correlation for the Mesonet Temperature 1**

**Figure 8.** Scatter plots for the first Mesonet Temperature run.

**Table 6.** Linear fit metrics for the first Mesonet Temperature run.

| Sensor | Predictor | Slope | Y-Intercept | $R^2$ | RMSE |
|--------|-----------|-------|-------------|-------|------|
| Ref_IN | Pressure | -0.08 | 8212.38 | 0.68 | 7.19 |
| | Temperature | 0.6 | 440.47 | 0.25 | 11 |
| | Relative Humidity | 7.42 | 117.68 | 0.33 | 10.43 |
| | Ref_OUT | 0.25 | 337.26 | 0.76 | 6.22 |
| K30_11 | Pressure | -0.14 | 13911.4 | 0.84 | 7.61 |
| | Temperature | 1.32 | 430.2 | 0.54 | 12.79 |
| | Relative Humidity | 9.19 | 44.89 | 0.23 | 16.57 |
| | Ref_IN | 1.41 | -178.06 | 0.9 | 5.98 |
| | Ref_OUT | 0.4 | 270.02 | 0.92 | 5.31 |
| K30_12 | Pressure | -0.16 | 15918.25 | 0.88 | 7.38 |
| | Temperature | 1.62 | 426.07 | 0.65 | 12.5 |
| | Relative Humidity | 8.46 | 82.31 | 0.15 | 19.46 |
| | Ref_IN | 1.5 | -218.19 | 0.82 | 9.08 |
| | Ref_OUT | 0.45 | 248.4 | 0.94 | 5.37 |

**3.3 Relative Humidity**

[Figure]

**Figure 9.** Experimental conditions during the first Mesonet Relative Humidity run.

**Table 7.** Metrics for the experimental conditions during the first Mesonet Relative Humidity run.

| Variable | Sensor | Minimum | Maximum | Average | Standard deviation |
|---|---|---|---|---|---|
| | Ref_OUT | 97950 | 98410 | 98264.73 | 121.25 |
| Pressure [Pa] | Ref_IN | 97590 | 98100 | 97930.66 | 121.5 |
| | T_1 | 98225.48 | 98674.64 | 98537.13 | 115.1 |
| | Ref_OUT | 50.91 | 50.91 | 50.91 | 0 |
| Temperature [°C] | Ref_IN | 51.17 | 51.28 | 51.23 | 0.01 |
| | Chamber | 25.47 | 25.5 | 25.48 | 0.01 |
| Relative Humidity [%] | Chamber | 14.97 | 95.48 | - | - |

[Figure]

**Figure 10.** Results for the first Mesonet Relative Humidity run.

**Table 8.** Carbon Dioxide metrics for the first Mesonet Relative Humidity run.

| Variable | Sensor | Minimum | Maximum | Average | Standard deviation |
|---|---|---|---|---|---|
| | Ref_OUT | 428.93 | 593.21 | 483 | 52.16 |
| | Ref_IN | 447.85 | 510.86 | 465.9 | 19.74 |
| $CO_2$ [ppm] | K30_11 | 449.41 | 532.41 | 476.07 | 24.97 |
| | K30_12 | 449.43 | 532.8 | 476.58 | 24.91 |

**Correlation for the Mesonet Relative Humidity 1**

[Figure]

**Figure 11.** Scatter plots for the first Mesonet Relative Humidity run.

**Table 9.** Linear fit metrics for the first Mesonet Relative Humidity run.

| Sensor | Predictor | Slope | Y-Intercept | $R^2$ | RMSE |
|--------|-----------|-------|-------------|-------|------|
| | Pressure | 0.14 | -13756.32 | 0.44 | 18.64 |
| | Temperature | 1747.35 | -44046.02 | 0.19 | 22.52 |
| K30_11 | Relative Humidity | 0.79 | 427.22 | 0.69 | 13.91 |
| | Ref_IN | 1.25 | -107.72 | 0.98 | 3.48 |
| | Ref_OUT | 0.47 | 249.43 | 0.96 | 4.96 |
| | Pressure | 0.15 | -14021.01 | 0.46 | 18.27 |
| | Temperature | 1765.67 | -44512.15 | 0.19 | 22.4 |
| K30_12 | Relative Humidity | 0.8 | 427.3 | 0.71 | 13.52 |
| | Ref_IN | 1.25 | -104.56 | 0.98 | 3.81 |
| | Ref_OUT | 0.47 | 250.86 | 0.96 | 5.13 |

**4 Mesonet Experiment 2**

**4.1 Reference Sensors**

[Figure]

**Figure 12.** Experimental conditions for the intercomparison before second Mesonet experiments.

**Table 10.** Metrics for the experimental conditions for the intercomparison before second Mesonet experiments.

| Variable | Sensor | Minimum | Maximum | Average | Standard deviation |
|---|---|---|---|---|---|
| | Ref_OUT | 97220 | 97320 | 97261.68 | 23.66 |
| | Ref_IN | 96830 | 96980 | 96904.07 | 26.97 |
| Pressure [Pa] | T_1 | 97395.68 | 97470.13 | 97427.53 | 20.73 |
| | T_2 | 97393.41 | 97483.26 | 97430.47 | 25.02 |
| Temperature [°C] | Ref_OUT | 50.91 | 50.91 | 50.91 | 0 |
| | Ref_IN | 51.2 | 51.28 | 51.23 | 0.01 |
| $H_2O$ [ppt] | Ref_IN | 13.57 | 13.75 | 13.65 | 0.03 |

[Figure]

**Figure 13.** Results for the intercomparison before the second Mesonet experiment. At this date, the two reference presented a constant 18.23 ppm offset. After correcting this offset, the RMSE was $1.53 * 10^{-14}$ ppm.

**Table 11.** Metrics for the intercomparison before the second Mesonet experiments.

| Variable | Sensor | Minimum | Maximum | Average | Standard deviation |
|---|---|---|---|---|---|
| | Ref_OUT | 465.75 | 520.45 | 490.28 | 13.02 |
| | Ref_IN | 484.46 | 538.11 | 508.51 | 13.17 |
| | ARM SGP | 413.32 | 420.2 | 417.05 | 1.42 |
| $CO_2$ [ppm] | K30_21 | 425.5 | 481.5 | 449.68 | 13.55 |
| | K30_22 | 482 | 539 | 508.29 | 13.93 |
| | K30_31 | 529 | 585 | 552.27 | 13.48 |
| | K30_32 | 441 | 497 | 467.25 | 14.28 |

[Figure]

**Figure 14.** Experimental conditions during the second Mesonet Temperature run.

**Table 12.** Metrics for the experimental conditions during the second Mesonet Temperature run.

| Variable | Sensor | Minimum | Maximum | Average | Standard deviation |
|---|---|---|---|---|---|
| | Ref_OUT | 97210 | 97400 | 97304.12 | 47.69 |
| | Ref_IN | 96830 | 97050 | 96944.33 | 45.83 |
| Pressure [Pa] | T_2 | 97385.18 | 97565.27 | 97470.79 | 48.26 |
| | T_3 | 97385.21 | 97563.27 | 97468.38 | 47.66 |
| | Ref_OUT | 50.91 | 50.91 | 50.91 | 0 |
| Temperature [°C] | Ref_IN | 51.17 | 51.28 | 51.23 | 0.01 |
| | Chamber | 10.28 | 40.26 | - | - |
| Relative Humidity [%] | Chamber | 43.48 | 47.81 | 45.17 | 1.23 |

[Figure]

**Figure 15.** Results for the second Mesonet Temperature run.

**Table 13.** Carbon Dioxide metrics for the second Mesonet Temperature run.

| Variable | Sensor | Minimum | Maximum | Average | Standard deviation |
|---|---|---|---|---|---|
| | Ref_OUT | 430.39 | 444.43 | 434.57 | 3.68 |
| | Ref_IN | 434.32 | 446.82 | 438.07 | 2.47 |
| | K30_21 | 434.62 | 477.32 | 454.36 | 11.88 |
| $CO_2$ [ppm] | K30_22 | 436.42 | 477.51 | 453.11 | 11.29 |
| | K30_31 | 427.33 | 448.05 | 439.02 | 5.08 |
| | K30_32 | 435.23 | 471.76 | 451.68 | 10.29 |

**Correlation for the Mesonet Temperature 2**

[Figure]

**Figure 16.** Scatter plots for the second Mesonet Temperature run.

**Table 14.** Linear fit metrics for the second Mesonet Temperature run.

| Sensor | Predictor | Slope | Y-Intercept | $R^2$ | RMSE |
|---|---|---|---|---|---|
| Ref_IN | Pressure | -0.02 | 2538.79 | 0.15 | 2.26 |
| | Temperature | -0.17 | 442.25 | 0.42 | 1.87 |
| | Relative Humidity | 0.68 | 407.18 | 0.12 | 2.31 |
| | Ref_OUT | 0.44 | 245.07 | 0.44 | 1.83 |
| K30_21 | Pressure | 0.2 | -18931.25 | 0.65 | 6.99 |
| | Temperature | 1.25 | 423.91 | 0.96 | 2.35 |
| | Relative Humidity | -0.81 | 490.97 | 0.01 | 11.83 |
| | Ref_IN | -2.47 | 1536.34 | 0.26 | 10.21 |
| | Ref_OUT | -2.09 | 1363.93 | 0.42 | 9.03 |
| K30_22 | Pressure | 0.18 | -17025.1 | 0.59 | 7.24 |
| | Temperature | 0.93 | 430.45 | 0.59 | 7.23 |
| | Relative Humidity | 4.22 | 262.52 | 0.21 | 10.03 |
| | Ref_IN | -1.05 | 912.85 | 0.05 | 10.98 |
| | Ref_OUT | -1.82 | 1242.14 | 0.35 | 9.08 |
| K30_31 | Pressure | 0.01 | -235.81 | 0 | 5.05 |
| | Temperature | 0.1 | 436.62 | 0.03 | 4.98 |
| | Relative Humidity | -3.61 | 602.12 | 0.76 | 2.47 |
| | Ref_IN | -0.15 | 503.29 | 0.01 | 5.05 |
| | Ref_OUT | 0.24 | 335.75 | 0.03 | 4.99 |
| K30_32 | Pressure | 0.18 | -16989.37 | 0.69 | 5.75 |
| | Temperature | 1.08 | 425.42 | 0.95 | 2.24 |
| | Relative Humidity | -0.48 | 473.53 | 0 | 10.27 |
| | Ref_IN | -2.08 | 1361.23 | 0.25 | 8.93 |
| | Ref_OUT | -1.82 | 1243.47 | 0.43 | 7.79 |

**4.3 Relative Humidity**

[Figure]

**Figure 17.** Experimental conditions during the second Mesonet Relative Humidity run.

**Table 15.** Metrics for the experimental conditions during the second Mesonet Relative Humidity run.

| Variable | Sensor | Minimum | Maximum | Average | Standard deviation |
|---|---|---|---|---|---|
| Pressure [Pa] | Ref_OUT | 97100 | 97500 | 97342.03 | 111.34 |
| | Ref_IN | 96720 | 97160 | 96980.96 | 111 |
| | T_2 | 97274.8 | 97669.28 | 97511.81 | 112.58 |
| | T_3 | 97270.3 | 97665.22 | 97508.13 | 112.64 |
| Temperature [°C] | Ref_OUT | 50.91 | 50.91 | 50.91 | 0 |
| | Ref_IN | 51.2 | 51.28 | 51.23 | 0.01 |
| | Chamber | 21.95 | 26.76 | 26.2 | 0.95 |
| Relative Humidity [%] | Chamber | 14.87 | 88.03 | - | - |

[Figure]

**Figure 18.** Results for the second Mesonet Relative Humidity run.

**Table 16.** Carbon Dioxide metrics for the second Mesonet Relative Humidity run.

| Variable | Sensor | Minimum | Maximum | Average | Standard deviation |
|---|---|---|---|---|---|
| | Ref_OUT | 432.36 | 434.37 | 433.36 | 0.57 |
| | Ref_IN | 427.27 | 439.11 | 430.78 | 2.54 |
| | K30_21 | 429.1 | 443.48 | 438.27 | 3.27 |
| $CO_2$ [ppm] | K30_22 | 429.23 | 445.32 | 439.59 | 3.75 |
| | K30_31 | 421.89 | 439.33 | 434.08 | 3.09 |
| | K30_32 | 429.27 | 451.75 | 444.42 | 6.1 |

**Correlation for the Mesonet Relative Humidity 2**

**Figure 19.** Scatter plots for the second Mesonet Relative Humidity run.

**Table 17.** Linear fit metrics for the second Mesonet Relative Humidity run.

| Sensor | Predictor | Slope | Y-Intercept | $R^2$ | RMSE |
|---|---|---|---|---|---|
| | Pressure | -0.01 | 1489.63 | 0.14 | 3.02 |
| | Temperature | 2.65 | 368.72 | 0.6 | 2.06 |
| K30_21 | Relative Humidity | 0.1 | 432.97 | 0.65 | 1.92 |
| | Ref_IN | -0.25 | 544.31 | 0.04 | 3.19 |
| | Ref_OUT | 3.42 | -1043.98 | 0.35 | 2.62 |
| | Pressure | -0.02 | 2062.65 | 0.25 | 3.24 |
| | Temperature | 3.22 | 355.25 | 0.67 | 2.16 |
| K30_22 | Relative Humidity | 0.11 | 433.84 | 0.58 | 2.43 |
| | Ref_IN | -0.18 | 519.04 | 0.02 | 3.71 |
| | Ref_OUT | 4.7 | -1597.44 | 0.5 | 2.64 |
| | Pressure | 0 | 128.65 | 0.01 | 3.05 |
| | Temperature | 0.84 | 412.08 | 0.07 | 2.96 |
| K30_31 | Relative Humidity | 0.08 | 429.5 | 0.55 | 2.07 |
| | Ref_IN | -0.36 | 589.56 | 0.09 | 2.93 |
| | Ref_OUT | 0.23 | 335.68 | 0 | 3.07 |
| | Pressure | -0.02 | 2753.61 | 0.19 | 5.48 |
| | Temperature | 5.39 | 303.33 | 0.7 | 3.32 |
| K30_32 | Relative Humidity | 0.18 | 434.63 | 0.63 | 3.69 |
| | Ref_IN | -0.52 | 668.32 | 0.05 | 5.95 |
| | Ref_OUT | 7.6 | -2850.09 | 0.5 | 4.32 |

**5 Mesonet Experiment 3**

**5.1 Temperature**

[Figure]

**Figure 20.** Experimental conditions during the third Mesonet Temperature run.

**Table 18.** Metrics for the experimental conditions during the third Mesonet Temperature run.

| Variable | Sensor | Minimum | Maximum | Average | Standard deviation |
|---|---|---|---|---|---|
| | Ref_OUT | 96730 | 97170 | 96886.92 | 133.07 |
| | Ref_IN | 96350 | 96820 | 96517.46 | 131.51 |
| Pressure [Pa] | T_1 | 96893.91 | 97325.16 | 97034.12 | 135.93 |
| | T_2 | 96909.97 | 97343.58 | 97050.6 | 134.92 |
| | T_3 | 96895.46 | 97326.49 | 97034.11 | 134.52 |
| | Ref_OUT | 50.91 | 50.91 | 50.91 | 0 |
| Temperature [°C] | Ref_IN | 51.17 | 51.28 | 51.23 | 0.01 |
| | Chamber | 10.44 | 40.98 | - | - |
| Relative Humidity [%] | Chamber | 43.78 | 49.27 | 45.56 | 1.05 |

[Figure]

**Figure 21.** Results for the third Mesonet Temperature run.

**Table 19.** Carbon Dioxide metrics for the third Mesonet Temperature run.

| Variable | Sensor | Minimum | Maximum | Average | Standard deviation |
|---|---|---|---|---|---|
| | Ref_OUT | 436.43 | 465.98 | 449.53 | 8.36 |
| | Ref_IN | 451.11 | 467.77 | 456.67 | 4.03 |
| | K30_13 | 454.01 | 486.43 | 468.41 | 8.42 |
| | K30_14 | 452.86 | 485.25 | 467.83 | 7.9 |
| $CO_2$ [ppm] | K30_21 | 452.92 | 499.52 | 475.51 | 12.3 |
| | K30_22 | 446.99 | 493.18 | 471.16 | 11.07 |
| | K30_31 | 446.81 | 469.85 | 458.34 | 4.97 |
| | K30_32 | 452.9 | 492.42 | 472.37 | 10.47 |

**Correlation for the Mesonet Temperature 3**

[Figure]

**Figure 22.** Scatter plots for the third Mesonet Temperature run.

**Table 20.** Linear fit metrics for the third Mesonet Temperature run.

| Sensor | Predictor | Slope | Y-Intercept | $R^2$ | RMSE |
|---|---|---|---|---|---|
| Ref_IN | Pressure | -0.02 | 2078.44 | 0.3 | 3.37 |
| | Temperature | -0.3 | 464.16 | 0.51 | 2.81 |
| | Relative Humidity | 0.09 | 452.78 | 0 | 4.02 |
| | Ref_OUT | 0.32 | 313.36 | 0.44 | 3.01 |
| K30_13 | Pressure | 0.04 | -3669.06 | 0.47 | 6.1 |
| | Temperature | 0.78 | 448.64 | 0.81 | 3.64 |
| | Relative Humidity | -1.81 | 550.89 | 0.05 | 8.2 |
| | Ref_IN | -0.82 | 842.92 | 0.15 | 7.74 |
| | Ref_OUT | -0.21 | 562.65 | 0.04 | 8.23 |
| K30_14 | Pressure | 0.04 | -3251.44 | 0.44 | 5.93 |
| | Temperature | 0.7 | 450.08 | 0.75 | 3.98 |
| | Relative Humidity | -1.03 | 514.56 | 0.02 | 7.82 |
| | Ref_IN | -0.67 | 772.65 | 0.12 | 7.42 |
| | Ref_OUT | -0.24 | 574.91 | 0.06 | 7.64 |
| K30_21 | Pressure | 0.06 | -5286.77 | 0.42 | 9.33 |
| | Temperature | 1.21 | 444.97 | 0.91 | 3.71 |
| | Relative Humidity | -3.32 | 626.54 | 0.08 | 11.8 |
| | Ref_IN | -1.53 | 1175.94 | 0.25 | 10.64 |
| | Ref_OUT | -0.36 | 637.62 | 0.06 | 11.92 |
| K30_22 | Pressure | 0.05 | -4347.35 | 0.37 | 8.81 |
| | Temperature | 1.05 | 444.56 | 0.85 | 4.25 |
| | Relative Humidity | -2.07 | 565.27 | 0.04 | 10.85 |
| | Ref_IN | -1.37 | 1094.61 | 0.25 | 9.6 |
| | Ref_OUT | -0.33 | 619.19 | 0.06 | 10.72 |
| K30_31 | Pressure | 0 | 321.94 | 0 | 4.96 |
| | Temperature | -0.1 | 460.86 | 0.04 | 4.87 |
| | Relative Humidity | -2.6 | 576.97 | 0.3 | 4.15 |
| | Ref_IN | 0.76 | 109.91 | 0.38 | 3.9 |
| | Ref_OUT | 0.44 | 262.08 | 0.54 | 3.36 |
| K30_32 | Pressure | 0.06 | -4964.63 | 0.52 | 7.27 |
| | Temperature | 1.01 | 446.77 | 0.88 | 3.59 |
| | Relative Humidity | -4.11 | 659.47 | 0.17 | 9.55 |
| | Ref_IN | -1.23 | 1032.26 | 0.22 | 9.24 |
| | Ref_OUT | -0.16 | 546.35 | 0.02 | 10.38 |

**5.2 Relative Humidity**

[Figure]

**Figure 23.** Experimental conditions during the third Mesonet Relative Humidity run.

**Table 21.** Metrics for the experimental conditions during the third Mesonet Relative Humidity run.

| Variable | Sensor | Minimum | Maximum | Average | Standard deviation |
|---|---|---|---|---|---|
| | Ref_OUT | 96660 | 97020 | 96787.52 | 101.93 |
| | Ref_IN | 96250 | 96680 | 96413.49 | 103.39 |
| Pressure [Pa] | T_1 | 96805.94 | 97161.59 | 96924.18 | 98.27 |
| | T_2 | 96821.58 | 96951.79 | 96908.85 | 36.22 |
| | T_3 | 96803.85 | 97169.14 | 96932.27 | 100.46 |
| | Ref_OUT | 50.91 | 50.91 | 50.91 | 0 |
| Temperature [°C] | Ref_IN | 51.17 | 51.28 | 51.23 | 0.01 |
| | Chamber | 25.83 | 27.61 | 27.08 | 0.33 |
| Relative Humidity [%] | Chamber | 15.1 | 85.4 | - | - |

[Figure]

**Figure 24.** Results for the third Mesonet Relative Humidity run.

**Table 22.** Carbon Dioxide metrics for the third Mesonet Relative Humidity run.

| Variable | Sensor | Minimum | Maximum | Average | Standard deviation |
|---|---|---|---|---|---|
| | Ref_OUT | 426.49 | 431.72 | 428.72 | 1.45 |
| | Ref_IN | 424.45 | 437.28 | 428.66 | 2.46 |
| | K30_13 | 429.9 | 442.38 | 436.38 | 3.13 |
| | K30_14 | 429.07 | 447.24 | 439.24 | 4.75 |
| $CO_2$ [ppm] | K30_21 | 429.52 | 441.41 | 436.28 | 2.66 |
| | K30_22 | 429.48 | 447.17 | 439.36 | 5.56 |
| | K30_31 | 428.83 | 442.83 | 436.88 | 3.59 |
| | K30_32 | 429.72 | 440.2 | 435.62 | 2.63 |

**Correlation for the Mesonet Relative Humidity 3**

[Figure]

**Figure 25.** Scatter plots for the third Mesonet Relative Humidity run.

**Table 23.** Linear fit metrics for the third Mesonet Relative Humidity run.

| Sensor | Predictor | Slope | Y-Intercept | $R^2$ | RMSE |
|---|---|---|---|---|---|
| | Pressure | 0.01 | -92.64 | 0.03 | 3.08 |
| | Temperature | 6.31 | 265.53 | 0.45 | 2.33 |
| K30_13 | Relative Humidity | 0.1 | 430.8 | 0.75 | 1.55 |
| | Ref_IN | -0.75 | 756.54 | 0.34 | 2.53 |
| | Ref_OUT | -1.27 | 981.63 | 0.35 | 2.52 |
| | Pressure | 0.01 | -236.3 | 0.02 | 4.68 |
| | Temperature | 9.54 | 180.9 | 0.44 | 3.53 |
| K30_14 | Relative Humidity | 0.16 | 430.51 | 0.8 | 2.1 |
| | Ref_IN | -1.2 | 955.06 | 0.39 | 3.71 |
| | Ref_OUT | -1.89 | 1249.99 | 0.33 | 3.86 |
| | Pressure | 0.06 | -5512.96 | 0.7 | 1.45 |
| | Temperature | 7.11 | 243.81 | 0.79 | 1.23 |
| K30_21 | Relative Humidity | 0.06 | 432.81 | 0.41 | 2.04 |
| | Ref_IN | -0.31 | 569.21 | 0.08 | 2.54 |
| | Ref_OUT | -1.59 | 1117.18 | 0.75 | 1.32 |
| | Pressure | 0.1 | -9486.42 | 0.45 | 4.13 |
| | Temperature | 12.72 | 94.92 | 0.57 | 3.62 |
| K30_22 | Relative Humidity | 0.08 | 435.27 | 0.13 | 5.18 |
| | Ref_IN | -0.39 | 604.51 | 0.03 | 5.46 |
| | Ref_OUT | -3.61 | 1988.63 | 0.89 | 1.84 |
| | Pressure | 0 | 18.48 | 0.01 | 3.55 |
| | Temperature | 6.75 | 254.08 | 0.39 | 2.8 |
| K30_31 | Relative Humidity | 0.12 | 430.45 | 0.76 | 1.75 |
| | Ref_IN | -0.73 | 748.53 | 0.25 | 3.11 |
| | Ref_OUT | -1.28 | 983.93 | 0.27 | 3.06 |
| | Pressure | 0.01 | -242.33 | 0.07 | 2.52 |
| | Temperature | 5.56 | 285.04 | 0.49 | 1.86 |
| K30_32 | Relative Humidity | 0.08 | 431.25 | 0.66 | 1.52 |
| | Ref_IN | -0.44 | 624.66 | 0.17 | 2.38 |
| | Ref_OUT | -1.14 | 923.1 | 0.4 | 2.03 |

[Figure]

**Figure 26.** Experimental conditions for the intercomparison after third Mesonet experiments.

**Table 24.** Metrics for the experimental conditions for the intercomparison after third Mesonet experiments.

| Variable | Sensor | Minimum | Maximum | Average | Standard deviation |
|---|---|---|---|---|---|
| Pressure [Pa] | Ref_OUT | 96030 | 96120 | 96075.97 | 22.5 |
| | Ref_IN | 95850 | 95980 | 95921.67 | 24.51 |
| Temperature [°C] | Ref_OUT | 50.91 | 50.91 | 50.91 | 0 |
| | Ref_IN | 51.17 | 51.25 | 51.23 | 0.01 |
| $H_2O$ [ppt] | Ref_IN | 18.03 | 20.88 | 18.87 | 0.61 |

[Figure]

**Figure 27.** Results for the intercomparison after the third Mesonet experiment. At this date, the two reference presented a constant 17.24 ppm offset. After correcting this offset, the RMSE was $1.12 * 10^{-14}$ ppm.

**Table 25.** Metrics for the intercomparison after the third Mesonet experiments.

| Variable | Sensor | Minimum | Maximum | Average | Standard deviation |
|---|---|---|---|---|---|
| | Ref_OUT | 398.62 | 437.86 | 413.05 | 8.74 |
| $CO_2$ [ppm] | Ref_IN | 417.4 | 453.29 | 430.29 | 8.27 |
| | ARM SGP | 415.3 | 420.81 | 418.31 | 1.15 |

**6 Benchtop Experiments 1**

**6.1 Temperature**

[Figure]

**Figure 28.** Experimental conditions during the first Bench Temperature run.

**Table 26.** Metrics for the experimental conditions during the first Bench Temperature run.

| Variable | Sensor | Minimum | Maximum | Average | Standard deviation |
|---|---|---|---|---|---|
| Pressure [Pa] | Ref_IN | 97300 | 97370 | 97342.1 | 13.57 |
| | T_2 | 97617.75 | 97634.8 | 97626.68 | 2.82 |
| Temperature [°C] | Ref_IN | 51.17 | 51.25 | 51.23 | 0.01 |
| | T_2 | 20.75 | 53.41 | - | - |
| Relative Humidity [%] | T_2 | 10.62 | 50.09 | 28.31 | 11.43 |

[Figure]

**Figure 29.** Results for the first Bench Temperature run.

**Table 27.** Carbon Dioxide metrics for the first Bench Temperature run.

| Variable | Sensor | Minimum | Maximum | Average | Standard deviation |
|---|---|---|---|---|---|
| | Ref_IN | 459.04 | 545.15 | 492.49 | 23.76 |
| $CO_2$ [ppm] | K30_21 | 497.84 | 559.03 | 525.9 | 17.94 |
| | K30_22 | 483.39 | 558.39 | 518.05 | 23.47 |

**Correlation for the Bench Temperature 1**

[Figure]

**Figure 30.** Scatter plots for the first Bench Temperature run.

**Table 28.** Linear fit metrics for the first Bench Temperature run.

| Sensor | Predictor | Slope | Y-Intercept | $R^2$ | RMSE |
|--------|-----------|-------|-------------|-------|------|
| K30_21 | Pressure | -3.01 | 294698.42 | 0.22 | 15.79 |
| | Temperature | 0.22 | 518.82 | 0.01 | 17.82 |
| | Relative Humidity | 0.28 | 517.94 | 0.03 | 17.63 |
| | Ref_IN | 0.36 | 346.62 | 0.23 | 15.7 |
| K30_22 | Pressure | -4.04 | 394541.7 | 0.23 | 20.51 |
| | Temperature | 0.1 | 514.98 | 0 | 23.43 |
| | Relative Humidity | 0.54 | 502.84 | 0.07 | 22.63 |
| | Ref_IN | 0.53 | 256.9 | 0.29 | 19.78 |

**6.2 Relative Humidity**

[Figure]

**Figure 31.** Experimental conditions during the first Bench Relative Humidity run.

**Table 29.** Metrics for the experimental conditions during the first Bench Relative Humidity run.

| Variable | Sensor | Minimum | Maximum | Average | Standard deviation |
|---|---|---|---|---|---|
| Pressure [Pa] | Ref_IN | 97320 | 97390 | 97353 | 13.16 |
| | T_2 | 97627.66 | 97659.9 | 97640.33 | 5.71 |
| Temperature [°C] | Ref_IN | 51.17 | 51.25 | 51.23 | 0.01 |
| | T_2 | 20.9 | 22.79 | 21.81 | 0.58 |
| Relative Humidity [%] | T_2 | 48.27 | 74.04 | - | - |

[Figure]

**Figure 32.** Results for the first Bench Relative Humidity run.

**Table 30.** Carbon Dioxide metrics for the first Bench Relative Humidity run.

| Variable | Sensor | Minimum | Maximum | Average | Standard deviation |
|---|---|---|---|---|---|
| | Ref_IN | 415.71 | 499.36 | 443.18 | 16.68 |
| CO$_2$ [ppm] | K30_21 | 429.07 | 440.17 | 434 | 2.74 |
| | K30_22 | 427.3 | 438.15 | 433.97 | 2.85 |

**Correlation for the Bench Relative Humidity 1**

[Figure]

**Figure 33.** Scatter plots for the first Bench Relative Humidity run.

**Table 31.** Linear fit metrics for the first Bench Relative Humidity run.

| Sensor | Predictor | Slope | Y-Intercept | $R^2$ | RMSE |
|---|---|---|---|---|---|
| K30_21 | Pressure | -0.28 | 27598.92 | 0.34 | 2.23 |
| | Temperature | -2.36 | 485.44 | 0.25 | 2.37 |
| | Relative Humidity | -0.04 | 436.72 | 0.02 | 2.71 |
| | Ref_IN | 0.06 | 406.28 | 0.14 | 2.54 |
| K30_22 | Pressure | -0.29 | 28917.54 | 0.34 | 2.31 |
| | Temperature | -2.99 | 499.14 | 0.37 | 2.26 |
| | Relative Humidity | -0.02 | 435.4 | 0.01 | 2.84 |
| | Ref_IN | 0.04 | 418.13 | 0.04 | 2.79 |

 ## 7.1 Temperature

[Figure]

**Figure 34.** Experimental conditions during the second Bench Temperature run.

**Table 32.** Metrics for the experimental conditions during the second Bench Temperature run.

| Variable | Sensor | Minimum | Maximum | Average | Standard deviation |
|---|---|---|---|---|---|
| Pressure [Pa] | Ref_IN | 97440 | 97530 | 97482.13 | 16.39 |
| | T_2 | 97754.06 | 97773.67 | 97762.57 | 5.1 |
| Temperature [°C] | Ref_IN | 51.17 | 51.25 | 51.23 | 0.01 |
| | T_2 | 21.51 | 39.81 | - | - |
| Relative Humidity [%] | T_2 | 19.61 | 41.42 | 33.56 | 6.33 |

[Figure]

**Figure 35.** Results for the second Bench Temperature run.

**Table 33.** Carbon Dioxide metrics for the second Bench Temperature run.

| Variable | Sensor | Minimum | Maximum | Average | Standard deviation |
|---|---|---|---|---|---|
| | Ref_IN | 406.13 | 427.64 | 413.68 | 6.41 |
| $CO_2$ [ppm] | K30_21 | 403.99 | 423.49 | 412.55 | 6.05 |
| | K30_22 | 397.77 | 416.77 | 407.28 | 4.96 |

**Correlation for the Bench Temperature 2**

[Figure]

**Figure 36.** Scatter plots for the second Bench Temperature run.

**Table 34.** Linear fit metrics for the second Bench Temperature run.

| Sensor | Predictor | Slope | Y-Intercept | $R^2$ | RMSE |
|---|---|---|---|---|---|
| K30_21 | Pressure | -0.62 | 61137.65 | 0.27 | 5.15 |
| | Temperature | -0.03 | 413.39 | 0 | 6.04 |
| | Relative Humidity | -0.18 | 418.56 | 0.04 | 5.94 |
| | Ref_IN | 0.73 | 112.51 | 0.59 | 3.87 |
| K30_22 | Pressure | -0.05 | 5327.96 | 0 | 4.95 |
| | Temperature | -0.23 | 413.29 | 0.04 | 4.86 |
| | Relative Humidity | 0.02 | 406.54 | 0 | 4.96 |
| | Ref_IN | 0.41 | 238.09 | 0.28 | 4.21 |

**7.2 Relative Humidity**

[Figure]

**Figure 37.** Experimental conditions during the second Bench Relative Humidity run.

**Table 35.** Metrics for the experimental conditions during the second Bench Relative Humidity run.

| Variable | Sensor | Minimum | Maximum | Average | Standard deviation |
|---|---|---|---|---|---|
| Pressure [Pa] | Ref_IN | 97400 | 97470 | 97439.71 | 15.51 |
| | T_2 | 97707.94 | 97724.89 | 97717.96 | 4.32 |
| Temperature [°C] | Ref_IN | 51.17 | 51.25 | 51.22 | 0.01 |
| | T_2 | 19.8 | 22.3 | 21.44 | 0.69 |
| Relative Humidity [%] | T_2 | 38.9 | 57.85 | 45.78 | 5.99 |

[Figure]

**Figure 38.** Results for the second Bench Relative Humidity run.

**Table 36.** Carbon Dioxide metrics for the second Bench Relative Humidity run.

| Variable | Sensor | Minimum | Maximum | Average | Standard deviation |
|----------|--------|---------|---------|---------|--------------------|
| | Ref_IN | 400.31 | 403.6 | 401.63 | 1.04 |
| $CO_2$ [ppm] | K30_21 | 399.36 | 402.36 | 400.68 | 0.79 |
| | K30_22 | 400.24 | 403.09 | 401.65 | 0.75 |

**Correlation for the Bench Relative Humidity 2**

[Figure]

**Figure 39.** Scatter plots for the second Bench Relative Humidity run.

**Table 37.** Linear fit metrics for the second Bench Relative Humidity run.

| Sensor | Predictor | Slope | Y-Intercept | $R^2$ | RMSE |
|--------|-----------|-------|-------------|-------|------|
| K30_21 | Pressure | -0.18 | 18403.68 | 0.59 | 0.67 |
| | Temperature | -0.57 | 413.78 | 0.14 | 0.96 |
| | Relative Humidity | 0.06 | 398.89 | 0.12 | 0.97 |
| | Ref_IN | 1.14 | -57.65 | 0.68 | 0.59 |
| K30_22 | Pressure | -0.04 | 4655.61 | 0.06 | 0.76 |
| | Temperature | -0.23 | 405.59 | 0.04 | 0.77 |
| | Relative Humidity | 0.03 | 399.45 | 0.04 | 0.77 |
| | Ref_IN | 0.55 | 180.63 | 0.27 | 0.67 |

**8 Benchtop Experiments 3**

**8.1 Relative Humidity**

[Figure]

**Figure 40.** Experimental conditions during the third Bench Relative Humidity run.

**Table 38.** Metrics for the experimental conditions during the third Bench Relative Humidity run.

| Variable | Sensor | Minimum | Maximum | Average | Standard deviation |
|---|---|---|---|---|---|
| Pressure [Pa] | Ref_IN | 97220 | 97290 | 97259.79 | 13.42 |
|  | T_2 | 97526.79 | 97542.3 | 97533.86 | 3.63 |
| Temperature [°C] | Ref_IN | 51.17 | 51.25 | 51.23 | 0.01 |
|  | T_2 | 22.38 | 25.39 | 23.82 | 1.02 |
| Relative Humidity [%] | T_2 | 38.16 | 80.56 | - | - |

[Figure]

**Figure 41.** Results for the third Bench Relative Humidity run.

**Table 39.** Carbon Dioxide metrics for the third Bench Relative Humidity run.

| Variable | Sensor | Minimum | Maximum | Average | Standard deviation |
|----------|--------|---------|---------|---------|--------------------|
|          | Ref_IN | 394.42  | 403.21  | 397.29  | 2.59               |
| $CO_2$ [ppm] | K30_21 | 393.69 | 400.28 | 396.63 | 1.79          |
|          | K30_22 | 393.31  | 400.31  | 395.75  | 1.55               |

**Correlation for the Bench Relative Humidity 3**

[Figure]

**Figure 42.** Scatter plots for the third Bench Relative Humidity run.

**Table 40.** Linear fit metrics for the third Bench Relative Humidity run.

| Sensor | Predictor | Slope | Y-Intercept | $R^2$ | RMSE |
|--------|-----------|-------|-------------|-------|------|
| K30_21 | Pressure | -0.28 | 27483.54 | 0.32 | 1.48 |
| | Temperature | -0.95 | 419.17 | 0.29 | 1.51 |
| | Relative Humidity | 0.02 | 395.46 | 0.02 | 1.77 |
| | Ref_IN | 0.59 | 161.25 | 0.73 | 0.92 |
| K30_22 | Pressure | -0.08 | 7801.26 | 0.03 | 1.52 |
| | Temperature | -0.68 | 411.93 | 0.2 | 1.39 |
| | Relative Humidity | 0.05 | 393.19 | 0.15 | 1.43 |
| | Ref_IN | 0.38 | 246.1 | 0.4 | 1.2 |

**9 Ideal Pressure Time Response**

[Figure]

**Figure 43.** Example of idealized signal for pressure time-response correction. It is important to note that the referred idealized signal is an artificial signal with the pressure error but without the pressure time-response error. Therefore, ideal means the ideal impact of pressure (without pressure time-response error). It serves only as an evaluator of the pressure time-response algorithms. It should not be used for any pressure correction algorithms. The ideal signal was generated using the timestamps of the pressure step changes and the $CO_2$ values of the K30 after stabilization, after the step change.

**10 Temperature Li-820 (Ref_OUT)**

In all experimental condition tables in this document, the reported temperature for the optical chamber of the Li-820 (a.k.a. Ref_OUT) is 50.91 degrees Celsius with its standard deviation equal to zero. This may appear to be a manuscript preparation error (e.g., a copy and paste error), but it is not. To investigate the matter we first evaluated if our dataset ever showed any temperature different than 50.91 for this sensor. At beginning of all experiments our loggers recorded the warm-up ramp of this sensor with temperatures below 50.91 (as can be seen in figures 44 and 45). However, after this warm-up period the sensor does not report a value different than 50.91. Analyzing the temperatures reported by the Li-840A (a.k.a. Ref_IN), the largest deviation reported by this sensor for all experiments was 0.02 degrees Celsius. Therefore it is possible that the analog to digital converter (ADC) in the Li-820 is not capable of detecting these small fluctuations. Another contributing factor to the standard

deviation equal to zero, is our data trimming strategy. We only calculates the deviation for the test periods. At the beginning of

65 each test period, the Li-820's heater has had at least one hour to stabilize temperature of the sensor's optical chamber. At this point we do not have any reason to believe any malfunction on the sensor.

[Figure]

**Figure 44.** Complete data series for the experimental conditions of the second run of the Mesonet Temperature and Relative humidity experiments showing temperatures lower than 50.91 for a minutes.

[Figure]

**Figure 45.** Zoomed view of figure 44 showing a temperature variation of the Li-820 (a.k.a. Ref_OUT) optical chamber.